# An updated version of the global interior ocean biogeochemical data product, GLODAPv2.2020

Are Olsen [1], Nico Lange [2], Robert M. Key [3], Toste Tanhua [2], Henry C. Bittig [4], Alex Kozyr [5], Marta Álvarez [6], Kumiko Azetsu-Scott [7], Susan Becker [8], Peter J. Brown [9], Brendan R. Carter [10,11], Leticia Cotrim da Cunha [12], Richard A. Feely [11], Steven van Heuven [13], Mario Hoppema [14], Masao Ishii [15], Emil Jeansson [16], Sara Jutterström [17], Camilla S. Landa [1], Siv K. Lauvset [16], Patrick Michaelis [2], Akihiko Murata [18], Fiz F. Pérez [19], Benjamin Pfeil [1], Carsten Schirnick [2], Reiner Steinfeldt [20], Toru Suzuki [21], Bronte Tilbrook [22], Anton Velo [19], Rik Wanninkhof [23], Ryan J. Woosley [24]

[1] Geophysical Institute, University of Bergen and Bjerknes Centre for Climate Research, Bergen, Norway
[2] GEOMAR Helmholtz Centre for Ocean Research Kiel, Kiel, Germany
[3] Atmospheric and Oceanic Sciences, Princeton University, Princeton, NJ, 08540, USA
[4] Leibniz Institute for Baltic Sea Research Warnemünde, Rostock, Germany
[5] NOAA National Centers for Environmental Information, Silver Spring, MD, USA
[6] Instituto Español de Oceanografía, A Coruña, Spain
[7] Departement of Fisheries and Oceans, Bedford Institute of Oceanography, Dartmouth, Nova Scotia, Canada
[8] UC San Diego, Scripps Institution of Oceanography, San Diego CA 92093, USA
[9] National Oceanography Centre, Southampton, UK
[10] Cooperative Institute for Climate Ocean and Ecosystem Studies, University Washington, Seattle, Washington, USA
[11] Pacific Marine Environmental Laboratory, National Oceanic and Atmospheric Administration, Seattle, Washington, USA
[12] Faculdade de Oceanografia, Universidade do Estado do Rio de Janeiro, Rio de Janeiro (RJ), Brazil
[13] Centre for Isotope Research, Faculty of Science and Engineering, University of Groningen, Groningen, the Netherlands
[14] Alfred Wegener Institute Helmholtz Centre for Polar and Marine Research, Bremerhaven, Germany
[15] Oceanography and Geochemistry Research Department, Meteorological Research Institute, Japan Meteorological Agency, Tsukuba, Japan
[16] NORCE Norwegian Research Centre, Bjerknes Centre for Climate Research, Bergen, Norway
[17] IVL Swedish Environmental Research Institute, Gothenburg, Sweden
[18] Research Institute for Global Change, Japan Agency for Marine-Earth Science and Technology, Yokosuka, Japan
[19] Instituto de Investigaciones Marinas, IIM – CSIC, Vigo, Spain
[20] University of Bremen, Institute of Environmental Physics, Bremen, Germany
[21] Marine Information Research Center, Japan Hydrographic Association, Tokyo, Japan
[22] CSIRO Oceans and Atmosphere and Antarctic Climate and Ecosystems Co-operative Research Centre, University of Tasmania, Hobart, Australia
[23] Atlantic Oceanographic and Meteorological Laboratory, National Oceanic and Atmospheric Administration, Miami, USA.
[24] Center for Global Change Science, Massachusetts Institute for Technology, Cambridge, Massachusetts, USA

*Correspondence to*: Are Olsen (are.olsen@uib.no)

**Abstract.** The Global Ocean Data Analysis Project (GLODAP) is a synthesis effort providing regular compilations of surface-to-bottom ocean biogeochemical data, with an emphasis on seawater inorganic carbon chemistry and related variables determined through chemical analysis of seawater samples. GLODAPv2.2020 is an update of the previous version, GLODAPv2.2019. The major changes are: data from 106 new cruises added, extension of time coverage to 2019, and the inclusion of available (also for historical cruises) discrete fugacity of $CO_2$ ($fCO_2$) values in the merged product files. GLODAPv2.2020 now includes measurements from more than 1.2 million water samples from the global oceans collected on 946 cruises. The data for the 12 GLODAP core variables (salinity, oxygen, nitrate, silicate, phosphate, dissolved inorganic carbon, total alkalinity, pH, CFC-11, CFC-12, CFC-113, and $CCl_4$) have undergone extensive quality control with a focus on systematic evaluation of bias. The data are available in two formats: (i) as submitted by the data originator but updated to WOCE exchange format and (ii) as a merged data product with adjustments applied to minimize bias. These adjustments were derived by comparing the data from the 106 new cruises with the data from the 840 quality-controlled cruises of the GLODAPv2.2019 data product using crossover analysis. Comparisons to empirical algorithm estimates provided additional context for adjustment decisions, this is new to this version. The adjustments are intended to remove potential biases from errors related to measurement, calibration, and data handling practices without removing known or likely time trends or variations in the variables evaluated. The compiled and adjusted data product is believed to be consistent to better than 0.005 in salinity, 1 % in oxygen, 2 % in nitrate, 2 % in silicate, 2 % in phosphate, 4 $\mu$mol kg$^{-1}$ in dissolved inorganic carbon, 4 $\mu$mol kg$^{-1}$ in total alkalinity, 0.01–0.02 in pH (depending on region), and 5 % in the halogenated transient tracers. The other variables included in the compilation, such as isotopic tracers and discrete $fCO_2$, were not subjected to bias comparison or adjustments.

The original data, their documentation and doi codes are available at the Ocean Carbon Data System of NOAA NCEI (https://www.nodc.noaa.gov/ocads/oceans/GLODAPv2_2020/, last access: 20 June 2020). This site also provides access to the merged data product, which is provided as a single global file and as four regional ones – the Arctic, Atlantic, Indian, and Pacific oceans – under https://doi.org/10.25921/2c8h-sa89 (Olsen et al., 2020). These bias-adjusted product files also include significant ancillary and approximated data. These were obtained by interpolation of, or calculation from, measured data. This living data update documents the GLODAPv2.2020 methods and provides a broad overview of the secondary quality control procedures and results.

## 1 Introduction

The oceans mitigate climate change by absorbing both atmospheric $CO_2$ corresponding to a significant fraction of anthropogenic $CO_2$ emissions (Friedlingstein et al., 2019; Gruber et al., 2019) and most of the excess heat in the Earth System caused by the enhanced greenhouse effect (Cheng et al., 2020; Cheng et al., 2017). The objective of GLODAP (Global Ocean Data Analysis Project, www.glodap.info, last access: 25 May 2020) is to ensure provision of high quality and bias-corrected water column bottle data from the ocean surface to bottom that document the state and the evolving changes in physical and chemical ocean properties, e.g., the inventory of the excess $CO_2$ in the ocean, natural oceanic carbon, ocean acidification, ventilation rates, oxygen levels, and vertical nutrient transports. The core quality-controlled and bias-adjusted variables are salinity, dissolved oxygen, inorganic macronutrients (nitrate, silicate, and phosphate), seawater $CO_2$ chemistry variables (dissolved inorganic carbon – $TCO_2$, total alkalinity – TAlk, and pH on the total H$^+$ scale), and the halogenated transient tracers chlorofluorocarbon-11 (CFC-11), CFC-12, CFC-113, and $CCl_4$.

Other chemical tracers are usually measured on the cruises included in GLODAP. A subset of these data is distributed as part of the product but has not been extensively quality controlled or checked for measurement biases in this effort. For some of these variables better sources of data may exist, for example the product by Jenkins et al. (2019) for helium isotope and tritium data. GLODAP also includes derived variables to facilitate interpretation, such as potential density anomalies and apparent oxygen utilization (AOU). A full list of variables included in the product is provided in Table 1.

The oceanographic community largely adheres to principles and practices for ensuring open access to research data, such as the FAIR (Findable, Accessible, Interoperable, Reusable) initiative (Wilkinson et al., 2016), but the plethora of file formats and different levels of documentation, combined with the need to retrieve data on a per cruise basis from different access points, limits the realization of their full scientific potential. For biogeochemical data there is the added complexity of different levels of standardization and calibration, and even different units used for the same variable, such that the comparability between data sets is often poor. Standard operating procedures have been developed for some variables (Dickson et al., 2007; Hood et al., 2010; Hydes et al., 2012) and certified reference materials (CRM) exist for seawater $TCO_2$ and TAlk measurements (Dickson et al., 2003) and for nutrients in seawater (CRMNS; Aoyama et al., 2012; Ota et al., 2010). Despite this, biases in data still occur. These can arise from poor sampling and preservation practices, calibration procedures, instrument design, and inaccurate calculations. The use of CRMs does not by itself ensure accurate measurements of seawater $CO_2$ chemistry (Bockmon and Dickson, 2015), and the CRMNS have only become available recently and are not universally used. For salinity and oxygen, lack of calibration of the data from conductivity-temperature-depth (CTD) profiler mounted sensors is an additional and widespread problem, particularly for oxygen (Olsen et al., 2016). For halogenated transient tracers, uncertainties in standard gas composition, extracted water volume, and purge efficiency typically provide the largest sources of uncertainty. In addition to bias, occasional outliers occur. In rare cases poor precision - many multiples worse than that expected with current measurement techniques - can render a set of data of limited use. GLODAP deals with these issues by presenting the data in a uniform format, including any meta data either publicly-available or submitted by the data originator, and by subjecting the data to primary and secondary quality control assessments, focusing on precision and consistency, respectively. The secondary quality control focuses on deep data, where natural variability is minimal. Adjustments are applied to the data to minimize cases of bias that could be confidently established relative to the measurement precision for the variables and cruises considered.

GLODAPv2.2020 builds on earlier synthesis efforts for biogeochemical data obtained from research cruises, GLODAPv1.1 (Key et al., 2004; Sabine et al., 2005), Carbon dioxide in the Atlantic Ocean (CARINA) (Key et al., 2010), Pacific Ocean Interior Carbon (PACIFICA) (Suzuki et al., 2013), and notably GLODAPv2 (Olsen et al., 2016). GLODAPv1.1 combined data from 115 cruises with biogeochemical measurements from the global ocean. The vast majority of these were the sections covered during the World Ocean Circulation Experiment and the Joint Global Ocean Flux Study (WOCE/JGOFS) in the 1990s, but data from important "historical" cruises were also included, such as from the Geochemical Ocean Sections Study (GEOSECS), Transient Traces in the Ocean (TTO), and South Atlantic Ventilation Experiment (SAVE). GLODAPv2 was released in 2016 with data from 724 scientific cruises, including those from GLODAPv1.1, CARINA, PACIFICA, and data from 168 additional cruises. A particularly important source of data were the cruises executed within the framework of the "repeat hydrography" program (Talley et al., 2016), instigated in the early 2000s as part of the Climate and Ocean: Variability, Predictability and Change (CLIVAR) program and since 2007 organized as the Global Ocean Ship-based Hydrographic Investigations Program (GO-SHIP) (Sloyan et al., 2019). GLODAPv2 is now updated regularly using the "living data format" of *Earth System Science Data* to document significant additions and changes to the dataset.

Within this there are two types of GLODAP updates: full and intermediate. Full updates involve a reanalysis, notably crossover and inversion, of the entire dataset (both historical and new cruises) and all adjustments are subject to change. This was carried out for GLODAPv2. For intermediate updates, recently-available data are added following quality control procedures to ensure their consistency with the cruises included in the latest GLODAP release. Except for obvious outliers and similar types of errors (Sect. 3.3.1), the data included in previous releases are not changed during intermediate updates. Additionally, the GLODAP mapped climatologies (Lauvset et al., 2016) are not updated for these intermediate products. A naming convention has been introduced to distinguish intermediate from full product updates. For the latter the version number will change, while for the former the year of release is appended. The exact version number and release year (if appended) of the product used should always be reported in studies, rather than making a generic reference to GLODAP.

Creating and interpreting inversions, and other checks of the full data set needed for full updates are too demanding in terms of time and resources to be preformed every year or two-years. The aim is to conduct a full analysis (i.e., including an inversion) again after the third GO-SHIP survey has been completed. This completion is currently scheduled for 2023, and we anticipate that GLODAPv3 will become available a few years thereafter. In the intermin, presented here is is the second intermediate update, which adds data from 106 new cruises to the last update, GLODAPv2.2019 (Olsen et al., 2019).

**2 Key features of the update**

GLODAPv2.2020 (Olsen et al., 2020) contains data from 946 cruises, covering the global ocean from 1972 to 2019, compared to 840 for the period 1972-2017 for GLODAPv2.2019. Information on the 106 cruises added to this version is provided in Table A1 in the Appendix. Cruise sampling locations are shown alongside those of GLODAPv2.2019 in Fig. 1, while the coverage in time is shown in Fig. 2. Not all cruises have data for all of the above-mentioned 12 core variables; for example, cruises with only seawater $CO_2$ chemistry or transient tracer data are still included even without accompanying nutrient data due to their value towards computation of, for example, carbon inventories. In some other cases, cruises without any of these properties measured were included – this was because they did contain data for other carbon related tracers such as carbon isotopes, with the main intention of ensuring their wider availability. The added cruises are from the years 2004-2019, with most being more recent than 2010. The majority of the new data were obtained from the two vessels RV *Keifu Maru II* and RV *Ryofu Maru III*, which are operated by the Japan Meteorological Agency in the western North Pacific (Oka et al., 2018; Oka et al., 2017). Another important addition is the data collected across the Davis Strait between Canada and Greenland, from 10 cruises between 2004-2015 through a collaboration between the Bedford Institute of Oceanography, Canada and the University of Washington, USA (Azetsu-Scott et al., 2012). Other cruises from the Atlantic include those carried out on the RV *Maria S. Merian* and RV *Meteor*, with transient tracer data but not nutrients or seawater $CO_2$ chemistry data; the 2016 occupation of the OVIDE line (Pérez et al., 2018); the 2019 occupation of A17 onboard RV *Hesperides*; the 2018 occupation of A9.5 onboard RRS *James Cook* (King et al., 2019); and A02 on the RV *Celtic Explorer* in 2017 (McGrath et al., 2019). Two older North Atlantic cruises that did not find their way into GLODAPv2 have been added, a 2008 occupation of AR07W including more extensive subpolar NA sampling (35TH20080825) and a 2007 RV *Pelagia* cruise (64PE20071026) covering the Northeast Atlantic. The final Atlantic cruise is 29GD20120910 onboard RV *Garcia del Cid*, with measurements for stable isotopes of carbon and oxygen ($\delta^{13}C$ and $\delta^{18}O$) off the Iberian Peninsula (Voelker et al., 2015) but no data for nutrients, seawater $CO_2$

chemistry, or transient tracers. Two new Indian Ocean cruises are included, both took place in the far south, in the Indian sector of the Southern Ocean: an Argo deployment cruise south and west of Kerguelen Island onboard the RV *S. A. Agulhas I*, and the 2018 occupation of GO-SHIP line SR03 onboard the RV *Investigator*. The JOIS cruise in 2015 is the sole addition for the Arctic. Finally, new data along the US West Coast are from two cruises conducted on board the RVs
*Wecoma* (WCOA2011, 32WC20110812) and *Ronald H. Brown* (WCOA2016, 33RO20160505) as part of NOAA's ocean acidification program.

All new cruises were subjected to primary (Sect. 3.1) and secondary (Sect. 3.2) quality control (QC). These procedures are essentially the same as for GLODAPv2.2019, aiming to ensure the consistency of the data from the 106 new cruises with the previous release of this data product (in this case, the GLODAPv2.2019 adjusted data product).

**3 Methods**

**3.1 Data assembly and primary quality control**

The data from the 106 new cruises were submitted directly to us or retrieved from data centers: typically the CLIVAR and Carbon Hydrographic Data Office (https://cchdo.ucsd.edu, last access: 20 October 2020), National Center for Environmental Information (https://www.ncei.noaa.gov, last access 20 October 2020), and PANGAEA
(https://pangaea.de, last access 20 October 2020). Each cruise is identified by an expedition code (EXPOCODE). The EXPOCODE is guaranteed to be unique and constructed by combining the country code and platform code with the date of departure in the format YYYYMMDD. The country and platform codes were taken from the ICES (International Council for the Exploration of the Sea) library (https://vocab.ices.dk/, last access: 20 June 2020).

The individual cruise data files were converted to the WOCE exchange format: a comma delimited ASCII format for
CTD and bottle data from hydrographic cruises. GLODAP deals only with bottle data and CTD data at bottle trip depths, and their exchange format is briefly reviewed here with full details provided in Swift and Diggs (2008). The first line of each exchange file specifies the data type, in the case of GLODAP this is "BOTTLE", followed by a date and time stamp and identification of the group and person who prepared the file, e.g., "PRINUNIVRMK" is Princeton University, Robert M. Key. Next follows the README section; this provides brief cruise specific information, such as dates, ship, region,
method plus quality notes for each variable measured, citation information, and references to any papers that used or presented the data. The README information was typically assembled from the information contained in the metadata submitted by the data originator. In some cases, issues noted during the primary QC and other information such as file update notes are included. The only rule for the README section is that it must be concise and informative. The README is followed by data column headers, units, and then the data. The headers and units are standardized and
provided in Table 1 for the variables included in GLODAP. Exchange file preparation required unit conversion in some cases, most frequently from milliliters per liter (mL $L^{-1}$; oxygen) or micromoles per liter ($\mu$mol $L^{-1}$; nutrients) to micromoles per kilogram of seawater ($\mu$mol $kg^{-1}$). The default conversion procedure for nutrients was to use seawater density at reported salinity, an assumed measurement-temperature of 22 ºC, and pressure of 1 atm. For oxygen, the factor 44.66 was used for the "milliliters of oxygen" to "micromoles of oxygen" conversion, while the density required for the
"per liter" to "per kilogram" conversion was calculated from the reported salinity and draw temperatures whenever possible. However, potential density was used instead when draw temperature was not reported. The potential errors introduced by any of these procedures are insignificant. Missing numbers are indicated by -999.

Each data column (except temperature and pressure, which are assumed "good" if they exist) has an associated column of data flags. For the original data exchange files, these flags conform to the WOCE definitions for water samples and are listed in Table 2. For the merged and adjusted product files these flags are simplified: questionable (WOCE flag 3) and bad (WOCE flag 4) data are removed and their flags are set to 9. The same procedure is applied to data flagged 8 (very few such data exist); WOCE flags 1 (Data not received) and 5 (Data not reported) are also set to 9, while flags of 6 (Mean of replicate measurements) and 7 (Manual chromatographic peak measurement) are set to 2, if the data appear good. Also, in the merged product files a flag of 0 is used to indicate a value that could be measured but is somehow approximated: for salinity, oxygen, phosphate, nitrate, and silicate, the approximation is conducted using vertical interpolation; for seawater $CO_2$ chemistry variables ($TCO_2$, TAlk, pH, and $fCO_2$), the approximation is conducted using calculation from two measured $CO_2$ chemistry variables (Sect 3.2.2). Importantly, interpolation of $CO_2$ chemistry variables is never performed and thus a flag value of 0 has a unique interpretation.

If no WOCE flags were submitted with the data, then they were assigned by us. Regardless, all incoming files were subjected to primary QC to detect questionable or bad data - this was carried out following Sabine et al. (2005) and Tanhua et al. (2010), primarily by inspecting property-property plots. Outliers showing up in two or more different such plots were generally defined as questionable and flagged. In some cases, outliers were detected during the secondary QC; the consequent flag changes have then also been applied in the GLODAP versions of the original cruise data files.

## 3.2 Secondary quality control

The aim of the secondary QC was to identify and correct any significant biases in the data from the 106 new cruises relative to GLODAPv2.2019, while retaining any signal due to temporal changes. To this end, secondary QC in the form of consistency analyses was conducted to identify offsets in the data. All identified offsets were scrutinized by the GLODAP reference group through a series of teleconferences during March and April 2020 in order to decide the adjustments to be applied to correct for the offset (if any). To guide this process, a set of initial minimum adjustment limits was used (Table 3). These are set according to the expected measurement precision for each variable, and are the same as those used for GLODAPv2.2019. In addition to the average magnitude of the offsets, factors such as the precision of the offsets, persistence towards the various cruises used in the comparison, regional dynamics, and the occurrence of time trends or other variations were considered. Thus, not all offsets larger than the initial minimum limits have been adjusted. A guiding principle for these considerations was to not apply an adjustment whenever in doubt. Conversely, in some cases where data and offsets were very precise and the cruise had been conducted in a region where variability is expected to be small, adjustments lower than the minimum limits were applied. Any adjustment was applied uniformly to all values for a variable and cruise, i.e., an underlying assumption is that cruises suffer from either no or a single and constant measurement bias. Adjustments for salinity, $TCO_2$, TAlk and pH are always additive, while adjustments for oxygen, nutrients and the halogenated transient traces are always multiplicative. Except where explicitly noted (Sect. 3.3.1), adjustments were not changed for data previously included in GLODAPv2.2019.

Crossover comparisons, multi-linear regressions (MLRs), and comparison of deep-water averages were used to identify offsets for salinity, oxygen, nutrients, $TCO_2$, TAlk, and pH (Sect. 3.2.2 and 3.2.3). In contrast to GLODAPv2 and GLODAPv2.2019, evaluation of the internal consistency of the seawater $CO_2$ chemistry variables was not used for the evaluation of pH (Sect. 3.2.4). New to the present version is more extensive use of two predictions from two empirical algorithms—"CArbonate system And Nutrients concentration from hYdrological properties and Oxygen using a Neural-network version B" (CANYON-B) and "CONsisTency EstimatioN and amounT" (CONTENT), (Bittig et al., 2018)—for

the evaluation of offsets in nutrients and seawater $CO_2$ chemistry data (Section 3.2.5). For the halogenated transient tracers, comparisons of surface saturation levels and the relationships among the tracers were used to assess the data consistency (Sect. 3.2.6). For salinity and oxygen, CTD and bottle values were merged into a "hybrid" variable prior to the consistency analyses (Sect. 3.2.1).

### 3.2.1 Merging of sensor and bottle data

Salinity and oxygen data can be obtained by analysis of water samples (bottle data) and/or directly from the CTD sensor pack. These two measurement types are merged and presented as a single variable in the product. The merging was conducted prior to the consistency checks, ensuring their internal calibration in the product. The merging procedures were only applied to the bottle data files, which commonly include values recorded by the CTD at the pressures where the water samples are collected. Whenever both CTD and bottle data were present in a data file, the merging step considered the deviation between the two and calibrated the CTD values if required and possible. Altogether seven scenarios are possible for each of the CTD-$O_2$ sensor properties individually, where the fourth (see below) never occurred during our analyses but is included to maintain consistency with GLODAPv2:

1. No data are available: no action needed.

2. No bottle values are available: use CTD values.

3. No CTD values are available: use bottle values.

4. Too few data of both types are available for comparison and more than 80 % of the records have bottle values: use bottle values.

5. The CTD values do not deviate significantly from bottle values: replace missing bottle values with CTD values.

6. The CTD values deviate significantly from bottle values: calibrate CTD values using linear fit with respect to bottle data and replace missing bottle values with the so-calibrated CTD values.

7. The CTD values deviate significantly from bottle values, and no good linear fit can be obtained for the cruise: use bottle values and discard CTD values.

The number of cases encountered for each scenario is summarized in Sect. 4.1.

### 3.2.2 Crossover analyses

The crossover analyses were conducted with the MATLAB toolbox prepared by Lauvset and Tanhua (2015) and with the GLODAPv2.2019 data product as the reference data product. The toolbox implements the 'running-cluster' crossover analysis first described by Tanhua et al. (2010). This analysis compares data from two cruises on a station-by-station basis and calculates a weighted mean offset between the two and its weighted standard deviation. The weighting is based on the scatter in the data such that data that have less scatter have a larger influence on the comparison than data with more scatter. Whether the scatter reflects actual variability or data precision is irrelevant in this context as increased scatter nevertheless decreases the confidence in the comparison. Stations are compared when they are within 2° arc distance (~ 200 km) of each other. Only deep data are used, to minimize the effects of natural variability. Either the 1500 or 2000 dbar depth surface was used as upper bound, depending on the number of available data, their variation at different depths, and the region in question. This was evaluated on a case-by-case basis by comparing crossovers with both depth limits and using the one that provided the most clear and robust information. In regions where deep mixing or convection occurs, such as the Nordic, Irminger and Labrador seas, the upper bound was always placed at 2000 dbar; while winter mixing in the first two regions is normally not deeper than this (Brakstad et al., 2019; Fröb et al., 2016),

convection beyond this limit has occasionally been observed in the Labrador Sea (Yashayaev and Loder, 2016). However, using an upper depth limit deeper than 2000 dbar will quickly give too few data for robust analysis. In addition, even below the deepest winter mixed layers properties do change over the time periods considered (e.g., Falck and Olsen, 2010), so this limit does not guarantee steady conditions. In the Southern Ocean deep convection beyond 2000 dbar seldom occurs, an exception being the processes accompanying the formation of the Weddell Polynya in the 1970s

(Gordon, 1978). Deep and bottom water formation usually occurs along the Antarctic coasts, where relatively thin nascent dense water plumes flow down the continental slope. We cautiously avoid such cases, which are easily recognizable. In order to avoid removing persistent temporal trends, all crossover results are also evaluated as a function of time (see below).

As an example of crossover analysis, the crossover for $TCO_2$ measured on the two cruises 49UP20160109, which is new

to this version, and 49UP20160703, which was included in GLODAPv2.2019, is shown in Fig. 3. For $TCO_2$ the offset is determined as the difference, as is the case for salinity, TAlk, and pH. For the nutrients, oxygen, and the halogenated transient tracers, ratios are used. This is in accordance with the procedures followed for GLODAPv2. The $TCO_2$ values from 49UP20160109 are higher, with a weighed mean offset of $3.62 \pm 2.67$ µmol kg$^{-1}$ compared to those measured on 49UP20160703.

For each of the 106 new cruises, such a crossover comparison was conducted against all possible cruises in GLODAPv2.2019, i.e., all cruises that had stations closer than 2° arc distance to any station for the cruise in question. The summary figure for $TCO_2$ on 49UP20160109 is shown in Fig. 4. The $TCO_2$ data measured on this cruise are high by $3.68 \pm 0.83$ µmol kg$^{-1}$ when compared to the data measured on nearby cruises included in GLODAPv2.2019. This is slightly less than the initial minimum adjustment limit for $TCO_2$ of 4 µmol kg$^{-1}$ (Table 3), but the offset is present against

all cruises and there is no obvious time trend (particularly important for $TCO_2$), and as such qualifies for an adjustment of the data in the merged data product. In this case -3 µmol kg$^{-1}$ was applied: this is somewhat less than indicated by the crossover analysis, but a smaller adjustment is supported by the CANYON-B and CONTENT results (Sect. 3.2.5). Adjustments are typically round numbers relative to the precision of the variable being considered (e.g., -3 not -3.4 for $TCO_2$ and 0.005 not 0.0047 for pH) to avoid the communicating that the ideal adjustments are known to high precision.

One exception to the above-described procedure exists, namely in the Sea of Japan where six new cruises were added. In this region, only two other cruises were included in GLODAPv2.2019. Therefore, all eight cruises were compared against each other and strong outliers were adjusted accordingly, instead of adjusting the six new cruises towards the existing two.

### 3.2.3 Other consistency analyses

MLR analyses and deep water averages, broadly following Jutterström et al. (2010), were also used for the secondary QC of salinity, oxygen, nutrients, $TCO_2$, and TAlk data. These approaches are particularly valuable when a cruise has either very few or no valid crossovers with GLODAPv2, but are used more generally to provide more insight on the consistency of the data. The latter was the case for the 106 new cruises; i.e., no adjustment decisions were reached on the basis of MLR and deep water average analyses alone. For the MLRs, the presence of bias in the data was identified by comparing

the MLR-generated values with the measured values. Both analyses were conducted on samples collected deeper than the 1500 or 2000 dbar pressure level to minimize the effects of natural variations, and both used available GLODAPv2.2019 data from within 2° of the cruise in question to generate the MLR or deep water average. The lower depth limit was set to the deepest sample for the cruise in question. For the MLRs, all of the above-mentioned variables could be included

among the independent variables (e.g., for a TAlk MLR, salinity, oxygen, nutrients, and $TCO_2$ were allowed), with the exact selection determined based on the statistical robustness of the fit, as evaluated using the coefficient of determination ($r^2$) and root mean square error (RMSE). MLRs based on variables that were suspect for the cruise in question were avoided (e.g., if oxygen appeared biased it was not included as an independent variable). The MLRs could be based on 10 to 500 samples, and the robustness of the fit ($r^2$, RMSE) and quantity of fitting data were considered when using the results to guide whether to apply a correction. The same applies for the deep-water averages (i.e., the standard deviation of the mean). MLR and deep-water average results showing offsets above the minimum adjustment limits were carefully scrutinized, along with available crossover values and CANYON-B and CONTENT estimates, to determine whether or not to apply an adjustment.

**3.2.4 pH scale conversion and quality control**

Altogether 82 of the 106 new cruises included measured pH data. For one of these, the pH data were not supplied on the total scale or at 25 °C and 0 dbar pressure, which is the GLODAP standard, and were thus converted. The conversion was conducted using CO2SYS (Lewis and Wallace, 1998) for MATLAB (van Heuven et al., 2011) with reported pH and TAlk as inputs, and generating pH output values at total scale at 25 °C and 0 dbar of pressure (named phts25p0 in the product). Missing TAlk data were approximated as 67 times salinity. The proportionality (67) is the mean ratio of TAlk to salinity in GLODAPv2 data. The uncertainties introduced with this approximation are negligible (order $10^{-7}$ pH units) for the scale conversions and order $10^{-3}$ pH units for the temperature and pressure conversion (evaluated by repeating conversions with 2 times the standard deviation of the ratio, i.e., $67 \pm 4.1$). This is sufficiently accurate relative to other sources of uncertainty, which are discussed below. Data for phosphate and silicate are also needed, and were, whenever missing, determined using CANYON-B (Bittig et al., 2018). The conversion was conducted with the carbonate dissociation constants of Lueker et al. (2000), the bisulfate dissociation constant of Dickson (1990), and the borate-to-salinity ratio of Uppström (1974). These procedures are the same as used for GLODAPv2.2019 (Olsen et al., 2019).

In contrast to past GLODAP pH QC, evaluation of the internal consistency of $CO_2$ system variables was not used for the secondary quality control of the pH data of the 106 new cruises; only crossover analysis was used, supplemented by CONTENT and CANYON-B (Sect. 3.2.5). Recent literature has demonstrated that internal consistency evaluation procedures are subject to errors owing to incomplete understanding of the thermodynamic constants, major ion concentrations, measurement biases, and potential contribution of organic compounds or other unknown protolytes to alkalinity (Takeshita et al., 2020), which lead to pH dependent offsets in calculated pH (Álvarez et al., 2020; Carter et al., 2018): these may be interpreted as biases and generate false corrections. The offsets are particularly strong at pH levels below 7.7, when calculated and measured pH are different by on average between 0.01 and 0.02 units. For the North Pacific this is a problem as pH values below 7.7 can occur at the depths interrogated during the QC (>1500 dbar for this region, Olsen et al., 2016). Since any corrections, which may thus be an artifact, are applied to the full profiles, we assign an uncertainty of 0.02 to the North Pacific pH data in the merged product files. Elsewhere, the uncertainties that have arisen are smaller, since deep pH is typically larger than 7.7 (Lauvset et al., 2020), and at such levels the difference between calculated and measured pH is less than 0.01 on average (Álvarez et al., 2020; Carter et al., 2018). Outside the North Pacific, we believe, therefore that the pH data are consistent to 0.01. Avoiding interconsistency considerations for these intermediate products helps to reduce the problem, but since the reference data set (also as used for the generation of the CANYON-B and CONTENT algorithms) has these issues, a full re-evaluation, envisioned for GLODAPv3, is needed to address the problem satisfactorily.

### 3.2.5 CANYON-B and CONTENT analyses

CANYON-B and CONTENT (Bittig et al., 2018) were used to support decisions regarding application of adjustments (or not). CANYON-B is a neural network for estimating nutrients and seawater $CO_2$ chemistry variables from temperature, salinity, and oxygen. CONTENT additionally considers the consistency among the estimated $CO_2$ chemistry variables to further refine them. These approaches were developed using the data included in the GLODAPv2 data product. Their advantage compared to crossover analyses for evaluating consistency among cruise data is that effects of water mass changes on ocean properties are represented in the non-linear relationships in the underlying neural network. For example, if elevated nutrient values are measured on a cruise but are not due to a measurement bias but actual aging of the water mass(es) that have been sampled and as such accompanied by a decrease in oxygen concentrations, the measured values and the CANYON-B estimates will be similar. Vice-versa, if the nutrient values are biased, the measured values and CANYON-B predictions will be dissimilar.

Used in the correct way and with caution this tool is a powerful supplement to the traditional crossover analyses. Specifically, we gave no weight to comparisons where the crossover analyses had suggested that the S and/or $O_2$ data were biased as this would lead to error in the predicted values. We also considered the uncertainties of the CANYON-B and CONTENT estimates. These uncertainties are determined for each predicted value, and for each comparison the ratio of the difference (between measured and predicted values) to the local uncertainty was used to gauge the comparability. As an example, the CANYON-B/CONTENT analyses of the data obtained at 49UP20160109 are presented in Fig. 5. The CANYON-B and CONTENT results confirmed the positive offset in the $TCO_2$ values revealed in the crossover comparisons discussed in Sect. 3.2.2. The magnitude of the inconsistency for the CANYON-B estimate was 3.4 μmol kg$^{-1}$, i.e., slightly less than that the weighted mean crossover offset of 3.7 μmol kg$^{-1}$, while the CONTENT estimate gave an inconsistency of 2.7 μmol kg$^{-1}$. The differences between these consistency estimates owes to differences in the actual approach, the weighting across stations, stations considered (i.e., crossover comparisons use only stations within ~200 km of each other, while CANYON-B and CONTENT considers all stations where necessary variables are sampled, and depth range considered (> 500 dbar for CANYON-B and CONTENT vs. >1500/2000 dbar for crossovers). The specific difference between the CANYON-B and CONTENT estimates is a result of the seawater $CO_2$ chemistry considerations by the latter. For the other variables, the inconsistencies are low and agree with the crossover results (not shown here but results can be accessed through the Adjustment Table) with the exception of pH. The pH results are further discussed in Sect. 4.2.

Another advantage of CANYON-B and CONTENT is that these procedures provide estimates at the level of individual data points, e.g., pH values are determined for every sampling location and depth where T, S, and $O_2$ data are available. Cases of strong differences between measured and estimated values are always examined. This has helped to identify primary QC issues for some variables and cruises, for example a case of an inverted pH profile at cruise 32PO20130829, which has been amended.

### 3.2.6 Halogenated transient tracers

For the halogenated transient tracers (CFC-11, CFC-12, CFC-113, and $CCl_4$; CFCs for short) inspection of surface saturation levels and evaluation of relationships between the tracers for each cruise were used to identify biases, rather than crossover analyses. Crossover analysis is of limited value for these variables given their transient nature and low concentrations at depth. As for GLODAPv2, the procedures were the same as those applied for CARINA (Jeansson et al., 2010; Steinfeldt et al., 2010).

**3.3 Merged product generation**

The merged product file for GLODAPv2.2020 was created by correcting known issues in the GLODAPv2.2019 merged file, and then appending a merged and bias-corrected file containing the 106 new cruises to this error-corrected GLODAPv2.2019 file.

**3.3.1 Updates and corrections for GLODAPv2.2019**

Several minor omissions and errors have been identified in the GLODAPv2 and v2.2019 data products since their release in 2016 and 2019, respectively. Most of these have been corrected in this release. In addition, some recently available data have been added for a few cruises. The changes are:

– For cruise 33RR20160208, the CFC-113 data of station 31 were found to be bad and have been removed. Additionally, the flags for CFC-11, CFC-12, $SF_6$ and $CCl_4$ were replaced with new ones received from the Principal Investigator, and recently published data for $\delta^{13}C$ and $\Delta^{14}C$ have been added to the product file.

– For 18HU20150504, the pH data measured at stations 196, 200, and 203 were found offset by approximately +0.1 units, because such large offset points to general data quality problems, these data have been removed.

– For 32PO20130829, pH values of station 133 cast 1 were in the wrong order in the file. This has been amended. Additionally, pH values from cast 2 at this station were deemed questionable and have been removed.

– For 33RR20050109, the $\delta^{13}C$ values of station 7 bottle 32 and station 16 bottle 22 were found bad (values were less than -6 ‰) and have been removed from the product file.

– For 35MF19850224, the $\delta^{13}C$ value of station 21 cast 3 bottle 4 was found bad and has been removed.

– For 74JC20100319 the $\delta^{13}C$ value at station 37 bottle 7 was found bad and has been removed.

– All $\delta^{13}C$ values from the large volume Gerard barrels (identified by bottle number greater than 80) were removed from the product files as these values often have poor precision and accuracy related to gas extraction procedures.

– For 33HQ20150809, temperatures of station 52 cast 1 were found bad (less than -2 °C) and have been removed, hence all other samples were removed for this cast as well (the same depths and variables were sampled at the other casts, however). Temperatures for casts 2 and 8 were replaced with updated values; these changes are very minor, on the order of 0.001 $^o$C.

– For cruises 33RO20110926, 33RO20150525, and 33RO20150410, $\delta^{13}C$ and $\Delta^{14}C$ data have become available and were added to the product.

– Ship code for all RV *Maria S. Merian* cruises have been changed from MM to M2.

– For cruises 49SH20081021 and 49UF20121024, an adjustment of + 6 μmol kg$^{-1}$ is now applied to the $TCO_2$ values.

– Additional primary QC have been applied to the cruises with *Keifu Maru II* and *Ryofu Maru III* that were included in GLODAPv2.2019.

– Neutral density values in GLODAPv2 and GLODAPv2.2019 had been calculated using the polynomial approximation of Sérazin (2011). All of these values were replaced with neutral density calculated following Jackett and McDougall (1997).

– Discrete $fCO_2$ data are now included in the product files whenever available. Discrete $fCO_2$ is one of the variables that describe seawater $CO_2$ chemistry, but is rarely measured and has not been included in GLODAP product files before, in particular as a result of apparent quality issues that were not fully understood during the secondary QC

for GLODAPv1.1 (Sabine et al., 2005). However, for some cruises $fCO_2$ data were included indirectly in both GLODAPv1.1 and GLODAPv2 as they had been used in combination with $TCO_2$ to calculate TAlk. We have now chosen to include the discrete $fCO_2$ values in the product files. This increases transparency and traceability of the product; the $fCO_2$ data are also highly relevant for ongoing efforts toward resolving recently identified inconsistencies in our understanding of the relationships among the seawater $CO_2$ chemistry variables (Carter et al., 2018; Fong and Dickson, 2019; Takeshita et al., 2020; Álvarez et al., 2020). A total of 33 924 discrete $fCO_2$ measurements from 34 cruises conducted between 1983-2014 are now included. All values were converted to 20° C and 0 dbar pressure using CO2SYS for MATLAB (van Heuven et al., 2011). This was also used for the conversion of partial pressure of $CO_2$ ($pCO_2$) to $fCO_2$ for the 20 cruises where $pCO_2$ was reported. The procedures for these conversions, in terms of dissociation constants and approximation of missing variables, were the same as for the pH conversions (Sect. 3.2.4). These $fCO_2$ data have not been subjected to secondary QC. The inclusion of discrete $fCO_2$ data has led to some changes in the calculations of missing seawater $CO_2$ chemistry variables; these are described towards the end of the next section.

### 3.3.2 Merging

The new data were merged into a bias-minimized product file following the procedures used for GLODAPv1.1 (Key et al., 2004; Sabine et al., 2005), CARINA (Key et al., 2010), PACIFICA (Suzuki et al., 2013), GLODAPv2 (Olsen et al., 2016), and GLODAPv2.2019 (Olsen et al., 2019), with some modifications:

– Data from the 106 new cruises were merged and sorted according to EXPOCODE, station, and pressure. GLODAP cruise numbers were assigned consecutively, starting from 2001, so they can be distinguished from the GLODAPv2.2019 cruises that ended at 1116.

– For some cruises the combined concentration of nitrate and nitrite was reported instead of nitrate. If explicit nitrite concentrations were also given, these were subtracted to get the nitrate values. If not, the combined concentration was renamed to nitrate. As nitrite concentrations are very low in the open ocean, this has no practical implications.

– When bottom depths were not given, they were approximated as the deepest sample pressure +10 dbar or extracted from ETOPO1 (Amante and Eakins, 2009), whichever was greater. For GLODAPv2, bottom depths were extracted from the Terrain Base (National Geophysical Data Center/NESDIS/NOAA/U.S. Department of Commerce, 1995). The intended use of this variable is only drawing approximate bottom topography for sections.

– Whenever temperature was missing in the original data file, all data for that record were removed and their flags set to 9. The same was done when both pressure and depth were missing. For all surface samples collected using buckets or similar, the bottle number was set to zero. There are some exceptions to this, in particular for cruises that also used Gerard barrels for sampling. These may have valuable tracer data that are not accompanied by a temperature, so such data have been retained.

– All data with WOCE quality flags 3, 4, 5, or 8 were excluded from the product files and their flags set to 9. Hence, in the product files a flag 9 can indicate not measured (as is also the case for the original exchange formatted data files) or excluded from the product; in any case, no data value appears. All flags 6 (replicate measurement) and 7 (manual chromatographic peak measurement) were set to 2, provided the data appeared good.

– Missing sampling pressures (depths) were calculated from depths (pressures) following UNESCO (1981).

– For both oxygen and salinity, CTD and bottle values were merged following procedures summarized in Sect. 3.2.1.

- Missing salinity, oxygen, nitrate, silicate, and phosphate values were vertically interpolated whenever practical, using a quasi-Hermetian piecewise polynomial. "Whenever practical" means that interpolation was limited to the vertical data separation distances given in Table 4 in Key et al. (2010). Interpolated salinity, oxygen, and nutrient values have been assigned a WOCE quality flag 0.
- The data for the 12 core variables were corrected for bias using the adjustments determined during the secondary QC.
- Values for potential temperature and potential density anomalies (referenced to 0, 1000, 2000, 3000, and 4000 dbar) were calculated using Fofonoff (1977) and Bryden (1973). Neutral density was calculated using Jackett and McDougall (1997), thus neutral density for all 946 cruises are calculated using this procedure
- Apparent oxygen utilization was determined using the combined fit in Garcia and Gordon (1992).
- Partial pressures for CFC-11, CFC-12, CFC-113, CCl4, and $SF_6$ were calculated using the solubilities by Warner and Weiss (1985), Bu and Warner (1995), Bullister and Wisegarver (1998), and Bullister et al. (2002).
- Missing seawater $CO_2$ chemistry variables were calculated whenever possible. The procedures for these calculations have been slightly altered as the product now contains four such variables; earlier versions of GLODAPv2 (Olsen et al., 2016; Olsen et al., 2019) included only three, so whenever two were included the one to calculate was unequivocal. Four $CO_2$ chemistry variables gives more degrees of freedom in this respect, e.g., a particular record may have measured data for $TCO_2$, TAlk, and pH, and then a choice needs to be made with regard to which pair to use for the calculation of $fCO_2$. We followed two simple principles. First, $TCO_2$ and TAlk was the preferred pair to calculate pH and $fCO_2$, because we have higher confidence in the $TCO_2$ and TAlk data than pH (given the issues summarized in Sect. 3.2.4) and $fCO_2$ (because it was not subjected to secondary QC). Second, if either $TCO_2$ or TAlk was missing and both pH and $fCO_2$ data existed, pH was preferred (because $fCO_2$ has not been subjected to secondary QC). All other combinations involve only two measured variables. The calculations were conducted using CO2SYS (Lewis and Wallace, 1998) for MATLAB (van Heuven et al., 2011), with the constants set as for the pH conversions (Sect. 3.2.4). For calculations involving $TCO_2$, TAlk, and pH, if less than a third of the total number of values, measured and calculated combined, for a specific cruise were measured, then all these were replaced by calculated values. The reason for this is that secondary QC of the few measured values was often not possible in such cases, for example due to a limited number of deep data available. Such replacements were not done for calculations involving $fCO_2$, as this would either overwrite all measured $fCO_2$ values or would entail replacing a measured variable that has been subjected to secondary QC (i.e., $TCO_2$, TAlk, or pH) with one calculated from a variable that has not been subjected to secondary QC (i.e., $fCO_2$). Calculated seawater $CO_2$ chemistry values have been assigned WOCE flag 0. Seawater $CO_2$ chemistry values have not been interpolated, so the interpretation of the 0 flag is unique.
- The resulting merged file for the 106 new cruises was appended to the merged product file for GLODAPv2.2019.

**4 Secondary quality control results and adjustments**

All material produced during the secondary QC is available via the online GLODAP Adjustment Table hosted by GEOMAR, Kiel, Germany at https://glodapv2-2020.geomar.de/ (last access: 18 June 2020), and which can also be accessed through www.glodap.info. This is similar in form and function to the GLODAPv2 Adjustment Table (Olsen et al., 2016) and includes a brief written justification for any adjustments applied.

### 4.1 Sensor and bottle data merge for salinity and oxygen

Table 4 summarizes the actions taken for the merging of the CTD and bottle data for salinity and oxygen. For 81 % of the 106 cruises added with this update, both CTD and bottle data were included for salinity in the original cruise data files and for all these cruises the two data types were found to be consistent. This is similar to the GLODAPv2.2019 results. For oxygen, only 25 % of the cruises included both CTD $O_2$ and bottle values; this is much less than for GLODAPv2.2019 where 50 % of the cruises included both. Having both CTD and bottle values in the data files is highly preferred as the information is valuable for quality control (bottle mistrips, leaking Niskin bottles, and oxygen sensor drift are among the issues that can be revealed). The extent to which the bottle data (i.e., OXYGEN in the individual cruise exchange files) in reality is mislabeled CTD data (i.e., should be CTDOXY) is uncertain. Regardless, the large majority of the CTD and bottle oxygen were consistent and did not need any further calibration of the CTD values (23 out of 25 cruises), while for two cruises no good fit could be obtained and their CTD $O_2$ data are not included in the product.

### 4.2 Adjustment summary

The secondary QC has 5 different outcomes, provided there are data. These are summarized in Table 5, along with the corresponding codes that appear in the online Adjustment Table and that are also occasionally used as shorthand for decisions in the coming text. The level of secondary QC varies among the cruises. Specifically, in some cases data were too shallow or geographically too isolated for full and conclusive consistency analyses. A secondary QC flag has been included in the merged product files to enable their identification, with "0" used for variables and cruises not subjected to full secondary QC (corresponding to code -888 in Table 5) and "1" for variables and cruises that were subjected to full secondary QC. The secondary QC flags are assigned per cruise and variable, not for individual data points and are independent of—and included in addition to—the primary (WOCE) QC flag. For example, interpolated (salinity, oxygen, nutrients) or calculated ($TCO_2$, TAlk, pH) values, which have a primary QC flag 0, may have a secondary QC flag of 1 if the measured data these values are based on have been subjected to full secondary QC. Conversely, individual data points may have a secondary QC flag of 0, even if their primary QC flag is 2 (good data). A 0 flag means that data were too shallow or geographically too isolated for consistency analyses or that these analyses were inconclusive, but that we have no reasons to believe that the data in question are of poor quality. Prominent examples for this version are the 10 new Davis Strait cruises: no data were available in this region in GLODAPv2.2019, which, combined with complex hydrography and differences in sampling locations, rendered conclusive secondary QC impossible. As a consequence, most, but not all, of these data (some being excluded because of poor precision after consultation with the PI) are included with a secondary QC flag of 0.

The secondary QC actions for the 12 core variables and the distribution of applied adjustments are summarized in Table 6 and Fig. 6, respectively. For most variables, only a very small fraction of the data are adjusted: no salinity data, 1 % of oxygen and nitrate data, 2 % of $TCO_2$ data, 5 % of TAlk data, 7 % of phosphate data, and 9 % of silicate data are adjusted. For the CFCs, data from one of 16 cruises with CFC-11 are adjusted, while for CFC-12 and CFC-113 the fractions are two of 21 cruises and one of three cruises respectively. The magnitudes of the various adjustments applied are also small, overall. Thus, the tendency observed during the production of GLODAPv2.2019 remains, namely that the large majority of recent cruises are consistent with earlier releases of this product.

For the Sea of Japan cruises, (where two existed in GLODAPv2.2019 and six were added in this version - Sect. 3.2.2), the crossover results showed biased $TCO_2$ data for one of the older cruises (49HS20081021, which is now adjusted up by 6

µmol kg$^{-1}$), and biased TAlk data for two of the presently added cruises (49UF20111004 and 49UF20121024, adjusted up by 5 and 6 µmol kg$^{-1}$, respectively)

The quality control of pH data proved challenging for this version. The large majority of new pH data had been collected in the northwestern Pacific on cruises conducted by the Japan Meteorological Agency. Figure 7 shows the distribution of pH crossover offsets vs. GLODAPv2.2019. Most of the pH values are higher, some by up to 0.02 pH units; this is

considerable, particularly as the data that are compared are from deeper than 2000 dbar where no changes due to ocean acidification are expected. The challenging aspect lies in the fact that the data added are comparatively many (~ 70 cruises vs. ~ 130 already included in this region in v2.2019) and also are more recent (2010-2018 vs. 1993-2016). As such they might be of higher quality given advances in pH measurement techniques over the years. Adjusting a large fraction of the new cruises down (following the adjustment limit of 0.01) is not advisable. We therefore chose to not

adjust any pH data, but to exclude the most serious outliers from the product file (using a limit of |0.015|, which led to exclusion of pH data from five cruises) and include the rest of the data without adjustments. We expect that a crossover and inversion analysis of all pH data in the northwestern Pacific will provide more information on the consistency among the cruises, and such an analysis will be conducted for the next update. For now, some caution should be exercised if looking at trends in ocean pH in the northwestern Pacific using GLODAPv2.2020. The crossover and inversion might

also result in re-inclusion of the excluded data. The formal decision for the excluded outliers is therefore to "suspend" them (Table 6).

For the nutrients, adjustments were applied to maintain consistency with data included in GLODAPv2 and GLODAPv2.2019. An alternative goal for the adjustments would be maintaining consistency with data from cruises that employed CRMNS to ensure accuracy of nutrient analyses. Such a strategy was adopted by Aoyama (2020) for

preparation of the Global Nutrients Dataset 2013 (GND13), and is being considered for GLODAP as well. However, as this would require a re-evaluation of the entire data set, this will not occur until the next full update of GLODAP, i.e., GLODAPv3. For now, we note the overall agreement between the adjustments applied in these two efforts (Aoyama, 2020), and that most disagreements appear to be related to cases where no adjustments were applied in GLODAP. This can be related to the strategy followed for nutrients for GLODAPv2, where data from GO-SHIP lines were considered a

priori more accurate than other data. CRMNS are used for nutrients on most GO-SHIP lines.

The improvement in data consistency due to the secondary QC process is evaluated by comparing the weighted mean of the absolute offsets for all crossovers before and after the adjustments have been applied. This "consistency improvement" for core variables is presented in Table 7. The data for CFCs were omitted from these analyses for previously discussed reasons (Sect. 3.2.6). Globally, the improvement is modest. Considering the initial data quality, this

result was expected. However, this does not imply that the data initially were consistent everywhere. Rather, for some regions and variables there are substantial improvements when the adjustments are applied. For example, Arctic Ocean phosphate, Indian Ocean silicate and TAlk, and Pacific Ocean pH data all show considerable improvements. For the latter, the improvement is a result of exclusion of data and not application of adjustments, as discussed above.

The various iterations of GLODAP provide insight into initial data quality covering more than 4 decades. Figure 8

summarizes the applied absolute adjustment magnitude per decade. These distributions are broadly unchanged compared to GLODAPv2.2019 (Fig. 6 in Olsen et al., 2019). Most TCO$_2$ and TAlk data from the 1970s needed an adjustment, but this fraction steadily declines until only a small percentage is adjusted in recent years. This is encouraging and demonstrates the value of standardizing sampling and measurement practices (Dickson et al., 2007), the widespread use of CRMs (Dickson et al., 2003), and instrument automation. The pH adjustment frequency also has a downward trend;

however, there remain issues with the pH adjustments and this is a topic for future development in GLODAP, with the support from the OCB Ocean Carbonate System Intercomparison Forum (OCSIF, https://www.us-ocb.org/ocean-carbonate-system-intercomparison-forum/, last accessed: 20 June 2020) working group (Álvarez et al., 2020). For the nutrients and oxygen, only the phosphate adjustment frequency decreases from decade to decade. However, we do note that the more recent data from the 2010s receive the fewest adjustments. This may reflect recent increased attention that

seawater nutrient measurements have received through an operation manual (Becker et al., 2019; Hydes et al., 2012) availability of CRMNS (Aoyama et al., 2012; Ota et al., 2010), and the SCOR working group #147, Towards comparability of global oceanic nutrient data (COMPONUT). For silicate, the fraction of cruises receiving adjustments peaks in the 1990s and 2000s. This is related to the 2 % offset between US and Japanese cruises in the Pacific Ocean that was revealed during production of GLODAPv2 and discussed in Olsen et al. (2016). For salinity and the halogenated

transient tracers, the number of adjusted cruises is small in every decade.

**5 Data availability**

The GLODAPv2.2020 merged and adjusted data product is archived at NOAA NCEI under https://doi.org/10.25921/2c8h-sa89 (Olsen et al., 2020). These data and ancillary information are also available via our web pages https://www.glodap.info and https://www.nodc.noaa.gov/ocads/oceans/GLODAPv2_2020/ (last access: 22

June 2020). The data are available as comma-separated ascii files (*.csv) and as binary MATLAB files (*.mat) that use the open-source Hierarchical Data Format version 5 (HDF5) data format. Regional subsets are available for the Arctic, Atlantic, Pacific, and Indian oceans. There are no data overlaps between regional subsets and each cruise exists in only one basin file even if data from that cruise crosses basin boundaries. The station locations in each basin file are shown in Fig. 9. The product file variables are listed in Table 1. A lookup table for matching the EXPOCODE of a cruise with

GLODAP cruise number is provided with the data files. In the MATLAB files this information is available as a cell array. A "known issues document" accompanies the data files and provides an overview of known errors and omissions in the data product files. It is regularly updated, and users are encouraged to inform us whenever any new issues are identified. It is critical that users consult this document whenever the data products are used.

The original cruise files are available through the GLODAPv2.2020 cruise summary table (CST) hosted by NOAA

NCEI: https://www.nodc.noaa.gov/ocads/oceans/GLODAPv2_2020/ (Last access: 22 June 2020). Each of these files has been assigned a doi, but these are not listed here. The CST also provides brief information on each cruise and access to metadata, cruise reports, and its Adjustment Table entry.

While GLODAPv2.2020 is made available without any restrictions, users of the data should adhere to the fair data use principles:

For investigations that rely on a particular (set of) cruise(s), recognize the contribution of GLODAP data contributors by at least citing the articles where the data are described and, preferably, contacting principal investigators for exploring opportunities for collaboration and co-authorship. To this end, relevant articles and principal investigator names are provided in the cruise summary table. Contacting principal investigators comes with the additional benefit that the principal investigators often possess expert insight into the data and/or particular region under investigation. This can

improve scientific quality and promote data sharing.

This paper should be cited in any scientific publications that result from usage of the product. Citations provide the most efficient means to track use, which is important for attracting funding to enable the preparation of future updates.

**6 Summary**

GLODAPv2.2020 is an update of GLODAPv2.2019. Data from 106 new cruises have been added to supplement the earlier release and extend temporal coverage by 2 years. GLODAP now includes 47 years, 1972–2019, of global interior ocean biogeochemical data from 946 cruises.

The total number of data records are 1 275 558. Records with measurements for all 12 core variables, salinity, oxygen, nitrate, silicate, phosphate, $TCO_2$, TAlk, pH, CFC-11, CFC-12, CFC-113, and $CCl_4$ are very rare; only 2026 records have measured data for all 12 in the merged product file (interpolated and calculated data excluded). Requiring only two measured seawater $CO_2$ chemistry variables in addition to all the other core variables brings the number of available records up to 9 230, so this is also very rare. A major limiting factor is simultaneous availability of data for all four freon species, only 26 277 records have measurements of CFC-11, CFC-12, CFC-113, and $CCl_4$ while 400 587 have data for at least one of these (not considering availability of other core variables). A total of 398 757 records have measured data for two out of the three $CO_2$ chemistry core variables. The number of measured $fCO_2$ data are 33 924; note that these data were not subjected to quality control. The number of records with measured data for salinity, oxygen, and nutrients are 798 703, while the number of records with salinity and oxygen data are 1 077 859. All of these numbers are for measured data, not interpolated or calculated values.

Figure 10 illustrates the seasonal distribution of the data. As for previous versions there is a bias around summertime in the data in both hemispheres; most data are collected during April through November in the Northern Hemisphere while most data are collected during November through April in the Southern Hemisphere. These tendencies are strongest for the poleward regions and reflect the harsh conditions during winter months which make fieldwork difficult. Figure 11 illustrates the distribution of data with depth. The upper 100 m is the best sampled part of the global ocean, both in terms of number (Fig. 11a) and density (Fig. 11b) of observations. The number of observations steadily declines with depth. In part, this is caused by the reduction of ocean volume towards greater depths. Below 1000 m the density of observations stabilizes and even increases between 5000 and 6000 m; the latter is a zone where the volume of each depth surface decreases sharply (Weatherall et al., 2015). In the deep trenches, i.e., areas deeper than ~ 6000 m, both number and density of observations are low.

Except for salinity and oxygen, the core data were collected exclusively through chemical analyses of individually collected water samples. The data of the 12 core variables were subjected to primary quality control to identify questionable or bad data points (outliers) and secondary quality control to identify systematic measurement biases. The data are provided in two ways: as a set of individual exchange-formatted original cruise data files with assigned WOCE flags, and as globally and regionally merged data product files with adjustments applied to the data according to the outcome of the consistency analyses. Importantly, no adjustments were applied to data in the individual cruise files while primary-QC changes were applied.

The consistency analyses were conducted by comparing the data from the 106 new cruises to GLODAPv2.2019. Adjustments were only applied when the offsets were believed to reflect biases relative to the earlier data product release related to measurement calibration and/or data handling practices, and not to natural variability or anthropogenic trends. The Adjustment Table at https://glodapv2-2020.geomar.de/ (last access: 18 June 2020) lists all applied adjustments and provides a brief justification for each. The consistency analyses rely on deep ocean data (>1500 or 2000 dbar depending on region), but supplementary CANYON-B and CONTENT analyses consider data below 500 dbar. Data consistency for cruises with exclusively shallow sampling was not examined. No pH data were adjusted for this version, but we note that

this is largely a consequence of problems in establishing a reasonable pH baseline level in the deep northwest Pacific (Sect. 4.2). A comprehensive analysis of all available pH data in that region should be conducted for the next update.

Secondary QC flags are included for the 12 core variables in the product files. These flags indicate whether (1) or not (0) the data successfully received secondary QC. A secondary QC flag of 0 does not by itself imply that the data are of lower quality than those with a flag of 1. It means these data have not been as thoroughly checked. For $\delta^{13}C$, the QC results by Becker et al. (2016) for the North Atlantic were applied, and a secondary QC flag was therefore added to this variable.

The primary WOCE QC flags in the product files are simplified (e.g., all questionable and bad data were removed). For salinity, oxygen, and the nutrients, any data flagged 0 are interpolated rather than measured. For $TCO_2$, TAlk, pH, and $fCO_2$ any data flags of 0 indicate that the values were calculated from two other measured seawater $CO_2$ variables. Finally, while questionable (WOCE flag =3) and bad (WOCE flag =4) data have been excluded from the product files, some may have gone unnoticed through our analyses. Users are encouraged to report on any data that appear suspicious.

Based on the initial minimum adjustment limits and the improvement of the consistency resulting from the adjustments (Table 7), the data subjected to consistency analyses are believed to be consistent to better than 0.005 in salinity, 1 % in oxygen, 2 % in nitrate, 2 % in silicate, 2 % in phosphate, 4 $\mu mol\ kg^{-1}$ in $TCO_2$, 4 $\mu mol\ kg^{-1}$ in TAlk, and 5 % for the halogenated transient tracers. For pH, the consistency among all data is estimated as 0.01–0.02, depending on region. As mentioned above, the included $fCO_2$ data have not been subjected to quality control, therefore no uncertainty estimate is given for this variable. This should be conducted in future efforts.

## 7 Author contributions.

AO and TT led the team that produced this update. RMK, AK, and BP compiled the original data files. NL conducted the secondary QC analyses. HCB conducted the CANYON-B and CONTENT analyses. CS manages the Adjustment Table e-infrastructure. AK maintains the GLODAPv2 webpages at NCEI/OCADS while CSL maintains www.glodap.info. PM prepared PYTHON scripts for the merging of the data. All authors contributed to the interpretation of the secondary QC results and decisions on whether to apply actual adjustments. Many conducted ancillary QC analyses. AO wrote the manuscript with input from all authors.

## 8 Competing interests

The authors declare that they have no competing interests.

## 9 Acknowledgements

GLODAPv2.2020 would not have been possible without the effort of the many scientists who secured funding, dedicated time to collect, and shared the data that are included. Chief scientists at the various cruises and principal investigators for specific variables are listed in the online cruise summary table. NL was funded by EU Horizon 2020 through the EuroSea action (grant agreement 862626). LCC was supported by Prociencia/UERJ grant 2019-2021. MA was supported by IEO RADIALES and RADPROF projects. PJB was part-funded by the UK Climate Linked Atlantic Sector Science (CLASS) NERC National Capability Long-term Single Centre Science Programme (Grant NE/R015953/1). AV & FFP were supported by the BOCATS2 Project (PID2019-104279GB-C21) co-funded by the Spanish Government and the Fondo Europeo de Desarrollo Regional (FEDER). RW and BRC acknowledge the NOAA Global Observations and Monitoring

Division (fund reference 100007298) and the Office of Oceanic and Atnospheric Research of NOAA. HCB gratefully acknowledges financial support by the BONUS INTEGRAL project (Grant No. 03F0773A). We acknowledge funding from the Initiative and Networking Fund of the Helmholtz Association through the project "Digital Earth" [ZT-0025].

This is JISAO and PMEL contribution numbers 2020-1074 and 5112, respectively. This activity is supported by the IOCCP.

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

**Table 1.** Variables in the GLODAPv2.2020 comma separated (csv) product files, their units, short and flag names, and corresponding names in the individual cruise exchange files. In the MATLAB product files that are also supplied a "G2" has been added to every variable name.

| Variable | Units | Product file name | WOCE flag name[a] | 2nd QC flag name[b] | Exchange file name |
|---|---|---|---|---|---|
| Assigned sequential cruise number | | cruise | | | |
| Station | | station | | | STANBR |
| Cast | | cast | | | CASTNO |
| Year | | year | | | DATE |
| Month | | month | | | DATE |
| Day | | day | | | DATE |
| Hour | | hour | | | TIME |
| Minute | | minute | | | TIME |
| Latitude | | latitude | | | LATITUDE |
| Longitude | | longitude | | | LONGITUDE |
| Bottom depth | m | bottomdepth | | | |
| Pressure of the deepest sample | dbar | maxsampdepth | | | DEPTH |
| Niskin botttle number | | bottle | | | BTLNBR |
| Sampling pressure | dbar | pressure | | | CTDPRS |
| Sampling depth | m | depth | | | |
| Temperature | °C | temperature | | | CTDTMP |
| potential temperature | °C | theta | | | |
| Salinity | | salinity | salinityf | salinityqc | CTDSAL/SALNTY |
| Potential density anomaly | kg m$^{-3}$ | sigma0 | (salinityf) | | |
| Potential density anomaly, ref 1000 dbar | kg m$^{-3}$ | sigma1 | (salinityf) | | |
| Potential density anomaly, ref 2000 dbar | kg m$^{-3}$ | sigma2 | (salinityf) | | |
| Potential density anomaly, ref 3000 dbar | kg m$^{-3}$ | sigma3 | (salinityf) | | |
| Potential density anomaly, ref 4000 dbar | kg m$^{-3}$ | sigma4 | (salinityf) | | |
| Neutral density anomaly | kg m$^{-3}$ | gamma | (salinityf) | | |
| Oxygen | µmol kg$^{-1}$ | oxygen | oxygenf | oxygenqc | CTDOXY/OXYGEN |
| Apparent oxygen utilization | µmol kg$^{-1}$ | aou | aouf | | |
| Nitrate | µmol kg$^{-1}$ | nitrate | nitratef | nitrateqc | NITRAT |
| Nitrite | µmol kg$^{-1}$ | nitrite | nitritef | | NITRIT |
| Silicate | µmol kg$^{-1}$ | silicate | silicatef | silicateqc | SILCAT |
| Phosphate | µmol kg$^{-1}$ | phosphate | phosphatef | phosphateqc | PHSPHT |
| TCO$_2$ | µmol kg$^{-1}$ | tco2 | tco2f | tco2qc | TCARBON |
| TAlk | µmol kg$^{-1}$ | talk | talkf | talkqc | ALKALI |
| pH on total scale, 25° C and 0 dbar of pressure | | phts25p0 | phts25p0f | phtsqc | PH_TOT |

| Variable | Units | Product file name | WOCE flag name[a] | 2nd QC flag name[b] | Exchange file name |
|---|---|---|---|---|---|
| pH on total scale, in situ temperature and pressure | | phtsinsitutp | phtsinsitutpf | phtsqc | |
| $f$CO$_2$ at 20° C and 0 dbar of pressure | µatm | fco2 | fco2f | | FCO2/PCO2 |
| $f$CO$_2$ temperature[c] | °C | $f$co2temp | (fco2f) | | FCO2_TMP/PCO2_TMP |
| CFC-11 | pmol kg$^{-1}$ | cfc11 | cfc11f | cfc11qc | CFC-11 |
| pCFC-11 | ppt | pcfc11 | (cfc11f) | | |
| CFC-12 | pmol kg$^{-1}$ | cfc12 | cfc12f | cfc12qc | CFC-12 |
| pCFC-12 | ppt | pcfc12 | (cfc12f) | | |
| CFC-113 | pmol kg$^{-1}$ | cfc113 | cfc113f | cfc113qc | CFC-113 |
| pCFC-113 | ppt | pcfc113 | (cfc113f) | | |
| CCl$_4$ | pmol kg$^{-1}$ | ccl4 | ccl4f | ccl4qc | CCL4 |
| pCCl$_4$ | ppt | pccl4 | (ccl4f) | | |
| SF$_6$ | fmol kg$^{-1}$ | sf6 | sf6f | | SF6 |
| pSF6 | ppt | psf6 | (sf6f) | | |
| $\delta^{13}$C | ‰ | c13 | c13f | c13qc | DELC13 |
| $\Delta^{14}$C | ‰ | c14 | c14f | | DELC14 |
| $\Delta^{14}$C counting error | ‰ | c14err | | | C14ERR |
| $^3$H | TU | h3 | h3f | | TRITIUM |
| $^3$H counting error | TU | h3err | | | TRITER |
| $\delta^3$He | % | he3 | he3f | | DELHE3 |
| $^3$He counting error | % | he3err | | | DELHER |
| He | nmol kg$^{-1}$ | he | hef | | HELIUM |
| He counting error | nmol kg$^{-1}$ | heerr | | | HELIER |
| Ne | nmol kg$^{-1}$ | neon | neonf | | NEON |
| Ne counting error | nmol kg$^{-1}$ | neonerr | | | NEONER |
| $\delta^{18}$O | ‰ | o18 | o18f | | DELO18 |
| Total organic carbon | µmol L$^{-1\,d}$ | toc | tocf | | TOC |
| Dissolved organic carbon | µmol L$^{-1\,d}$ | doc | docf | | DOC |
| Dissolved organic nitrogen | µmol L$^{-1\,d}$ | don | donf | | DON |
| Dissolved total nitrogen | µmol L$^{-1\,d}$ | tdn | tdnf | | TDN |
| Chlorophyll $a$ | µg kg$^{-1\,d}$ | chla | chlaf | | CHLORA |

[a]The only derived variable assigned a separate WOCE flag is AOU as it depends strongly on both temperature and oxygen (and less strongly on salinity). For the other derived variables, the applicable WOCE flag is given in parenthesis. [b] Secondary QC flags indicate whether data have been subjected to full secondary QC (1) or not (0), as described in Sect. 3. [c] Included for clarity, is 20 °C for all occurences. [d]Units have not been checked; some values in micromoles per kilogram (for TOC, DOC, DON, TDN) or microgram per liter (for Chl $a$) are probable.


**Table 2.** WOCE flags in GLODAPv2.2020 exchange format original data files (briefly; for full details see Swift, 2010) and the simplified scheme used in the merged product files.

| WOCE Flag Value | Interpretation | |
| --- | --- | --- |
| | Original data exchange files | Merged product files |
| 0 | Flag not used | Interpolated or calculated value |
| 1 | Data not received | Flag not used[a] |
| 2 | Acceptable | Acceptable |
| 3 | Questionable | Flag not used[b] |
| 4 | Bad | Flag not used[b] |
| 5 | Value not reported | Flag not used[b] |
| 6 | Average of replicate | Flag not used[c] |
| 7 | Manual chromatographic peak measurement | Flag not used[c] |
| 8 | Irregular digital peak measurement | Flag not used[b] |
| 9 | Sample not drawn | No data |

[a]Flag set to 9 in product files

[b]Data are not included in the GLODAPv2.2020 product files and their flags set to 9.

[c]Data are included, but flag set to 2

**Table 3.** Initial minimum adjustment limits.

| Variable | Minimum Adjustment |
|---|---|
| Salinity | 0.005 |
| Oxygen | 1 % |
| Nutrients | 2 % |
| $TCO_2$ | 4 $\mu$mol kg$^{-1}$ |
| TAlk | 4 $\mu$mol kg$^{-1}$ |
| pH | 0.01 |
| CFCs | 5 % |

**Table 4.** Summary of salinity and oxygen calibration needs and actions; number of cruises with each of the scenarios identified.

| Case | Description | Salinity | Oxygen |
|------|-------------|----------|--------|
| 1 | No data are available: no action needed. | 0 | 8 |
| 2 | No bottle values are available: use CTD values. | 20 | 5 |
| 3 | No CTD values are available: use bottle values. | 0 | 67 |
| 4 | Too few data of both types are available for comparison and >80% of the records have bottle values: use bottle values. | 0 | 0 |
| 5 | The CTD values do not deviate significantly from bottle values: replace missing bottle values with CTD values. | 86 | 23 |
| 6 | The CTD values deviate significantly from bottle values: calibrate CTD values using linear fit and replace missing bottle values with calibrated CTD values. | 0 | 1 |
| 7 | The CTD values deviate significantly from bottle values, and no good linear fit can be obtained for the cruise: use bottle values and discard CTD values. | 0 | 2 |

**Table 5: Possible outcomes of the secondary QC and their codes in the online Adjustment Table**

| Secondary QC result | Code |
|---|---|
| The data are of good quality, consistent with the rest of the dataset and should not be adjusted. | 0/1[a] |
| The data are of good quality but are biased: adjust by adding (for salinity, $TCO_2$, TAlk, pH) or by multiplying (for oxygen, nutrients, CFCs) the adjustment value | Adjustment value |
| The data have not been QC'd, are of uncertain quality, and suspended until full secondary QC has been carried out | -666 |
| The data are of poor quality and excluded from the data product. | -777 |
| The data appear of good quality but their nature, being from shallow depths, coastal regions, without crossovers or similar, prohibits full secondary QC | -888 |
| No data exist for this variable for the cruise in question | -999 |

[a]The value of 0 is used for variables with additive adjustments (salinity, $TCO_2$, TAlk, pH) and 1 for variables with multiplicative adjustments (for oxygen, nutrients, CFCs). This is mathematically equivalent to 'no adjustment' in each case


**Table 6.** Summary of secondary QC results for the 106 new cruises, in number of cruises per result and per variable.

| | Sal. | Oxy. | NO$_3$ | Si | PO$_4$ | TCO$_2$ | TAlk | pH | CFC-11 | CFC-12 | CFC-113 | CCl$_4$ |
|---|---|---|---|---|---|---|---|---|---|---|---|---|
| With data | 106 | 101 | 97 | 97 | 97 | 92 | 96 | 82 | 16 | 21 | 3 | 0 |
| No data | 0 | 5 | 9 | 9 | 9 | 14 | 10 | 24 | 90 | 85 | 103 | 106 |
| Unadjusted[a] | 89 | 85 | 82 | 73 | 75 | 68 | 67 | 65 | 12 | 17 | 2 | 0 |
| Adjusted[b] | 0 | 1 | 1 | 9 | 7 | 2 | 6 | 0 | 1 | 2 | 0 | 0 |
| -888[c] | 17 | 14 | 14 | 14 | 14 | 22 | 23 | 12 | 2 | 2 | 1 | 0 |
| -666[d] | 0 | 0 | 0 | 0 | 0 | 0 | 0 | 5 | 0 | 0 | 0 | 0 |
| -777[e] | 0 | 1 | 0 | 1 | 1 | 0 | 0 | 0 | 1 | 0 | 0 | 0 |

[a]The data are included in the data product file as is, with a secondary QC flag of 1.

[b]The adjusted data are included in the data product file with a secondary QC flag of 1.

[c]Data appear of good quality but have not been subjected to full secondary QC. They are included in data product with a secondary QC flag of 0.

[d]Data are of uncertain quality and suspended until full secondary QC has been carried out; they are excluded from the data product.

[e]Data are of poor quality and excluded from the data product.



**Table 7.** Improvements resulting from quality control of the 106 new cruises, per basin and for the global data set. The numbers in the table are the weighted mean of the absolute offset of unadjusted and adjusted data versus GLODAPv2.2019. *n* is the total number of valid crossovers in the global ocean for the variable in question.

| | ARCTIC | | | ATLANTIC | | | INDIAN | | | PACIFIC | | | GLOBAL | | | *n* |
|---|---|---|---|---|---|---|---|---|---|---|---|---|---|---|---|---|
| | Unadj | | Adj | Unadj | | Adj | Unadj | | Adj | Unadj | | Adj | Unadj | | Adj | (global) |
| Sal ( x1000) | 1.7 | => | 1.7 | 5.6 | => | 5.6 | 4.0 | => | 4.0 | 1.9 | => | 1.9 | 2.4 | => | 2.4 | 2841 |
| Oxy (%) | 0.8 | => | 0.8 | 0.7 | => | 0.7 | 0.5 | => | 0.5 | 0.5 | => | 0.5 | 0.5 | => | 0.5 | 2462 |
| $NO_3$ (%) | 0.9 | => | 0.9 | 1.6 | => | 1.5 | 0.6 | => | 0.6 | 0.5 | => | 0.5 | 0.5 | => | 0.5 | 2158 |
| Si (%) | 3.6 | => | 3.6 | 2.5 | => | 2.4 | 1.9 | => | 1.1 | 1.0 | => | 0.8 | 1.0 | => | 0.8 | 1956 |
| $PO_4$ (%) | 5.0 | => | 2.6 | 2.2 | => | 2.0 | 0.8 | => | 0.8 | 0.8 | => | 0.7 | 0.8 | => | 0.8 | 2047 |
| $TCO_2$ (μmol/kg) | 3.4 | => | 3.4 | 2.6 | => | 2.6 | 1.9 | => | 1.9 | 2.1 | => | 1.8 | 2.2 | => | 1.9 | 512 |
| TAlk (μmol/kg) | 2.9 | => | 2.9 | 1.7 | => | 1.7 | 2.4 | => | 1.6 | 2.5 | => | 2.1 | 2.4 | => | 2.1 | 521 |
| pH ( x1000) | NA | => | NA | 8.5 | => | 8.5 | NA | => | NA | 8.3 | => | 7.4 | 8.3 | => | 7.5 | 458 |


**Appendix A.** Supplementary tables

**Table A1.** Cruises included in GLODAPv2.2020 that did not appear in GLODAPv2.2019. Complete information on each cruise, such as variables included, and chief scientist and principal investigator names is provided in the cruise summary table at https://www.nodc.noaa.gov/ocads/oceans/GLODAPv2_2020/cruise_table_v20202.html

| No | EXPOCODE | Region | Alias | Start | End | Ship |
|---|---|---|---|---|---|---|
| 2001 | 06M220120625 | Atlantic | MSM21/2 | 20120625 | 20120724 | *Maria S. Merian* |
| 2002 | 06M220130419 | Atlantic | MSM27 | 20130419 | 20130506 | *Maria S. Merian* |
| 2003 | 06M220130509 | Atlantic | MSM28 | 20130509 | 20130620 | *Maria S. Merian* |
| 2004 | 06M220140507 | Atlantic | MSM38 | 20140507 | 20140605 | *Maria S. Merian* |
| 2005 | 06M220150502 | Atlantic | MSM42 | 20150502 | 20150522 | *Maria S. Merian* |
| 2006 | 06M220150525 | Atlantic | MSM43 | 20150525 | 20150627 | *Maria S. Merian* |
| 2007 | 06M320100804 | Atlantic | M82/2 | 20100804 | 20100901 | *Meteor* |
| 2008 | 096U20180111 | Indian | SR03.2018 | 20180111 | 20180222 | *Investigator* |
| 2009 | 18HU20050904 | Atlantic | Davis Strait 2005 | 20050904 | 20050922 | *Hudson* |
| 2010 | 18SN20150920 | Arctic | JOIS2015 | 20150920 | 20151016 | *Louis S. St-Laurent* |
| 2011 | 29AH20160617 | Atlantic | OVIDE-16, A25, A01W | 20160617 | 20160731 | *Sarmiento de Gamboa* |
| 2012 | 29GD20120910 | Atlantic | EUROFLEETS | 20120910 | 20120915 | *Garcia del Cid* |
| 2013 | 29HE20190406 | Atlantic | FICARAM_XIX, A17 | 20190406 | 20190518 | *Hesperides* |
| 2014 | 316N20040922 | Atlantic | Davis Strait 2004, KN179-05 | 20040922 | 20041004 | *Knorr* |
| 2015 | 316N20061001 | Atlantic | Davis Strait 2006, KN187-02 | 20061001 | 20061004 | *Knorr* |
| 2016 | 316N20071003 | Atlantic | Davis Strait 2007, DKN192-02 | 20071003 | 20071021 | *Knorr* |
| 2017 | 316N20080901 | Atlantic | Davis Strait 2008, KN194-02 | 20080901 | 20080922 | *Knorr* |
| 2018 | 316N20091006 | Atlantic | Davis Strait 2009, KN196-02 | 20091006 | 20091028 | *Knorr* |
| 2019 | 316N20100804 | Atlantic | Davis Strait 2010 | 20100804 | 20100929 | *Knorr* |
| 2020 | 316N20101015 | Atlantic | KN199-04, GEOTRACES-2010 | 20101015 | 20101105 | *Knorr* |
| 2021 | 316N20111002 | Atlantic | Davis Strait 2011, KN203-04 | 20111002 | 20111021 | *Knorr* |
| 2022 | 316N20130914 | Atlantic | Davis Strait 2013, KN213-02 | 20130914 | 20131003 | *Knorr* |
| 2023 | 316N20150906 | Atlantic | Davis Strait 2015 | 20150906 | 20150924 | *Knorr* |
| 2024 | 32WC20110812 | Pacific | WCOA2011 | 20110812 | 20110830 | *Wecoma* |
| 2025 | 33RO20160505 | Pacific | WCOA2016 | 20160505 | 20160606 | *Ronald H. Brown* |
| 2026 | 35TH20080825 | Atlantic | SUBPOLAR08 | 20080825 | 20080915 | *Thalassa* |
| 2027 | 45CE20170427 | Atlantic | CE17007, A02 | 20170427 | 20170522 | *Celtic Explorer* |
| 2028 | 49UF20101002 | Pacific | ks201007 | 20101002 | 20101104 | *Keifu Maru II* |
| 2029 | 49UF20101109 | Pacific | ks201008 | 20101109 | 20101126 | *Keifu Maru II* |
| 2030 | 49UF20101203 | Pacific | ks201009 | 20101203 | 20101222 | *Keifu Maru II* |
| 2031 | 49UF20111004 | Pacific | ks201109 | 20111004 | 20111127 | *Keifu Maru II* |
| 2032 | 49UF20111205 | Pacific | ks201110 | 20111205 | 20111221 | *Keifu Maru II* |
| 2033 | 49UF20120410 | Pacific | ks201203 | 20120410 | 20120424 | *Keifu Maru II* |
| 2034 | 49UF20120602 | Pacific | ks201205 | 20120602 | 20120614 | *Keifu Maru II* |
| 2035 | 49UF20131006 | Pacific | ks201307 | 20131006 | 20131022 | *Keifu Maru II* |
| 2036 | 49UF20131029 | Pacific | ks201308 | 20131029 | 20131210 | *Keifu Maru II* |
| 2037 | 49UF20140107 | Pacific | ks201401 | 20140107 | 20140125 | *Keifu Maru II* |
| 2038 | 49UF20140206 | Pacific | ks201402 | 20140206 | 20140326 | *Keifu Maru II* |
| 2039 | 49UF20140410 | Pacific | ks201403 | 20140410 | 20140505 | *Keifu Maru II* |
| 2040 | 49UF20140512 | Pacific | ks201404 | 20140512 | 20140617 | *Keifu Maru II* |
| 2041 | 49UF20140623 | Pacific | ks201405, P09, P13 | 20140623 | 20140826 | *Keifu Maru II* |
| 2042 | 49UF20140904 | Pacific | ks201406 | 20140904 | 20141019 | *Keifu Maru II* |

| 2043 | 49UF20150107 | Pacific | ks201501 | 20150107 | 20150126 | *Keifu Maru II* |
|------|--------------|---------|----------|----------|----------|-----------------|
| 2044 | 49UF20150202 | Pacific | ks201502 | 20150202 | 20150306 | *Keifu Maru II* |
| 2045 | 49UF20150415 | Pacific | ks201504 | 20150415 | 20150504 | *Keifu Maru II* |
| 2046 | 49UF20150511 | Pacific | ks201505 | 20150511 | 20150611 | *Keifu Maru II* |
| 2047 | 49UF20150620 | Pacific | ks201506, P09, P13 | 20150620 | 20150823 | *Keifu Maru II* |
| 2048 | 49UF20151021 | Pacific | ks201508 | 20151021 | 20151202 | *Keifu Maru II* |
| 2049 | 49UF20160107 | Pacific | ks201601 | 20160107 | 20160126 | *Keifu Maru II* |
| 2050 | 49UF20160201 | Pacific | ks201602 | 20160201 | 20160310 | *Keifu Maru II* |
| 2051 | 49UF20160407 | Pacific | ks201604 | 20160407 | 20160507 | *Keifu Maru II* |
| 2052 | 49UF20160512 | Pacific | ks201605 | 20160512 | 20160610 | *Keifu Maru II* |
| 2053 | 49UF20160618 | Pacific | ks201606 | 20160618 | 20160723 | *Keifu Maru II* |
| 2054 | 49UF20160730 | Pacific | ks201607 | 20160730 | 20160912 | *Keifu Maru II* |
| 2055 | 49UF20160917 | Pacific | ks201608 | 20160917 | 20161007 | *Keifu Maru II* |
| 2056 | 49UF20161116 | Pacific | ks201609 | 20161116 | 20161219 | *Keifu Maru II* |
| 2057 | 49UF20170110 | Pacific | ks201701, P09, P10 | 20170110 | 20170223 | *Keifu Maru II* |
| 2058 | 49UF20170228 | Pacific | ks201702 | 20170228 | 20170326 | *Keifu Maru II* |
| 2059 | 49UF20170408 | Pacific | ks201703 | 20170408 | 20170426 | *Keifu Maru II* |
| 2060 | 49UF20170502 | Pacific | ks201704 | 20170502 | 20170606 | *Keifu Maru II* |
| 2061 | 49UF20170612 | Pacific | ks201705 | 20170612 | 20170713 | *Keifu Maru II* |
| 2062 | 49UF20170719 | Pacific | ks201706, P09, P10 | 20170719 | 20170907 | *Keifu Maru II* |
| 2063 | 49UF20171107 | Pacific | ks201708 | 20171107 | 20171208 | *Keifu Maru II* |
| 2064 | 49UF20180129 | Pacific | ks201802 | 20180129 | 20180309 | *Keifu Maru II* |
| 2065 | 49UF20180406 | Pacific | ks201804 | 20180406 | 20180512 | *Keifu Maru II* |
| 2066 | 49UF20180518 | Pacific | ks201805 | 20180518 | 20180703 | *Keifu Maru II* |
| 2067 | 49UF20180709 | Pacific | ks201806 | 20180709 | 20180829 | *Keifu Maru II* |
| 2068 | 49UF20180927 | Pacific | ks201808 | 20180927 | 20181021 | *Keifu Maru II* |
| 2069 | 49UP20110912 | Pacific | rf201109 | 20110912 | 20110929 | *Ryofu Maru III* |
| 2070 | 49UP20120306 | Pacific | rf201202 | 20120306 | 20120325 | *Ryofu Maru III* |
| 2071 | 49UP20121116 | Pacific | rf201208 | 20121116 | 20121218 | *Ryofu Maru III* |
| 2072 | 49UP20130307 | Pacific | rf201302 | 20130307 | 20130327 | *Ryofu Maru III* |
| 2073 | 49UP20130426 | Pacific | rf201304 | 20130426 | 20130527 | *Ryofu Maru III* |
| 2074 | 49UP20131128 | Pacific | rf201310 | 20131128 | 20131223 | *Ryofu Maru III* |
| 2075 | 49UP20140108 | Pacific | rf201401, P09, P10 | 20140108 | 20140301 | *Ryofu Maru III* |
| 2076 | 49UP20140307 | Pacific | rf201402 | 20140307 | 20140326 | *Ryofu Maru III* |
| 2077 | 49UP20140429 | Pacific | rf201404 | 20140429 | 20140530 | *Ryofu Maru III* |
| 2078 | 49UP20140609 | Pacific | rf201405 | 20140609 | 20140629 | *Ryofu Maru III* |
| 2079 | 49UP20141112 | Pacific | rf201409 | 20141112 | 20141202 | *Ryofu Maru III* |
| 2080 | 49UP20150110 | Pacific | rf201501 | 20150110 | 20150223 | *Ryofu Maru III* |
| 2081 | 49UP20150228 | Pacific | rf201502 | 20150228 | 20150326 | *Ryofu Maru III* |
| 2082 | 49UP20150408 | Pacific | rf201503 | 20150408 | 20150419 | *Ryofu Maru III* |
| 2083 | 49UP20150426 | Pacific | rf201504 | 20150426 | 20150528 | *Ryofu Maru III* |
| 2084 | 49UP20150604 | Pacific | rf201505 | 20150604 | 20150623 | *Ryofu Maru III* |
| 2085 | 49UP20150627 | Pacific | rf201506 | 20150627 | 20150716 | *Ryofu Maru III* |
| 2086 | 49UP20151115 | Pacific | rf201509 | 20151115 | 20151216 | *Ryofu Maru III* |
| 2087 | 49UP20160109 | Pacific | rf201601, P09, P10 | 20160109 | 20160222 | *Ryofu Maru III* |
| 2088 | 49UP20160227 | Pacific | rf201602 | 20160227 | 20160324 | *Ryofu Maru III* |
| 2089 | 49UP20160408 | Pacific | rf201603 | 20160408 | 20160421 | *Ryofu Maru III* |
| 2090 | 49UP20160427 | Pacific | rf201604 | 20160427 | 20160601 | *Ryofu Maru III* |

| 2091 | 49UP20160608 | Pacific | rf201605 | 20160608 | 20160628 | *Ryofu Maru III* |
|------|--------------|---------|----------|----------|----------|------------------|
| 2092 | 49UP20161021 | Pacific | rf201608 | 20161021 | 20161206 | *Ryofu Maru III* |
| 2093 | 49UP20170107 | Pacific | rf201701 | 20170107 | 20170126 | *Ryofu Maru III* |
| 2094 | 49UP20170201 | Pacific | rf201702 | 20170201 | 20170310 | *Ryofu Maru III* |
| 2095 | 49UP20170425 | Pacific | rf201705 | 20170425 | 20170508 | *Ryofu Maru III* |
| 2096 | 49UP20170623 | Pacific | rf201707 | 20170623 | 20170827 | *Ryofu Maru III* |
| 2097 | 49UP20170815 | Pacific | rf201708 | 20170815 | 20171006 | *Ryofu Maru III* |
| 2098 | 49UP20171125 | Pacific | rf201710 | 20171125 | 20171224 | *Ryofu Maru III* |
| 2099 | 49UP20180110 | Pacific | rf201801 | 20180110 | 20180222 | *Ryofu Maru III* |
| 2100 | 49UP20180228 | Pacific | rf201802 | 20180228 | 20180326 | *Ryofu Maru III* |
| 2101 | 49UP20180501 | Pacific | rf201804 | 20180501 | 20180605 | *Ryofu Maru III* |
| 2102 | 49UP20180614 | Pacific | rf201805 | 20180614 | 20180722 | *Ryofu Maru III* |
| 2103 | 49UP20180806 | Pacific | rf201806, P13 | 20180806 | 20180927 | *Ryofu Maru III* |
| 2104 | 64PE20071026 | Atlantic | PE278 | 20071026 | 20071117 | *Pelagia* |
| 2105 | 740H20180228 | Atlantic | JC159 | 20180228 | 20180410 | *James Cook* |
| 2106 | 91AA20171209 | Indian | NCAOR, SOE2017-18 | 20171209 | 20180204 | *S.A. Agulhas I* |


**Figure Captions**

**Figure 1.** Location of stations in (a) GLODAPv2.2019 and for (b) the new data added in this update.

**Figure 2.** Number of cruises per year in GLODAPv2, GLODAPv2.2019, and GLODAPv2.2020.

**Figure 3.** Example crossover figure, for $TCO_2$ for cruises 49UP20160109 (blue) and 49UP20160703 (red), as it was generated during the crossover analysis. Panel **(a)** show all station positions for the two cruises and **(b)** show the specific stations used for the crossover analysis. Panel **(d)** shows the data of $TCO_2$ ($\mu$mol kg$^{-1}$) below the upper depth limit (in this case 2000 dbar) versus potential density anomaly referenced to 4000 dbar, as points and the interpolated profiles as lines. Non-interpolated data either did not meet minimum depth separation requirements (Table 4 in Key et al., 2010) or are the deepest sampling depth. The interpolation does not extrapolate.
Panel **(e)** shows the mean $TCO_2$ ($\mu$mol kg$^{-1}$) difference profile (black, dots) with its standard deviation, and also the weighted mean offset (straight, red) and weighted standard deviation. Summary statistics are provided in **(c)**.

**Figure 4.** Example summary figure, for $TCO_2$ crossovers for 49UP20160109 versus the cruises in GLODAPv2.2019 (with cruise EXPOCODE listed on x-axis sorted according to year the cruise was conducted). The black dots and vertical error bars show the weighted mean offset and standard deviation for each crossover (in $\mu$mol kg$^{-1}$). The weighted mean and standard deviation of all these
offsets are shown in the red lines and are $3.68 \pm 0.83$ $\mu$mol kg$^{-1}$. The black dashed line is the reference line for a +4 $\mu$mol kg$^{-1}$ offset (the corresponding line for – 4 $\mu$mol kg$^{-1}$ offset is right on top of x-axis and not visible).

**Figure 5.** Example summary figure for CANYON-B and CONTENT analyses for 49UP20160109. Any data from regions where CONTENT and CANYON-B were not trained are excluded (in this case, the Sea of Japan). The top row shows the nutrients and the bottom row the seawater $CO_2$ chemistry variables (Note, different abbreviations for $TCO_2$ (CT) and TAlk (AT)). All are shown versus
sampling pressure (dbar) and the unit is $\mu$mol kg$^{-1}$ for all except pH, which is unitless. Black dots (which to a large extent are hidden by the predicted estimates) are the measured data, blue dots are CANYON-B estimates and red dots are the CONTENT estimates. Each variable has two figure panels. The left shows the depth profile while the right shows the absolute difference between measured and estimated values divided by the CANYON-B/CONTENT uncertainty estimate, which is determined for each estimated value. These values are used to gauge the comparability; a value below 1 indicates a good match as it means that the difference between measured
and estimated values is less than the uncertainty of the latter. The statistics in each panel are for all data deeper than 500 dbar and N is the number of samples; considered. A gain ratio and its interquartile range is given for the nutrients. For the seawater $CO_2$ chemistry variables the numbers on each panel are the median difference between measured and predicted values for CANYON-B (upper) and CONTENT (lower). Both are given with their interquartile range.

**Figure 6.** Distribution of applied adjustments for each core variable that received secondary QC, in $\mu$mol kg$^{-1}$ for $TCO_2$ and TAlk,
unitless for salinity and pH (but multiplied with 1000 in both cases so a common x-axis can be used), while for the other properties adjustments are given in percent ((adjustment ratio-1)x100)).  Grey areas depict the initial minimum adjustment limits. The figure includes numbers for data subjected to secondary quality control only. Note also that the y-axis scale is set to render the number of adjustments to be visible, so the bar showing zero offset (the 0 bar) for each variable is cut off (see Table 6 for these numbers).

**Figure 7**. Distribution of pH offsets for the cruises from Japan Meteorological Agency added in GLODAPv2.2020.

**Figure 8.** Magnitude of applied adjustments relative to minimum adjustment limits (Table 3) per decade for the 946 cruises included in GLODAPv2.2020.

**Figure 9.** Locations of stations included in the (a) Arctic, (b) Atlantic, (c) Indian, and (d) Pacific Ocean product files for the complete GLODAPv2.2020 dataset.

**Figure 10.** Distribution of data in GLODAPv2.2020 in (a) December–February, (b) March–May, (c) June–August, (d) September–
November, and (e) number of observations for each month in four latitude bands..

**Figure 11.** Number (a) and density (b) of observations in 100 m depth layers. The latter was calculated by dividing the number of observations in each layer by its global volume calculated from ETOPO2 (National Geophysical Data Center, 2006). For example, in the layer between 0 and 100 m there are on average 0.0075 observations per cubic kilometer. One observation is one water sampling point and has data for several variables.


(a)

(b)

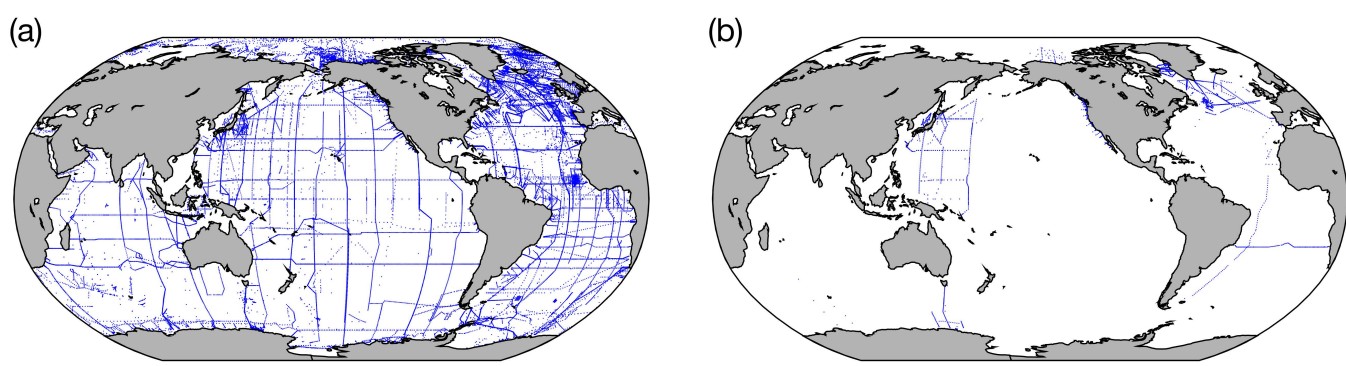

Figure 2

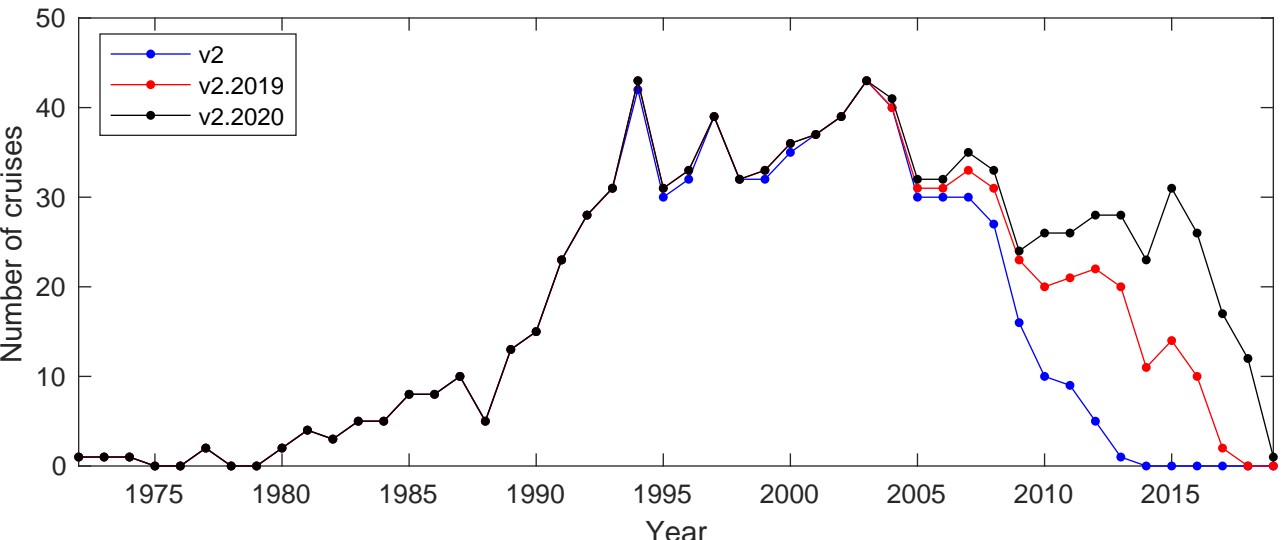

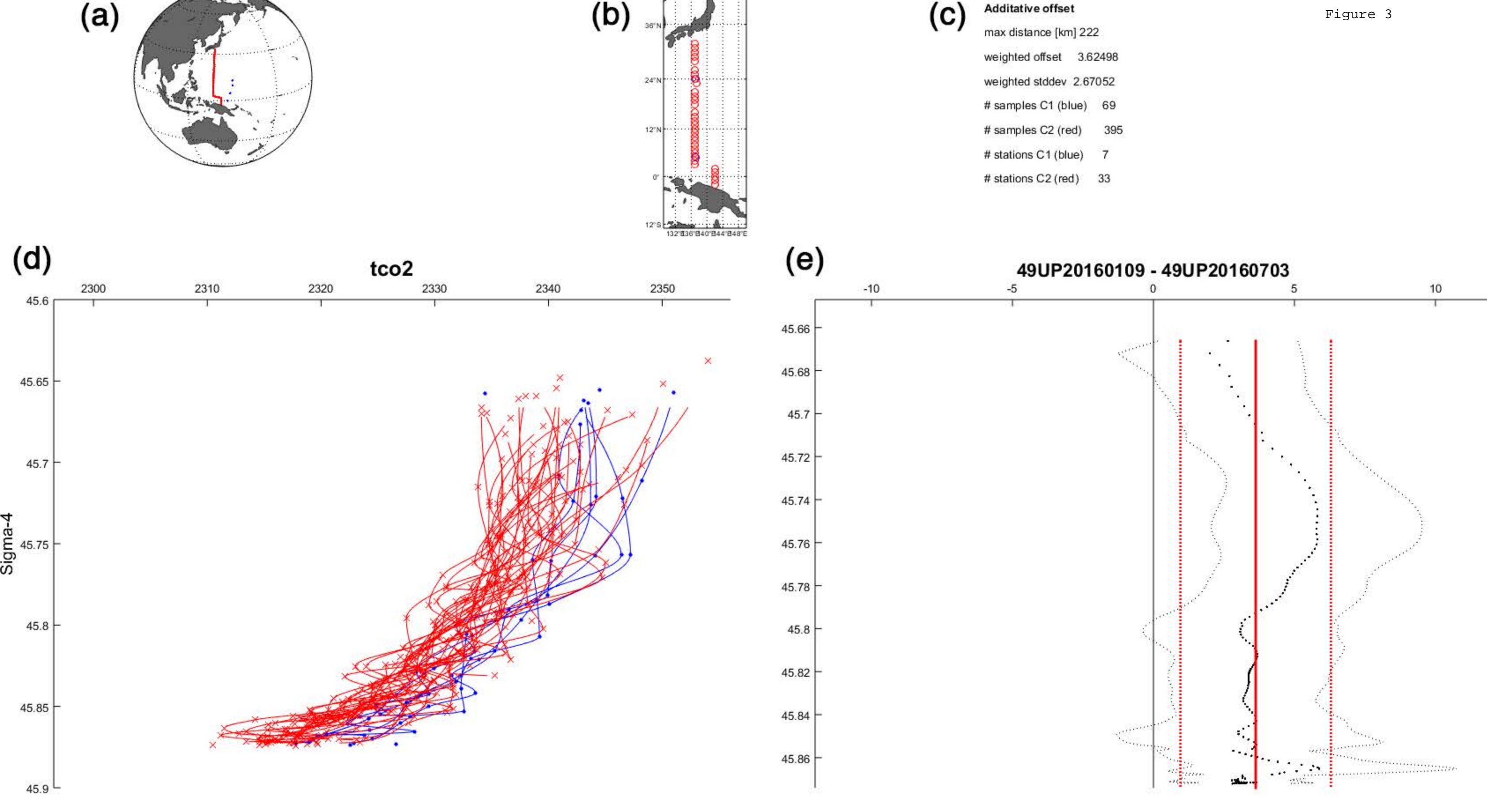

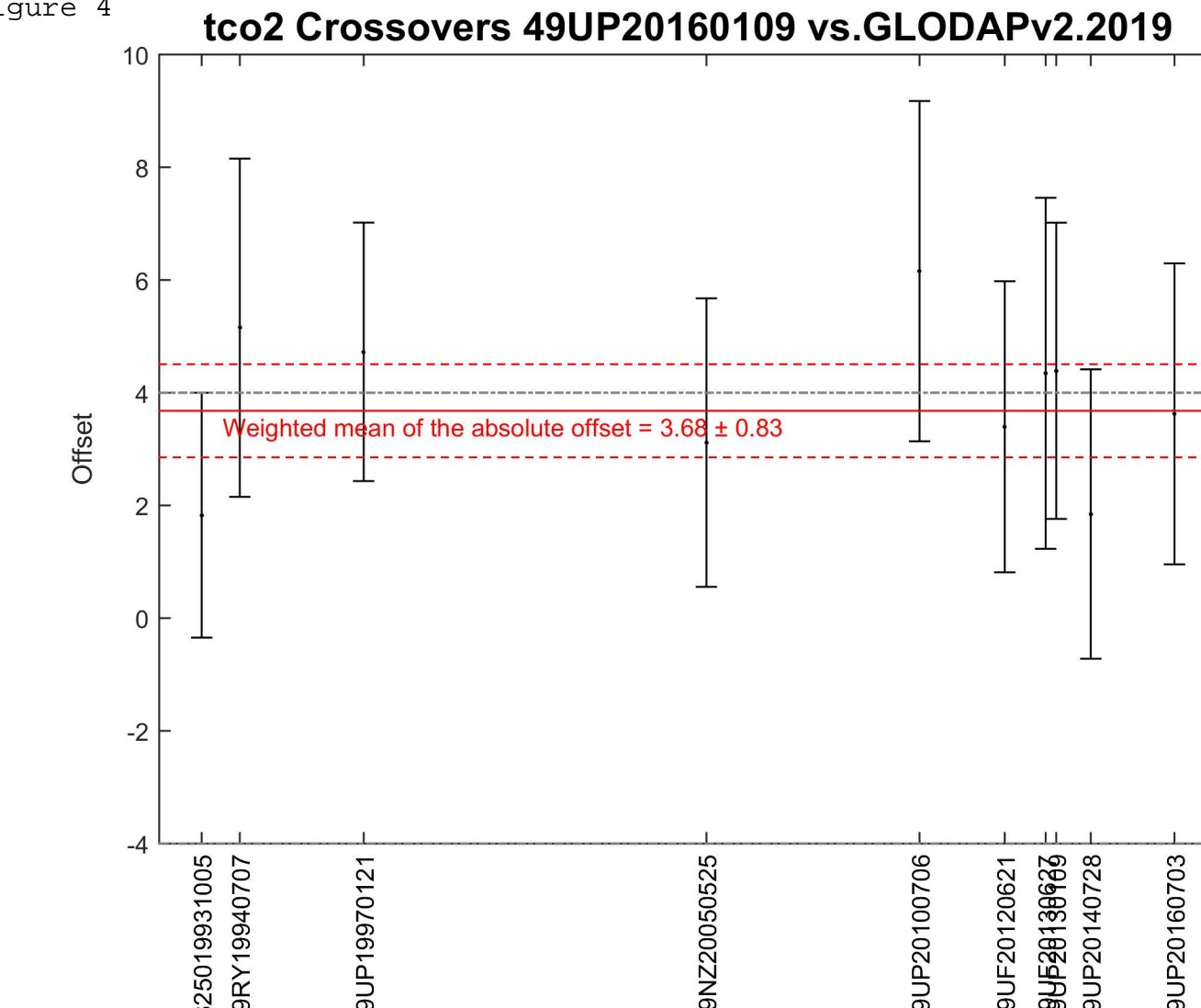

**tco2 Crossovers 49UP20160109 vs.GLODAPv2.2019**

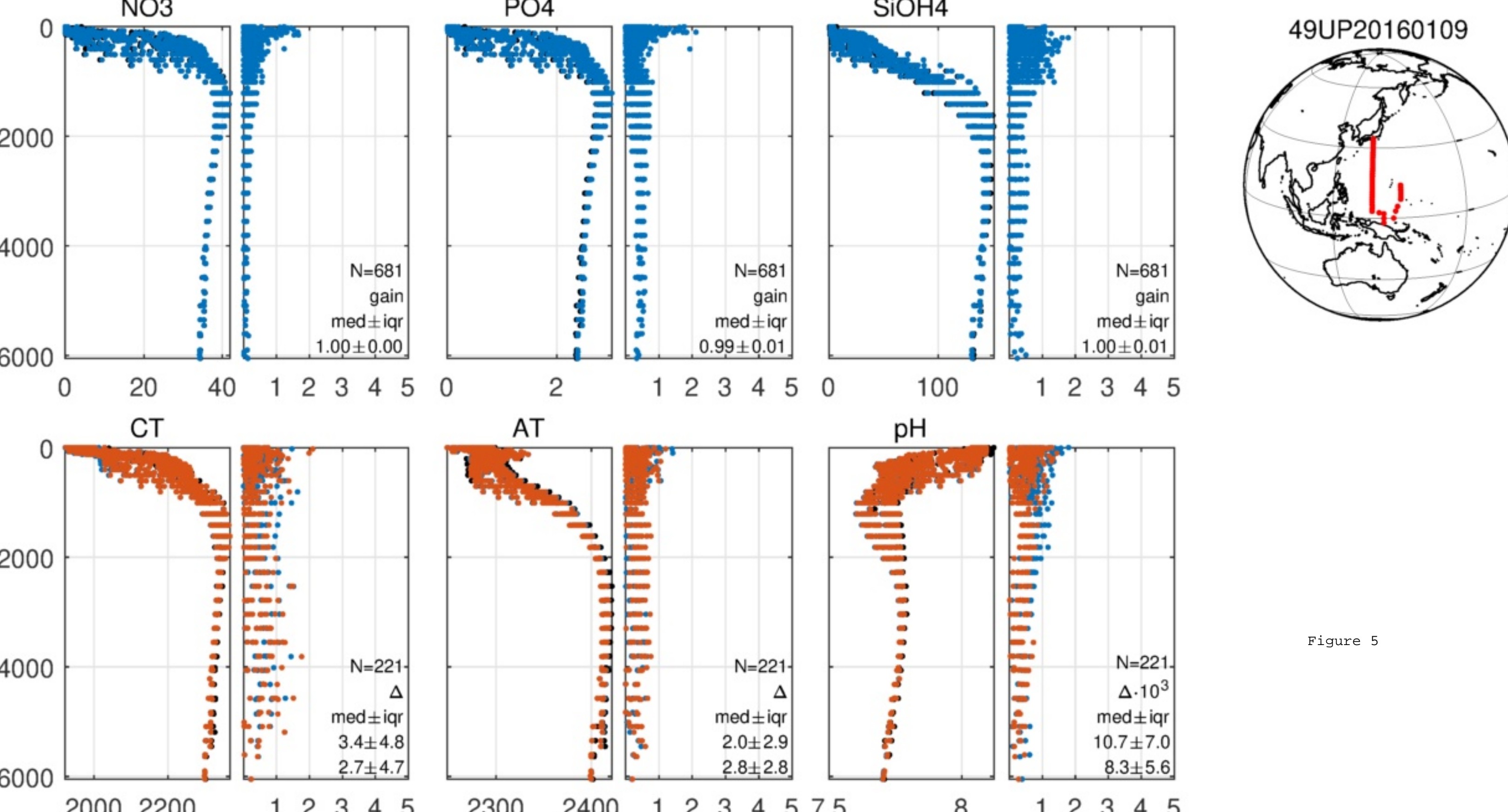

Figure 5

Figure 6

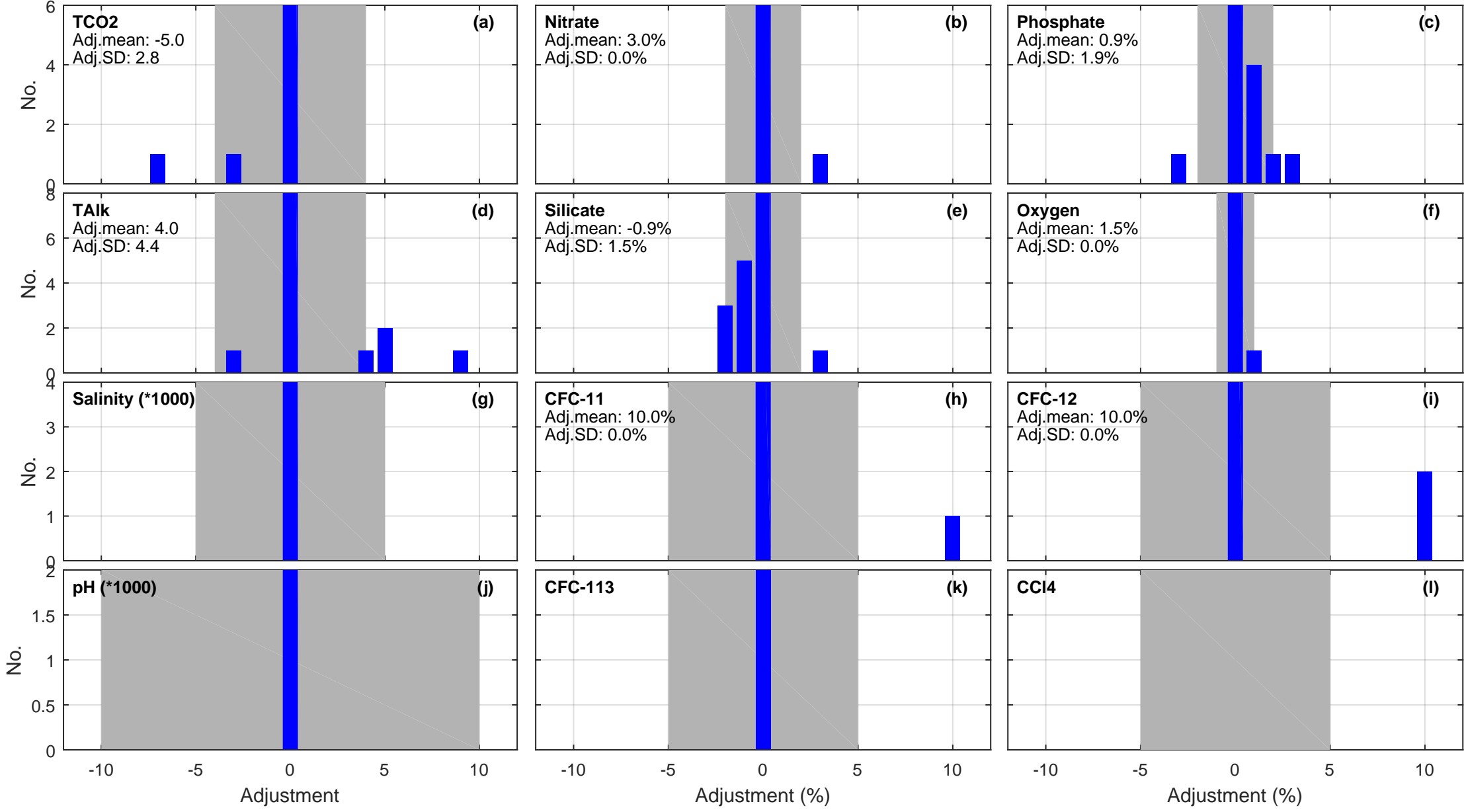

Figure 7

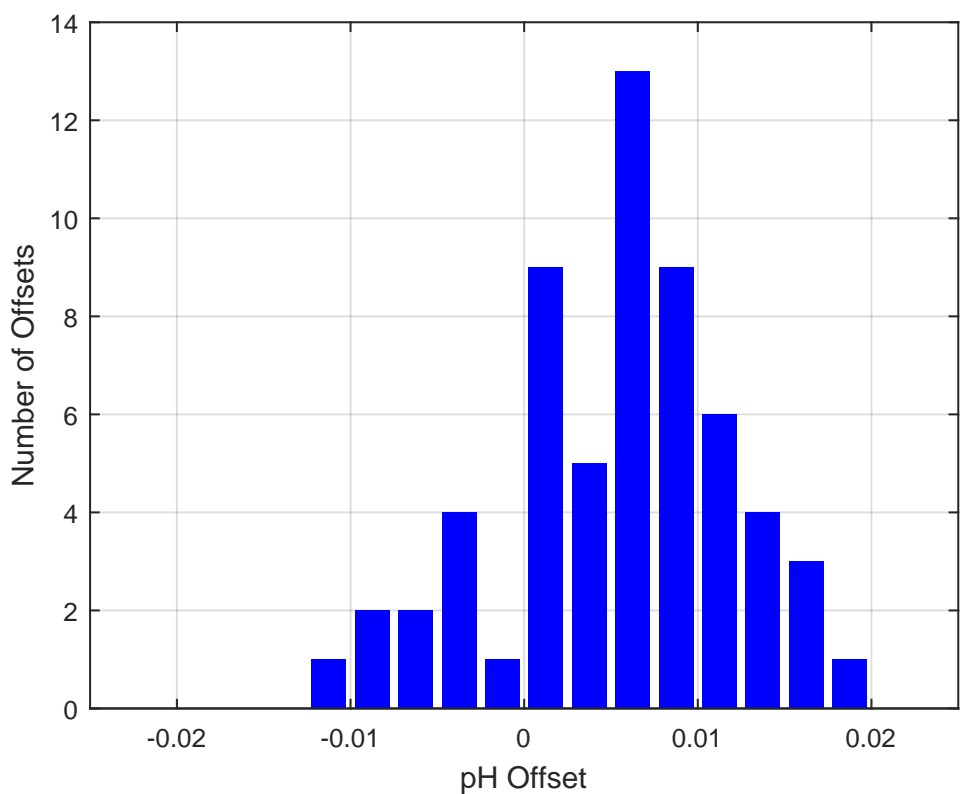

Figure 8

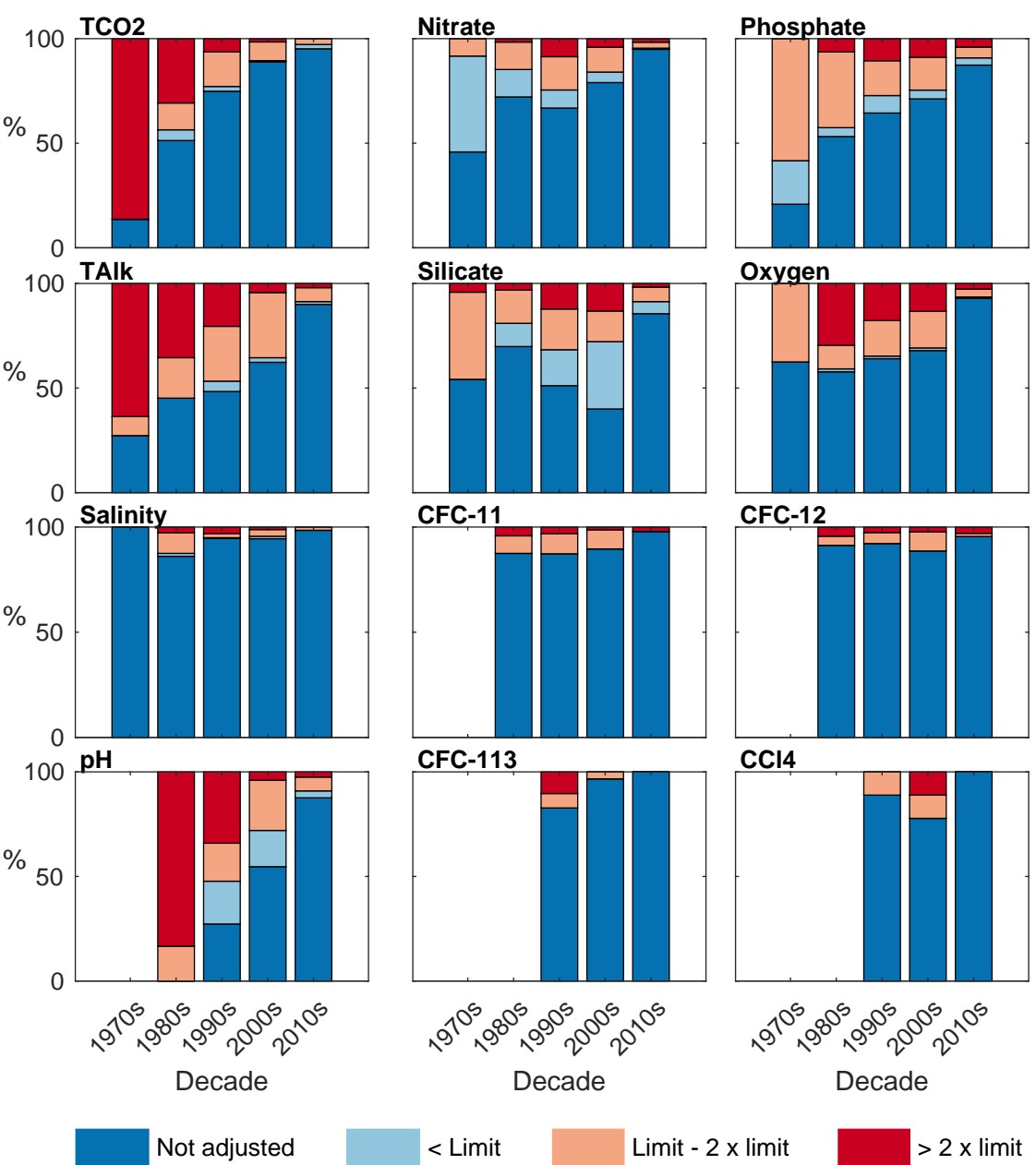

Figure 9

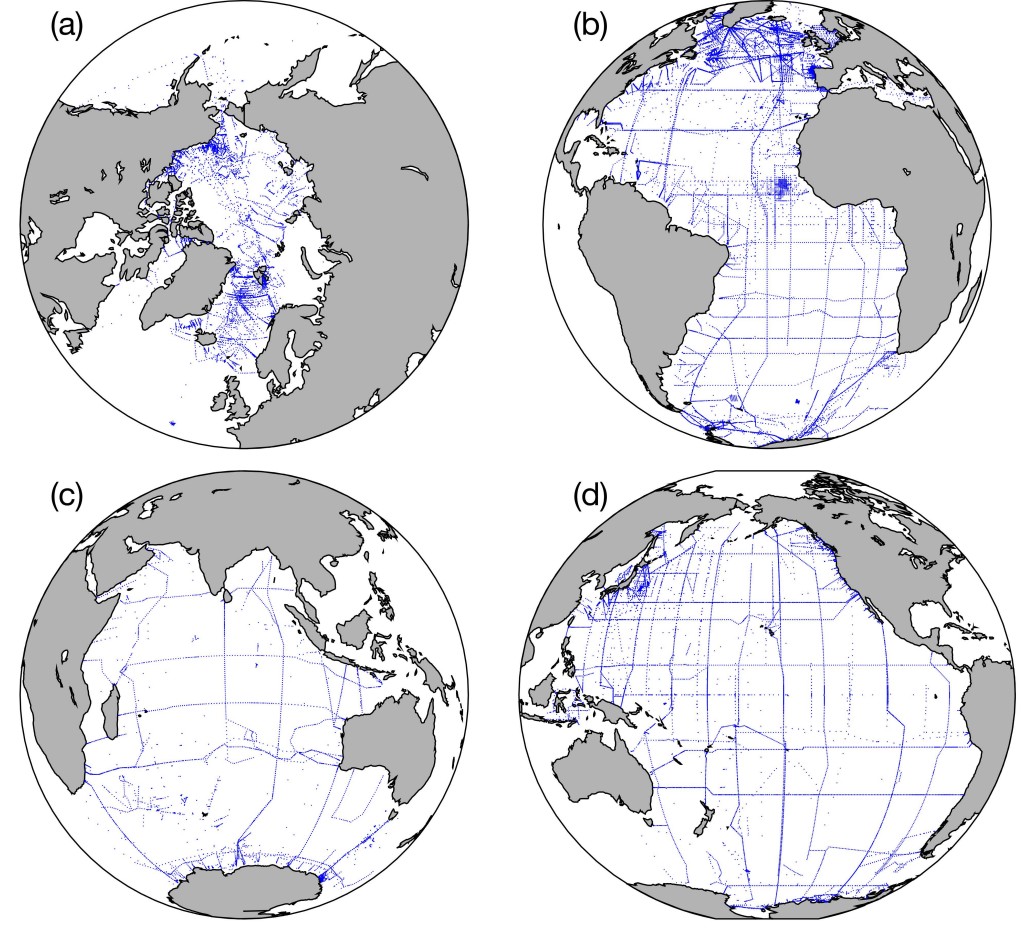

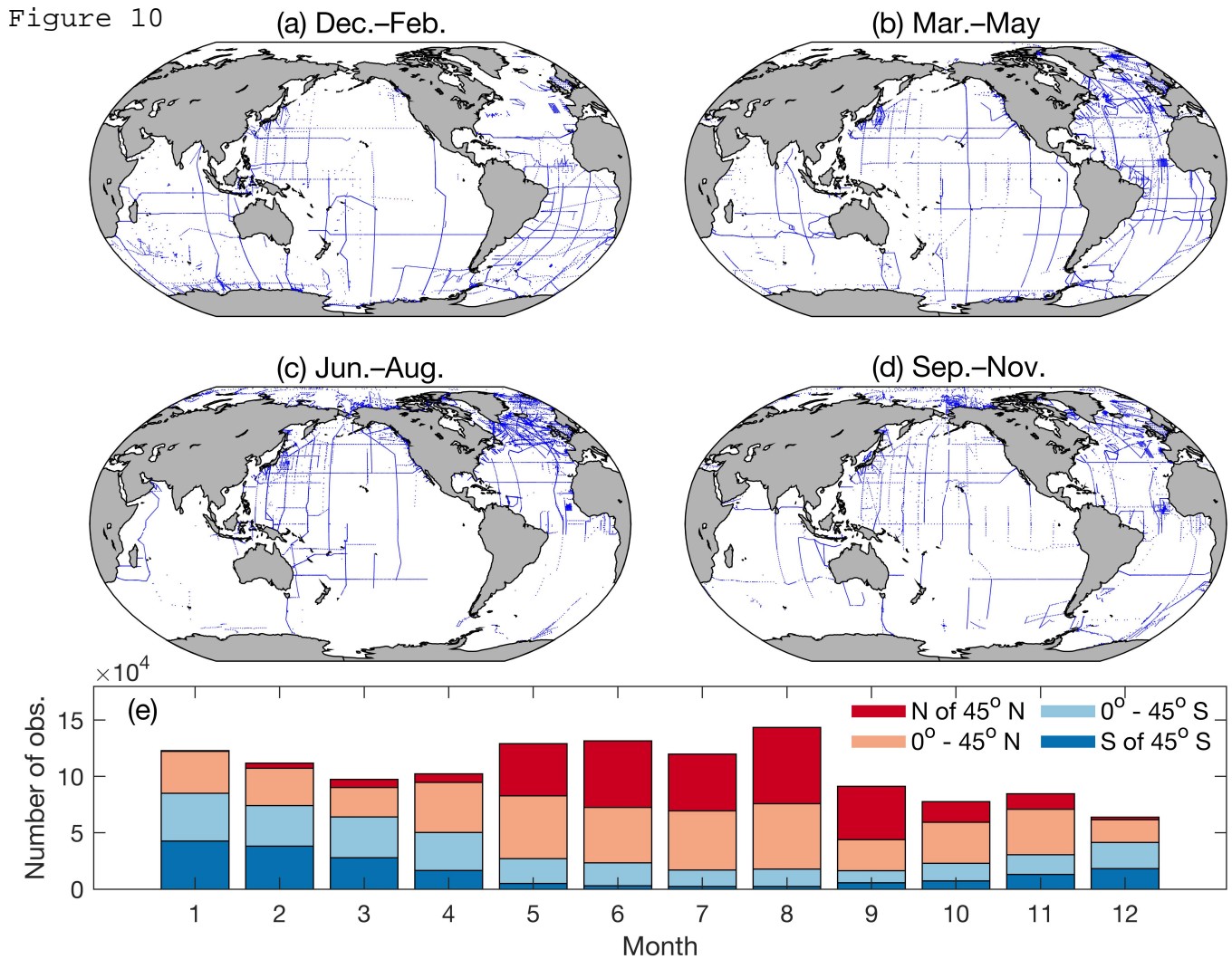

Figure 10
(a) Dec.–Feb.
(b) Mar.–May
(c) Jun.–Aug.
(d) Sep.–Nov.

Figure 11

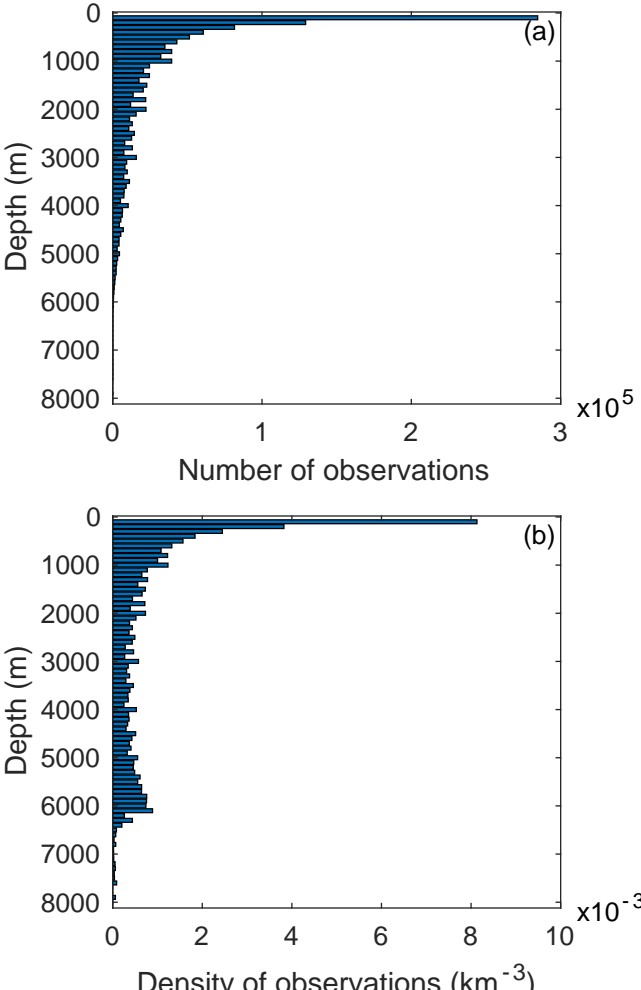