# Peer review of "An updated version of the global interior ocean biogeochemical data product, GLODAPv2.2020"

_Earth System Science Data, 2020_

## Referee Comment (RC1) · Matthew Humphreys (Referee) · 6 Aug 2020

The new cruise datasets added to GLODAP in this release constitute a substantial update to this already invaluable data product. The manuscript is clearly written and virtually ready to publish as it is. The first section of my comments below raises a few minor issues that the authors should consider before publication of this paper. The second section contains broader suggestions that might benefit future releases, but which are not necessary to include in this version.

[Figure]

**1 Comments for this manuscript**

**1.1 Version naming convention**

The new version number/naming convention outlined in lines 146–147 is intuitive and clear to follow. It could be more strongly emphasised here that the exact version number used should always be reported in studies, rather than making a generic reference to GLODAP.

It might be helpful to also explicitly commit to what may and may not be changed between different levels of version release. For example, in the "minor" version increments new cruises may be added but data that was already there will not change (with the exception of bug fixes, such as described in section 3.3.1), whereas it sounds like a "major" version increment would involve a reanalysis of the entire dataset, in which the adjustments applied to existing datasets could be more fundamentally altered.

Even if it's not exactly as I've described, some sort of explicit commitment like this could be helpful — users who switch to a newer version could immediately know what they can rely on to be consistent, and what changes they need to watch out for — and now, as the new versioning system is introduced, seems like a good opportunity to do this.

**1.2 Carbonate ion measurements**

The "four variables" statement in line 360 ignores the increasing reliability of carbonate ion measurements (e.g. Sharp and Byrne, 2019). I suggest to modify this statement accordingly; it is not really necessary to specify "four" or any specific number here at all.

**1.3 pH adjustments — or not**

It would be useful to recap that pH adjustments were not applied to the new data in this version where this is mentioned in the summary on lines 554–555.

**1.4 Figures**

The axis labels and other text notes on a couple of the figures are a bit too small relative to the figure size, making reading difficult (e.g. Figure 3).

Although you can work these out from context — if you are familiar with the field — several of the figures are missing axis labels and units for the variables shown (e.g. Figures 3 through 6).

**1.5 Typos**

Abstract: add a comma after "discrete $f\mathrm{CO}_2$" on line 56. Change "bias corrected product" to "bias-corrected product" on line 60.

I suggest to change "are released regularly" to "will be released regularly" on line 145.

Summary: the sentence on lines 554–555 is missing a full stop at the end.

**2 Suggestions for future releases**

The following points are not revisions that are necessary for this publication, but rather ideas that could be taken under consideration for future releases of GLODAP.

**2.1 Expand dataset sourcing**

The latest GEOTRACES Intermediate Data Product (Schlitzer et al., 2018) contains some datasets with the core GLODAP variables that are not included in this GLODAP release. While it's unreasonable to expect the GLODAP team to continually seek out new data from an endless list of sources, it may be worth including the GEOTRACES IDPs for future versions given the typically high quality of the carbonate system data therein, abundance of auxiliary variables to aid secondary QC, and consistent data format.

**2.2 Accept carbonate ion measurements**

As noted above, carbonate ion measurements are now becoming usefully reliable (e.g. Sharp and Byrne, 2019) and becoming more widespread. Accepting this type of data into GLODAP would be a natural extension to the current set of core variables, adding a new dimension to some applications of the GLODAP database such as evaluating dissociation constants based on over-determined data points (e.g. Sulpis et al., 2020).

**2.3 Update carbonate system calculations**

The analysis here still uses CO2SYS for MATLAB v1 (van Heuven et al., 2011). Updating to at least CO2SYSv2 (Orr et al., 2018) would enable uncertainty propagation — given that some calculated marine carbonate system variables are reported, it would be useful to also propagate uncertainties from the measured variables and dissociation constants into the calculated variables.

Updating further still to the recently released CO2SYSv3 (Sharp et al., 2020) would also enable calculations with carbonate ion as an input variable, if these measurements were to be accepted in future GLODAP releases. Ammonia and sulfide speciation

are also included in the alkalinity equation as of CO2SYSv3, which could improve the accuracy of marine carbonate system calculations in areas where these species are significantly abundant.

**3 References**

van Heuven, S., Pierrot, D., Rae, J. W. B., Lewis, E. and Wallace, D. W. R.: CO2SYS v 1.1, MATLAB program developed for $CO_2$ system calculations, ORNL/CDIAC-105b, Carbon Dioxide Information Analysis Center, Oak Ridge National Laboratory, U.S. Department of Energy, Oak Ridge, TN, USA, doi:10.3334/CDIAC/otg.CO2SYS$_MATLAB_v1.1, 2011$.

Orr, J. C., Epitalon, J.-M., Dickson, A. G. and Gattuso, J.-P.: Routine uncertainty propagation for the marine carbon dioxide system, Mar. Chem., 207, 84–107, doi:10.1016/j.marchem.2018.10.006, 2018.

Schlitzer, R., Anderson, R. F., Dodas, E. M., Lohan, M., Geibert, W., Tagliabue, A., Bowie, A., Jeandel, C., Maldonado, M. T., Landing, W. M., Cockwell, D., Abadie, C., Abouchami, W., Achterberg, E. P., Agather, A., Aguliar-Islas, A., van Aken, H. M., Andersen, M., Archer, C., Auro, M., de Baar, H. J., Baars, O., Baker, A. R., Bakker, K., Basak, C., Baskaran, M., Bates, N. R., Bauch, D., van Beek, P., Behrens, M. K., Black, E., Bluhm, K., Bopp, L., Bouman, H., Bowman, K., Bown, J., Boyd, P., Boye, M., Boyle, E. A., Branellec, P., Bridgestock, L., Brissebrat, G., Browning, T., Bruland, K. W., Brumsack, H.-J., Brzezinski, M., Buck, C. S., Buck, K. N., Buesseler, K., Bull, A., Butler, E., Cai, P., Mor, P. C., Cardinal, D., Carlson, C., Carrasco, G., Casacuberta, N., Casciotti, K. L., Castrillejo, M., Chamizo, E., Chance, R., Charette, M. A., Chaves, J. E., Cheng, H., Chever, F., Christl, M., Church, T. M., Closset, I., Colman, A., Conway, T. M., Cossa, D., Croot, P., Cullen, J. T., Cutter, G. A., Daniels, C., Dehairs, F., Deng, F., Dieu, H. T., Duggan, B., Dulaquais, G., Dumousseaud, C., Echegoyen-Sanz, Y., Edwards, R.

L., Ellwood, M., Fahrbach, E., Fitzsimmons, J. N., Russell Flegal, A., Fleisher, M. Q., van de Flierdt, T., Frank, M., Friedrich, J., Fripiat, F., Fröllje, H., Galer, S. J. G., Gamo, T., Ganeshram, R. S., Garcia-Orellana, J., Garcia-Solsona, E., Gault-Ringold, M., et al.: The GEOTRACES Intermediate Data Product 2017, Chem. Geol., 493, 210–223, doi:10.1016/j.chemgeo.2018.05.040, 2018.

Sharp, J. D. and Byrne, R. H.: Carbonate ion concentrations in seawater: Spectrophotometric determination at ambient temperatures and evaluation of propagated calculation uncertainties, Mar. Chem., 209, 70–80, doi:10.1016/j.marchem.2018.12.001, 2019.

Sharp, J. D., Pierrot, D., Humphreys, M. P., Epitalon, J.-M., Orr, J. C., Lewis, E. R. and Wallace, D. W. R.: CO2SYSv3 for MATLAB, Zenodo, doi:10.5281/zenodo.3950562, 2020.

Sulpis, O., Lauvset, S. K. and Hagens, M.: Current estimates of $K_1^*$ and $K_2^*$ appear inconsistent with measured $CO_2$ system parameters in cold oceanic regions, Ocean Science, 16(4), 847–862, doi:10.5194/os-16-847-2020, 2020.

---

## Referee Comment (RC2) · Nicolas Metzl (Referee) · 21 Aug 2020

General comments:

Since 15 years GLODAP data-bases (from 2004 to 2019, including CARINA, PACIFICA) are widely used in the community, not only to evaluate the change of CO2 in the ocean or acidification (e.g. Gruber et al 2019; Jiang et al 2019), but also to compare and validate ocean and climate models (e.g. CMIP5, Bronselaer and Zanna, 2020 for a recent publication). The GLODAP data-set is also an important synthesis for GOA-ON activities and to construct climatology (e.g. Broullón et al, 2020).

Here, authors present an updated version of the GLODAP effort. This includes 106 new cruises quality controlled (QC), inclusion of new fCO2 observations (not QCed) and comparison of secondary QC with reconstructed properties using neural network methods (named CANYON-B and CONTENT).

The effort consists mainly in (i) format and check the data received from PI or available in different locations (NCEI/OCADS, PANGAEA, CCHDO), (ii) performed a secondary QC to identify data biases (if any) and separate from real temporal changes of the properties that could be low relative to the mean concentrations and (ii) construct final formatted products with adjusted data and associated flags for easy use at global or regional scales.

The paper is basically structured from the previous manuscript (Olsen et al 2019) and I therefore have only few comments regarding this new version (v2020). Most suggestions are for clarity, here thinking to readers that would discover only now the GLODAP project (e.g. new students in the field).

As fCO2 data are now included, GLODAP is in a way a companion data-base to SOCAT dedicated to surface fCO2 data (Bakker et al 2016) also annually updated (Bakker et al 2020). Both products were already used together for specific analysis (e.g. comparing pH fields from GLODAP and SOCAT, Jiang et al 2019). It might be useful for future to attempt incorporate fCO2 data that are in GLODAP but not yet in SOCAT. In this context few words might be added at the end in the conclusions/perspectives.

In this version, authors used CANYON-B and CONTENT methods (I think this was not systematically performed in v2019). This is a new and an elegant way to check and compare secondary control (and bias if any). This is a new step in GLODAP that might be recalled in the abstract for this version.

Something not very clear concerns the QC for historical cruises. With the new cruises in hand, I was not sure at the start if the QC of previous cruises in the same regions has been checked again and would lead to new corrections for cruises already in GLODAP-v1, CARINA or v2019. However, as specify in the manuscript (line 145) I understand that a complete revision of QC would be performed in 2023 (after 3d GO-SHIP).

Also, many colleagues used the GLODAP gridded products that were constructed from GLODAP-v2 (Lauvset et al 2016). Will you also revisiting this gridded product now or

wait for the 2023 version ? This might be specified in the manuscript.

Another remark concerns the new cruises to be added in GLODAP. I understand that new cruises (106) were recently obtained from NCEI or PANGAEA or from PIs. However, I suspect there are many other cruises in the community (published) and it would be useful to find the best way to get more cruises in the future and invite new PIs to contribute.

Overall, I recommend publication after few minor revisions.

Below I list specific and minor comments (mostly details for clarity for a reader who discover Glodap for the first time). At the end of the review few technical questions regarding the files on-line.

......................
,,,,,,,,,,,,,,,,,,,,,,,

Specific comments:

C-01: Title: The title includes only acronyms of the project (GLODAP). Would it be useful to recall that this concerns Ocean biogeochemical observations in the water column? A Suggestion for a title: "An updated version of global interior ocean biogeo-chemical observations, GLODAPv2-2020".

C-02: Page 2, line 44: "the inclusion of available discrete fugacity of CO2 (fCO2) values in the merged product files". Does this new inclusion concerns only the new cruises added in v2020 or did you also add this parameter for historical cruises ? (this is specify later, Line 369).

C-03: Page 4, Line 121: "The data collected across the Davis Strait". Maybe specify where is the Davis Strait for those not familiar with the Indian Ocean... (...Atlantic of course)

C-04: Page 5, Line 175: For new users: Not sure to clearly understand all Flag definitions listed in Table 2.

C-05: Table 2: for clarity, it might be useful to assign different flag for interpolated and calculated values (both flag 0). Maybe for the next version.

C-06: In table 2, you list "b" "Data are not included in the GLODAPv2.2020 product files and their flags set to 9. " Does that mean that original flag 3 (Questionable but sometimes maybe real signal) are not included in the files ? However this is explained later, line 395...

C-07: In table 2, you list "c" for replicate: "Data are included, but flag set to 2 ". This suggests that all replicate are acceptable (or some were also identify as outliers and thus moved to flag 9 or deleted ?).

C-08: Page 6, Line 197: "comparison of deep-water averages". Specify the layers here ? How this is selected in the high latitude (e.g. bottom water formations, where anthropogenic CO2 is found to be relatively high in water column ?).

C-09: Page 6, Line 200: Add reference to CANYON-B and CONTENT (first time listed here) ?

C-10: Page 6, Line 226: "In areas where a strong trend in salinity was present". Any example for this version ?

C-11: Page 7, Line 235: "convection occurs (such as the Nordic, Labrador, and Irminger seas)". How do you select the layer in region of bottom water formation (e.g. SR03 for this version) ? Might be interesting for new readers to show another QC example (as presented in Figure 3 for the North Pacific).

C-12: Page 7, Line 238: Maybe recall that 49UP20160109 is new while 49UP20160703 was QCed in v2019.

C-13: The example in Figure 3 shows in 3a blue dots on the map, but I suspect these stations (far east) were not used to evaluate the QC.

C-14: Page 7, Lines 245-250: For 49UP20160109, maybe specify that no temporal

changes was observed for salinity (i.e. you used TCO2 here, not normalized TCO2 as suggested in Line 227 for some cruises).

C-15: Page 7, Line 245: Figure 4 shows the TCO2 cross-over for 49UP20160109 versus GLODAPv2-v2019. The cruise 49UP20160703 is also plotted and thus was in GLODAPv2-v2019, although conducted after 49UP20160109 (just to clarify for a new user).

C-16: Page 7, Line 256: "they are included in the product but with a secondary QC flag of 0 (Sect.6)". Sect 6 (?)

C-17: Page 7, Line 259: "A few new cruises had no or very few valid crossovers with GLODAPv2 data." Which cruises ? Would it be relevant to add a column in Table-Annexe 1 with a remark specifying what kind of secondary QC has been performed for each cruise (e.g. Standard QC, MLR, no QC) ?

C-18: Page 8, Section 3.2.3: I understand the description but what are the results and which cruise ? Would be interesting to show an example for a cruise that is QCed using MLR.

C-19: Page 8, Line 277: "Altogether 82 of the 106 new cruises included pH data." Here specify this is measured pH, not calculated (so there is no confusion with pH calculated for other cruises).

C-20: Page 8, Line 291: "The pH data of 840 of the 936 cruises in GLODAPv2.2020". Again, specify if pH data here were measured or calculated or both.

C-21: Page 8, Line 305: Maybe recall the mean uncertainty associated to CANYON-B and CONTENT (see table 1 in Bittig et al 2018, i.e. about twice the adjustment limits fixed for GLODAP listed in Table 3).

C-22: Page 8, Line 305: As it is new results presented here (and probably also used in the next version), I think some more information is needed. For CANYON-B and CON-TENT are you using results based on GLODAP-v2 data (Bittig et al 2018) or an updated

version using GLODAPv2-2019. Is the comparison presented here (Figure 5) validate the QC for the new cruises or validate CANYON-B and CONTENT reconstructed fields ? It is reassuring to get about the same results as CANYON-B and CONTENT were trained with GLODAP.

C-23: Page 8, Line 308: Figure 5: not easy to see the black dots (measured values).

C-24: Figure 5: there is no units (to be added in captions ?).

C-25: Figure 5: Like for Figure 3 and 4, it would be nice to show another example, e.g. SR3 or Davis Strait ? Or an example where the comparison between QC from GLODAP and CANYON-B/CONTENT does not work (if any). This is a suggestion not absolutely needed.

C-26: Page 9, line 320: "Another advantage of CANYON-B and CONTENT is that by considering the each data point in it self, primary QC issues has been revealed and corrected for some of the cruises." Which cruises ? Give some examples ?

C-27: Page 9-10: Section 3.3.1. Lines 332-358: This is a list of revisions and would be better to move this section in an Annex but keep in Section 3.3.1 the fCO2 information (lines 359-375) as it is new data added in v2020.

C-28: Page 10: Concerning fCO2, in the GLODAP files there are now both fCO2 measured and calculated in the same column. Authors indicate that all values were converted to 20°C. However, in the data-files, there are fCO2 values with fCO2temp fixed at -9999. I missed something here and not sure if all fCO2 values in the files are at the same temperature, pressure or at local temperature etc... Also, there are fCO2 values with flag 0 or 2. What was the criteria for fCO2 with flag 2 ? How users can easily separate the fCO2 measured and calculated in the files ? This is important to clarify if one uses both GLODAP (in surface) and SOCAT to merge both products.

C-29: Page 10, line 364: "These calculated TAlk values were, however, not included in v2.2019." Does that mean that all TALK values with flag 0 in the files are only

interpolated values (i.e. not calculated as an option suggested in table 2).

C-30; Page 11, Lines 397-398: For flags 6 and 7 now set to flag 2, recall that this only applied for valid data (i.e. obvious outliers deleted also for these replicates ?).

C-31: Page 11, Line 399: "Missing sampling pressures or depths were calculated following UNESCO (1981)." This is obvious but maybe rewrite following: "Missing sampling pressures (resp. depths) were calculated from depths (reps. pressures) following UNESCO (1981)."

C-32: Page 11-12, Lines 405 and 432: Flag 0 is used for both interpolated and calculated values. Why not using different flag ? (for next version)

C-33: Page 11, Line 416. Concerning the "Missing seawater CO2 chemistry variables". Are the calculated properties used only measured data (i.e. TALK and TCO2) or also interpolated values ? In other words, are the fCO2 and pH interpolated values based on calculated fCO2 and pH or recalculated with interpolated TALK/TCO2 ?

C-34: Page 13, Line 486: "For example, Arctic Ocean phosphate, Indian Ocean silicate and TCO2, and Pacific Ocean pH data all show considerable improvements." For Indian, in Table 6 improvement is for TALK, not TCO2 ?

C-35: Page 15, Line 544: Weatherall et al., (2015): not in references.

C-36: Now concerning the files, for curiosity I had a look at the Indian.cvs file and have few questions that could be also valid for other basin. The questions below are obvious for someone familiar with Glodap, but mainly addressed here to help new users.

C-36a: Why the QC flags for S or O2 are 0 for several cruises although flag WOCE are 2 ? Is it because the secondary QC is not available for these cruises ?

C-36c: There are data with WOCE flag=0 for O2, Nitrate, Silicates, Phosphates, TCO2, TALK, pH, and associated to QC flag = 1. Is it because these are interpolated values for a cruise/station for which a secondary QC was performed ? If QC has been performed

(QCF=1) one would expect a WOCE flag different from 0 ? I thought the QC is based on original data (not interpolated or calculated). Could that be clarified ?

C-36d: There are data with flag 9 associated to QC flag=1. Again, is it because QC flag (0,1) are assigned for a cruise/station not for each data?

C-37: In the data files on-line (e.g. GLODAPv2.2020_Indian_Ocean.cvs) I would suggest to add units for each column.

C-38: And for next versions, I think for clarity a different flag should be assign for calculated (e.g. fCO2, pH) and interpolated values. This might help some users to select only measured+interpolated values.

In references:

I think each reference should now have a DOI

Line 663: "Hood, E. M., Sabine, C. L., and Sloyan, B. M.: The GO-SHIP hydrography manual: A collection of expert reports and guidelines, 2010." Specify the publisher ? DOI ?

References in this review not listed in the Manuscript

Bakker, D. C. E., et al., 2016. A multi-decade record of high-quality fCO2 data in version 3 of the Surface Ocean CO2 Atlas (SOCAT), Earth Syst. Sci. Data, 8, 383-413, doi:10.5194/essd-8-383-2016.

Bakker, Dorothee C. E.; et al., (2020). Surface Ocean CO2 Atlas Database Version 2020 (SOCATv2020) (NCEI Accession 0210711). [indicate subset used]. NOAA National Centers for Environmental Information. Dataset. https://doi.org/10.25921/4xkx-ss49. Accessed [date].

Bronselaer, B., Zanna, L. Heat and carbon coupling reveals ocean warming due to circulation changes. Nature 584, 227–233 (2020). Doi:10.1038/s41586-020-2573-5

Broullón, D., Pérez, F. F., Velo, A., Hoppema, M., Olsen, A., Takahashi, T., Key, R. M., Tanhua, T., Santana-Casiano, J. M., and Kozyr, A.: A global monthly climatology of oceanic total dissolved inorganic carbon: a neural network approach, Earth Syst. Sci. Data, 12, 1725–1743, https://doi.org/10.5194/essd-12-1725-2020, 2020.

Jiang, L.-Q., Carter, B. R., Feely, R. A., Lauvset, S. K. and Olsen, A. Surface ocean pH and buffer capacity: past, present and future. Sci Rep 9, 18624 (2019) doi:10.1038/s41598-019-55039-4

Lauvset, S. K, R. M. Key, A. Olsen, S. van Heuven, A. Velo, X. Lin, C. Schirnick, A. Kozyr, T. Tanhua, M. Hoppema, S. Jutterström, R. Steinfeldt, E. Jeansson, M. Ishii, F. F. Pérez, T. Suzuki & S. Watelet, 2016. A new global interior ocean mapped climatology: the 1°x1° GLODAP version 2. Earth Syst. Sci. Data, 8, 325-340, doi:10.5194/essd-8-325-2016.

;;;;;;;;;;; end review

---

## Referee Comment (RC3) · Anonymous Referee #3 · 28 Aug 2020

This is a "living data" update document that discussed the addition of 106 cruises to the GLODAPv2.2019 data set. These data have been extremely valuable to the community and represent an important asset to maintain and update. The manuscript is well written and informative. I only have a few minor comments below.

Line 92-93: The authors don't distinguish between discrete and in situ sensor measurements here. I assume they are referring to CTD calibration problems with respect to the sensor measurements of salinity and oxygen, not the measurements of collected samples. Please clarify, particularly in light of the merging discussed in section 3.2.1.

Lines 95-99: The manuscript uses some rather subjective terms without defining their meaning in this context. For example, "poor precision can render a set of data unusable" or "to minimize severe cases of bias". What is the definition of poor precision or severe bias?

Lines 98, 108: There are a notable number of grammatical errors in the text that should be fixed. A couple of examples are, "Adjustments are applied on the data"(should be 'to the data') or "A particular important source" (should be 'A particularly important source'). Please review the entire document for these grammatical errors.

Line 123-124: The authors decided to include cruises on the Merian, Meteor, and the Garcia del Cid that did not have any nutrient or carbon data. I thought nutrients and carbon were the primary parameters for this data set. Why did the authors decide to include these data and not the thousands of other cruises that also do not have carbon data. This seem inconsistent with the goal of this project.

Line 150: define data center acronyms the first time they are used, or at least provide links to the data centers.

Line 193-195: We the original data generators consulted before adjustments were made to the data? I believe in the past there was a step that involved checking with the people that originally made the measurement to get their perspective on possible offsets.

Line 256: This is the first time that a -888 label is discussed in the text. What does this mean? The same comes in later with -777 and -666 labels.

Lines 280-282: Why did the authors use the full GLODAPv2 data to estimate TAlk from Salinity. Wouldn't it make more sense to calculate an average ratio for the data from that cruise rather than use a global ratio that includes data from other oceans? Also, doesn't the ratio change with depth?

———————————

---

## Referee Comment (RC4) · Nancy Williams (Referee) · 3 Sep 2020

General Comments:

This is an update to the GLODAPv2.2019 by adding 106 new cruises from 2004-2019, expanding the coverage of GLODAP to 946 cruises over 47 years, 1972–2019. Most of the new cruises are from the western North Pacific and the Davis Strait, with a few from the Atlantic, South Indian, and U.S. West coast. The methods for primary and secondary quality control (QC) are essentially the same as in the earlier version. However, there has been no full consistency analysis of the entire data product as was done with the original GLODAPv2 product. A full consistency analysis will be performed in the future for the next GLODAP update (will be termed "GLODAPv3") which is set to occur after the completion of the third GO-SHIP survey around year 2023. The researchers have also fixed some minor errors in the GLODAPv2.2019 dataset.

[Figure]

Throughout the manuscripts the researchers discuss alternate ways of adjusting the dataset, and tend to take a conservative approach, saving any major changes for the next full GLODAP update, i.e., GLODAPv3. As such, this update could be considered by some to be incremental, but it should be noted that incremental and timely updates to GLODAP are critical to advancing ocean observing. GLODAP, and other such data products that have come before it, forms the backbone for studying large-scale changes in water column properties and has also become increasingly important as autonomous platforms and sensors rapidly begin to fill the world's oceans. Many autonomous biogeochemical sensors are prone to drift and rely on GLODAP data and methods such as linearly interpolated regressions (LIRs; Carter et al. (2016, 2018) or machine-learning methods such as CANYON/CONTENT (Bittig et al., 2018, Sauzède et al. 2017) for ongoing quality control after deployment. GLODAP also serves as a benchmark for background concentrations in ocean and earth system models.

Where available, the researchers have also added isotopic data for $\delta$13C, $\delta$18O, and $\Delta$14C which are not quality controlled/adjusted in the same way as the core GLODAP variables but can provide context for the other data.

They have also added discrete fCO2 values which will be useful in addressing inconsistencies in the carbonate system variables. Importantly, fCO2 has not been subjected to any secondary QC. There has also been more extensive use of CANYON-B and CONTENT predictions to evaluate offsets in nutrients and CO2 data.

One important change that has been made to this version is that there is no internal consistency evaluation of seawater CO2 chemistry variables to evaluate pH. This leads to an inconsistency between the pH data for cruises added in this version, and pH data in previous versions of GLODAP. My understanding is that this will likely manifest as a bias, and not a random uncertainty. This potential bias is indeed encompassed by the stated consistency of "0.01 to 0.02 pH units," but will be critically important for those using this dataset and should be explained more clearly earlier in the manuscript, and perhaps even in the abstract. I also do not think that the consistency for pH should be

stated as a range. Yes, it varies by region but unless each region/cruise/data point has its own uncertainty estimate, the overall consistency should be stated as $\pm$ 0.02 pH units. If it is the case that there is only one region where the consistency is $\pm$ 0.02 pH units, and the rest of the ocean is closer to $\pm$ 0.01, then that region should be explicitly defined.

The original and adjusted data, a detailed adjustment table, and a "known issues" document are available online at the links provided in several formats, and as both global and regional subsets. The "known issues document" is updated regularly and users are encouraged to consult that document when using the data products and identify new issues when they find them.

I was also expecting to hear if/when the next GLODAP gridded product will be produced. Will it always only come with "major" GLODAP updates or are there any plans to do incremental updates?

Specific comments:

Line 249: An adjustment of -3 $\mu$mol/kg is made for a cruise which has a mean offset of 3.68 $\mu$mol/kg. Are adjustments always whole numbers? If so, do you always round down?

Line 251: Because they are an exception, provide more detail about how these eight Japanese Sea cruises were adjusted.

Line 319-320: Needs editing for clarity

Lines 280-282: While it is stated that TAlk estimated from 67 times salinity is sufficient for such pH conversions, it would be useful to explicitly state the amount of uncertainty introduced to pH by such a TAlk approximation.

Lines 427-429: Why was this decision made to replace measured values with calculated values?

[Figure]

Lines 537-541 and 558-559: It is acknowledged twice in the summary that the surface data are both seasonally biased and not examined for consistency in GLODAP. This is an important caveat and should be stated in the introduction.

Figures 3, 5, 8, 10: Include a legend for the colors
* * *

---

## Short Comment (SC1) · 24 Sep 2020

**Short summary**

The authors present an update of the GLODAPv2.2019 data product, by adding new data from 106 cruises. Before addition, observations of 12 core variables have undergone a primary (f flag) and secondary (qc flag) quality control. The secondary quality control is based on the comparison of new data with those contained within GLODAPv2.2019. Adjustments were - if necessary - applied to the new data, in order to correct for biases between measurements from different cruises, but preserve temporal trends in the variables. The merged data product includes observations from 946 cruises and extends until 2019.

**General comments**

[Figure]

The overall quality of this data product and its description in the companion manuscript appear very high. I have no general comments which would require a revision of fundamental aspects of the data set as a whole. The updated product GLODAPv2.2020 is an invaluable contribution for the scientific community and an essential prerequisite to reach the stated goal of documenting "the state and the evolving changes in physical and chemical ocean properties, e.g., the inventory of the excess CO2 in the ocean". This review is written from the perspective of a new user of the product.

**Specific comments**

Following specific issues were identified and might (if taken into account) require a revision of some aspects of the data product:

-l.412: "Neutral density was calculated using Sérazin (2011)." It should be noted that the reference given here refers to a master thesis and that the proposed polynomial approximation of neutral density in this thesis has not undergone peer review. Furthermore, polynomials were fitted to a preliminary neutral density data set with known issues (pers. comm. P. Barker and G. Sérazin). To take those limitations into account, the computed density variable gamma could either be revised, removed or labelled as preliminary in the main text.

-It might be helpful for some users if the f flag value would distinguish between interpolated and calculated values.

-l.190: It is stated that "not all offsets larger than the initial minimum limits have been adjusted for. ... Conversely, in some cases where data and offsets were very precise and the cruise had been conducted in a region where variability is expected to be small, adjustments lower than the minimum limits were applied." I was wondering whether at all an initial minimum adjustment limit needs to be defined and what the added value of this definition is. Would it be possible to define an offset-to-precision ratio that could rigorously be applied to all decisions?
-l.249: An adjustment of -3 $\mu$mol kg-1 was applied, although an offset of 3.68 $\pm$ 0.83 $\mu$mol kg-1 was found. Is this difference intentional? What is the general rule on how the adjustment values are set?

**Technical corrections**

Following comments address the presentation of the data product, and cover also aspects that are not purely technically:

-The presentation of the flagging scheme could be improved, aiming at clarity from a user perspective. Taking table 2 as an example, it confused me that labels 0-9 are presented, whereas the data product only uses f flag values 0, 2, and 9. Readers currently need to refer to footnotes in column "Merged product files" to find out that WOCE flags 6 and 7 were set to 2, whereas 3, 4, 5, and 8 were set to 9. Furthermore, the term "Not used" might add to the confusion, as it can easily be misinterpreted as "observations were not used" rather than the intended "the flag value was not used". Starting table 2 with the first column indicating f flag values that are actually used in the data product would greatly improve clarity and avoid potential misinterpretation of the flagging scheme. Likewise, in table 5 rownames (first column) are not intuitive. I was wondering what -888 does stand for. Does this label occur in the data set? Finally, several important information about flags are given in section 3.3.2 (Merging), but might be better placed in 3.1 (Data assembly and primary quality control) and 3.2 (Secondary quality control).

-l.45: The entire data product contains "measurements from more than 1.2 million water samples". However, this number decreases significantly when the number of available core variables is considered. As an example, I found in the merged master file <0.5 million dissolved inorganic carbon (tco2) observations and <10.000 observations with all core variables being available (in both cases ignoring f and qc flags). To this end, readers might benefit from a more detailed description of the data set. Giving expected row numbers for a few exemplary combinations of subsetting conditions would be valuable

for users to check if they handle the data set correctly.

-l.51: Adjustments are applied in a way that takes "into account any known or likely time trends or variations in the variables evaluated". However, I could not figure out in which way an unwanted bias correction is avoided, in particular with respect to variables for which a temporal change is expected, such as dissolved inorganic carbon (tco2). Maybe this was covered in previous versions of this living document, but I would find it useful and appropriate if this information could be included.

-Some qualitative statements could be replaced by more quantitative and exact descriptions. Examples include:

*l. 226: "In areas where a strong trend in salinity was present": What exactly is a strong gradient?

*l. 259: "A few new cruises had no or very few valid crossovers with GLODAPv2 data" What means very few here?

*l. 268: "exact selection determined based on the statistical robustness of the fit, as evaluated using the coefficient of determination (r2) and root mean square error (RMSE)" How exactly were r2 and RMSE evaluated jointly? Was one given preference under certain conditions?

-l.173: It is stated that "Missing numbers are indicated by -999, with trailing zeros to comply with the number format for the variable in question", but in the file "GLODAPv2.2020_Merged_Master_File.csv" all NAs seem to be coded as "-9999". Description and data product should be checked for coherence.

-l.197: Crossover comparisons, multi-linear regressions (MLRs), comparison of deep-water averages and predictions made with CANYON-B and CONTENT are introduced and were used to identify offsets. Information about which method was finally used to judge and if necessary adjust individual cruises seems to be missing.

-l.510: Definition of boundaries for Arctic, Atlantic, Pacific, and Indian ocean (or a

reference to the applied basin mask) is missing. Ideally, basin boundaries could also be displayed in Fig. 9.

-l.519 This links seems not to work: https://www.ncei.noaa.gov/products/ocean-carbon-data-system/oceans/GLODAPv2_2020/

L.536-545: Some information on data coverage appears for the first time in the summary but might deserve a dedicated chapter.

---

## Author Comment (AC1) · 11 Nov 2020

Response to review by referee #1, Dr. Matthew Humphreys

We thank Dr. Humphreys for the helpful comments and suggestions, each one is addressed below (comment in black, response in red).

The new cruise datasets added to GLODAP in this release constitute a substantial update to this already invaluable data product. The manuscript is clearly written and virtually ready to publish as it is. The first section of my comments below raises a few minor issues that the authors should consider before publication of this paper. The second section contains broader suggestions that might benefit future releases, but which are not necessary to include in this version.

**1 Comments for this manuscript**

1.1 Version naming convention
The new version number/naming convention outlined in lines 146–147 is intuitive and clear to follow. It could be more strongly emphasised here that the exact version number used should always be reported in studies, rather than making a generic reference to GLODAP.
Agreed
- Changes made: The following sentence has been added to the second final paragraph of the introduction "The exact version number and release year (if appended) of the product used should always be reported in studies, rather than making a generic reference to GLODAP."

It might be helpful to also explicitly commit to what may and may not be changed between different levels of version release. For example, in the "minor" version increments new cruises may be added but data that was already there will not change (with the exception of bug fixes, such as described in section 3.3.1), whereas it sounds like a "major" version increment would involve a reanalysis of the entire dataset, in which the adjustments applied to existing datasets could be more fundamentally altered.
Even if it's not exactly as I've described, some sort of explicit commitment like this could be helpful — users who switch to a newer version could immediately know what they can rely on to be consistent, and what changes they need to watch out for — and now, as the new versioning system is introduced, seems like a good opportunity to do this.
This is a good suggestion.
- Changes made: The following two paragraphs have been added at the end of the introduction (part of the material appeared at the end of Section 2, which is now shorter. Being fundamental to the procedures, we believe it fits better in the introduction):
  "Within this there are two types of GLODAP updates: full and intermediate. Full updates involve a reanalysis, notably crossover and inversion, of the entire dataset (both historical and new cruises) and all adjustments are subject to change. This was carried out for GLODAPv2. For intermediate updates, recently-available data are added following quality control procedures to ensure their consistency with the cruises included in the latest GLODAP release. Except for obvious outliers and similar types of errors (Sect. 3.3.1), the data included in previous releases are not changed during intermediate updates. Additionally, the GLODAP mapped climatologies (Lauvset et al., 2016) are not updated for these intermediate products. A naming convention has been introduced to distinguish intermediate from full product updates. For the latter the version number will change, while for the former the year of release is appended. The exact version number and release year (if appended) of

the product used should always be reported in studies, rather than making a generic reference to GLODAP.
Creating and interpreting the inversions, and other checks of the full data set needed for full updates are too demanding in terms of time and resources to be preformed every year or two-years. The aim is to conduct a full analysis (i.e., including an inversion) again after the third GO-SHIP survey has been completed. This completion is currently scheduled for 2023, and we anticipate that GLODAPv3 will become available a few years thereafter. In the intermin, presented here is is the second intermediate update, which adds data from 106 new cruises to the last update, GLODAPv2.2019 (Olsen et al., 2019)."

**1.2 Carbonate ion measurements**

The "four variables" statement in line 360 ignores the increasing reliability of carbonate ion measurements (e.g. Sharp and Byrne, 2019). I suggest to modify this statement accordingly; it is not really necessary to specify "four" or any specific number here at all.
Agreed.

- Changes made: "four" has been deleted here, and in other places were this number was mentioned as the number of measurable sea water $CO_2$ chemistry variables.

**1.3 pH adjustments — or not**

It would be useful to recap that pH adjustments were not applied to the new data in this version where this is mentioned in the summary on lines 554–555.
Agreed

- Changes made: The following text has been added to the end of the paragraph in question: "No pH data were adjusted for this version, but we note that this is largely a consequence of problems in establishing a reasonable pH baseline level in the deep northwest Pacific (Sect. 4.2). A comprehensive analysis of all available pH data in that region should be conducted for the next update".

**1.4 Figures**

The axis labels and other text notes on a couple of the figures are a bit too small relative to the figure size, making reading difficult (e.g. Figure 3).
Indeed, this is a problem for some of the figures, Figure 3 and 6, in particular. This problem arises as a consequence of downsizing of the submitted pdf, when the ESSD header is added to convert it into a discussion paper. We will take care during the production of the final paper to ensure text and notes on all figures are legible.

Although you can work these out from context — if you are familiar with the field — several of the figures are missing axis labels and units for the variables shown (e.g. Figures 3 through 6).
Thank you for pointing this out. Figures 3-5 are produced by the various QC algorithms, where context is clear, but we readily acknowledge that labels and units should be stated in the paper, so we have included this information in the captions. For Figure 6, we have also added an explanation on what is shown for the various variables.

- Changes made: Captions for Figure 3-6 have been revised.

**1.5 Typos**

Abstract: add a comma after "discrete **f**CO**2**" on line 56. Change "bias corrected product" to "bias-corrected product" on line 60.

- Changes made: Corrected

I suggest to change "are released regularly" to "will be released regularly" on line 145.
This sentence has been removed, following the changes in Sect 1 and 2 in response to your comment 2, on explicitly committing to what may and may not be changed between different levels of version release.

Summary: the sentence on lines 554–555 is missing a full stop at the end.
- Changes made: Full stop added.

**2 Suggestions for future releases**
The following points are not revisions that are necessary for this publication, but rather ideas that could be taken under consideration for future releases of GLODAP.

2.1 Expand dataset sourcing
The latest GEOTRACES Intermediate Data Product (Schlitzer et al., 2018) contains some datasets with the core GLODAP variables that are not included in this GLODAP release. While it's unreasonable to expect the GLODAP team to continually seek out new data from an endless list of sources, it may be worth including the GEOTRACES IDPs for future versions given the typically high quality of the carbonate system data therein, abundance of auxiliary variables to aid secondary QC, and consistent data format.
Thanks. We will scrutinize this dataset for cruises to include in the next version of GLODAP.

2.2 Accept carbonate ion measurements
As noted above, carbonate ion measurements are now becoming usefully reliable (e.g. Sharp and Byrne, 2019) and becoming more widespread. Accepting this type of data into GLODAP would be a natural extension to the current set of core variables, adding a new dimension to some applications of the GLODAP database such as evaluating dissociation constants based on over-determined data points (e.g. Sulpis et al., 2020).
Thanks for the suggestion. We do strive to increase the utility of GLODAP for evaluation of dissociation constants and other factors that biases the measurements. Plans are on the table for preparing a product with all of our alterations removed (adjustments, interpolations, calculations etc.); i.e. all data 'as reported', in a uniform format.
The GLODAP Reference Group discussed the suggestion of including carbonate ion measurements in the product, and we came to the conclusion that it is premature as unresolved issues with these measurements remain; specifically there are too few measurements to perform secondary QC, as carbonate ion is measured by few groups and (similar to pH) there is no certified standard to evaluate accuracy. Probably the main issue is that after the seminal work by Byrne and Yao (2008), four other manuscripts (Easley et al. 2013; Patsavas et al., 2015; Sharp et al., 2017; Sharp and Byrne, 2019) were published with modifications in the reagents, equations and other method settings, consequently the method is still under development and still improving.

2.3 Update carbonate system calculations
The analysis here still uses CO2SYS for MATLAB v1 (van Heuven et al., 2011). Updating to at least CO2SYSv2 (Orr et al., 2018) would enable uncertainty propagation — given that some calculated marine carbonate system variables are reported, it would be useful to also propagate uncertainties from the measured variables and dissociation constants into the calculated variables.

Updating further still to the recently released CO2SYSv3 (Sharp et al., 2020) would also enable calculations with carbonate ion as an input variable, if these measurements were to be accepted in future GLODAP releases. Ammonia and sulfide speciation are also included in

the alkalinity equation as of CO2SYSv3, which could improve the accuracy of marine carbonate system calculations in areas where these species are significantly abundant

Thanks, this is a useful reminder. We plan to use the updated CO2SYS software for future versions and inclusion of robust uncertainty estimates is a priority for GLODAP.

References:

Byrne, R. H. and Yao, W. S.: Procedures for measurement of carbonate ion concentrations in seawater by direct spectrophotometric observations of Pb(II) complexation, Mar Chem, 112, 128-135, 2008.

Easley, R. A., Patsavas, M. C., Byrne, R. H., Liu, X., Feely, R. A. and Mathis, J. T.: Spectophotometric measurement of calcium carbonate saturation states in seawater, Environ. Sci. Technol., 47, 1468-1477, 2012

Patsavas, M. C., Byrne, R. H., Yang, B., Easley, R. A., Wanninkhof, R., and Liu, X. W.: Procedures for direct spectrophotometric determination of carbonate ion concentrations: Measurements in US Gulf of Mexico and East Coast waters, Mar Chem, 168, 80-85, 2015.

Sharp, J. D. and Byrne, R. H.: Carbonate ion concentrations in seawater: Spectrophotometric determination at ambient temperatures and evaluation of propagated calculation uncertainties, Mar Chem, 209, 70-80, 2019.

Sharp, J. D., Byrne, R. H., Liu, X. W., Feely, R. A., Cuyler, E. E., Wanninkhof, R., and Alin, S. R.: Spectrophotometric Determination of Carbonate Ion Concentrations: Elimination of Instrument-Dependent Offsets and Calculation of In Situ Saturation States, Environ Sci Technol, 51, 9127-9136, 2017.

---

## Author Comment (AC2) · 11 Nov 2020

Response to review by referee #2, Dr. Nicolas Metzl

We thank Dr. Metzl for the helpful comments and suggestions, each one is addressed below (comment in black, response in red).

General comments:
Since 15 years GLODAP data-bases (from 2004 to 2019, including CARINA, PACIFICA) are widely used in the community, not only to evaluate the change of CO2 in the ocean or acidification (e.g. Gruber et al 2019; Jiang et al 2019), but also to compare and validate ocean and climate models (e.g. CMIP5, Bronselaer and Zanna, 2020 for a recent publication). The GLODAP data-set is also an important synthesis for GOA-ON activities and to construct climatology (e.g. Broullón et al, 2020).

Here, authors present an updated version of the GLODAP effort. This includes 106 new cruises quality controlled (QC), inclusion of new fCO2 observations (not QCed) and comparison of secondary QC with reconstructed properties using neural network methods (named CANYON-B and CONTENT).

The effort consists mainly in (i) format and check the data received from PI or available in different locations (NCEI/OCADS, PANGAEA, CCHDO), (ii) performed a secondary QC to identify data biases (if any) and separate from real temporal changes of the properties that could be low relative to the mean concentrations and (ii) construct final formatted products with adjusted data and associated flags for easy use at global or regional scales.

The paper is basically structured from the previous manuscript (Olsen et al 2019) and I therefore have only few comments regarding this new version (v2020). Most suggestions are for clarity, here thinking to readers that would discover only now the GLODAP project (e.g. new students in the field).

As fCO2 data are now included, GLODAP is in a way a companion data-base to SOCAT dedicated to surface fCO2 data (Bakker et al 2016) also annually updated (Bakker et al 2020). Both products were already used together for specific analysis (e.g. comparing pH fields from GLODAP and SOCAT, Jiang et al 2019). It might be useful for future to attempt incorporate fCO2 data that are in GLODAP but not yet in SOCAT. In this context few words might be added at the end in the conclusions/perspectives.

This is an interesting suggestion, thanks. There are indeed many sources of $f$CO$_2$ data, and there are also potentially many issues related to the various measurement techniques and different levels of, and approaches to, their QC. For GLODAPv2.2020, $f$CO$_2$ was not quality controlled. A unified look at ocean $f$CO$_2$ data seems worthwhile but would be very demanding, in particular related to differences in sampling strategies.

- Changes made: The following sentence was added to the second paragraph of Sect. 6 Summary, to make it clear that the $f$CO$_2$ data in GLODAP have not been subjected to quality control: "The number of measured $f$CO$_2$ data are 33 924; note that these data were not subjected to quality control."

  The following sentence has been added at the very end of Sect. 6 Summary, to make it clear that QC of $f$CO$_2$ data is needed, although at this stage we are not in a position to suggest any particular procedure: "As mentioned above, the included $f$CO$_2$ data have not been subjected to quality control, therefore no uncertainty estimate is given for this variable. This should be conducted in future efforts."

In this version, authors used CANYON-B and CONTENT methods (I think this was not systematically performed in v2019). This is a new and an elegant way to check and compare secondary control (and bias if any). This is a new step in GLODAP that might be recalled in the abstract for this version.

Thank you. We now mention this in the abstract.

- Changes made: We have added the following sentence to the abstract: "Comparisons to empirical algorithm estimates provided additional context for adjustment decisions, this is new to this version."

Something not very clear concerns the QC for historical cruises. With the new cruises in hand, I was not sure at the start if the QC of previous cruises in the same regions has been checked again and would lead to new corrections for cruises already in GLODAPv1, CARINA or v2019. However, as specify in the manuscript (line 145) I understand that a complete revision of QC would be performed in 2023 (after 3d GO-SHIP).

We realise that this is mentioned rather late in the manuscript, but hope that the paragraph on the different types of GLODAP updates now included in the introduction in response to the comment from Matthew Humphreys, clarifies this early on.

- Changes made: The following paragraphs have been added at the end of the introduction:
  "Within this there are two types of GLODAP updates: full and intermediate. Full updates involve a reanalysis, notably crossover and inversion, of the entire dataset (both historical and new cruises) and all adjustments are subject to change. This was carried out for GLODAPv2. For intermediate updates, recently-available data are added following quality control procedures to ensure their consistency with the cruises included in the latest GLODAP release. Except for obvious outliers and similar types of errors (Sect. 3.3.1), the data included in previous releases are not changed during intermediate updates. Additionally, the GLODAP mapped climatologies (Lauvset et al., 2016) are not updated for these intermediate products. A naming convention has been introduced to distinguish intermediate from full product updates. For the latter the version number will change, while for the former the year of release is appended. The exact version number and release year (if appended) of the product used should always be reported in studies, rather than making a generic reference to GLODAP.
  Creating and interpreting inversions, and other checks of the full data set needed for full updates are too demanding in terms of time and resources to be preformed every year or two-years. The aim is to conduct a full analysis (i.e., including an inversion) again after the third GO-SHIP survey has been completed. This completion is currently scheduled for 2023, and we anticipate that GLODAPv3 will become available a few years thereafter. In the intermin, presented here is is the second intermediate update, which adds data from 106 new cruises to the last update, GLODAPv2.2019 (Olsen et al., 2019). "

Also, many colleagues used the GLODAP gridded products that were constructed from GLODAP-v2 (Lauvset et al 2016). Will you also revisiting this gridded product now or wait for the 2023 version ? This might be specified in the manuscript.

The gridded product will not be updated now. The changes would likely be rather small, as the main source of uncertainty in the gridded product is lack of observations in certain regions. The data added in GLODAPv2.2019 and GLODAPv2.2020 are mostly repeat observations, extending the coverage in time and not in space. We cannot commit, now, to making new climatologies for v3. This depends on funding. Therefore, we simply add a statement that the intermediate products are not accompanied by a gridded product

update.

- Changes made: The sentence "Additionally, the GLODAP mapped climatologies (Lauvset et al., 2016) are not updated for these intermediate products." has been included in the second final paragraph of the introduction.

Another remark concerns the new cruises to be added in GLODAP. I understand that new cruises (106) were recently obtained from NCEI or PANGAEA or from PIs. However, I suspect there are many other cruises in the community (published) and it would be useful to find the best way to get more cruises in the future and invite new PIs to contribute.

Yes, there is certainly room for improvement. Right now, apart from close interaction with GO-SHIP and CCHDO, the level of formalization for addition of data is very low. While no changes were made to this end in the manuscript, we will explore ways to obtain more publicly available datasets.

Overall, I recommend publication after few minor revisions.

Below I list specific and minor comments (mostly details for clarity for a reader who discover Glodap for the first time). At the end of the review few technical questions regarding the files on-line.
;;;;;;;;;;;;;;;;;;;

Specific comments:

C-01: Title: The title includes only acronyms of the project (GLODAP). Would it be useful to recall that this concerns Ocean biogeochemical observations in the water column? A Suggestion for a title: "An updated version of global interior ocean biogeochemical observations, GLODAPv2-2020".

That is a good suggestion

- Changes made: Title has been changed to "An updated version of the global interior ocean biogeochemical data product, GLODAPv2.2020"

C-02: Page 2, line 44: "the inclusion of available discrete fugacity of CO2 (fCO2) values in the merged product files". Does this new inclusion concerns only the new cruises added in v2020 or did you also add this parameter for historical cruises ? (this is specify later, Line 369).

- Changes made, added "(also for historical cruises)" to sentence in question.

C-03: Page 4, Line 121: "The data collected across the Davis Strait". Maybe specify where is the Davis Strait for those not familiar with the Indian Ocean….(….Atlantic of course) ☺

- Changes made, added "between Canada and Greenland" after "Davis Strait"

C-04: Page 5, Line 175: For new users: Not sure to clearly understand all Flag definitions listed in Table 2.

Indeed, this Table is a bit brief, and may lead to misunderstandings, as also pointed out by Jens Muller in his short comment.

- Changes made:
  - We have added a citation to Swift (2010) in the Table header, which provides full details on the flags used in the exchange format original data files
  - We have expanded the table caption to make it clear that the flagging scheme in the merged product files is simplified (added text is underlined):

"Table 2. WOCE flags in GLODAPv2.2020 exchange format original data files (briefly; for full details see Swift, 2010) and the simplified scheme used in the merged product files"

- o We have added the underlined text in the paragraph of section 3.1 where Table 2 is first mentioned: "Each data column (except temperature and pressure, which are assumed "good" if they exist) has an associated column of data flags. For the original data exchange files, these flags conform to the WOCE definitions for water samples and are listed in Table 2. For the merged and adjusted product files these flags are simplified: questionable (WOCE flag 3) and bad (WOCE flag 4) data are removed and their flag set to 9. The same procedure is applied to data flagged 8 (very few such data exist). WOCE flags 1 (Data not received) and 5 (Data not reported) are also set to 9, while 6 (Mean of replicate measurement) and 7 (Manual chromatographic peak measurement) are set to 2, if the data appear good. Also, in the merged product file a flag of 0 is used to indicate a value that could be measured but is somehow approximated: for salinity, oxygen, phosphate, nitrate, and silicate, the approximation is conducted using vertical interpolation; for seawater $CO_2$ chemistry variables ($TCO_2$, TAlk, pH, and $fCO_2$), the approximation is conducted using calculation from two measured $CO_2$ chemistry variables (Sect 3.2.2). Importantly, interpolation of $CO_2$ chemistry variables is never preformed and thus a flag value of 0 has unique interpretation."
- o For the 'Merged product files' column in Table 2 we have changed "Not used" to "Flag not used"

C-05: Table 2: for clarity, it might be useful to assign different flag for interpolated and calculated values (both flag 0). Maybe for the next version.

- • Changes made. To be clear about the unique interpretation of the 0 flag for different variables, we have added the following sentences in Sect. 3.1: "Also, in the merged product file a flag of 0 is used to indicate a value that could be measured but is somehow approximated: for salinity, oxygen, phosphate, nitrate, and silicate, the approximation is conducted using vertical interpolation; for seawater $CO_2$ chemistry variables ($TCO_2$, TAlk, pH, and $fCO_2$), the approximation is conducted using calculation from two measured $CO_2$ chemistry variables (Sect 3.2.2). Importantly, interpolation of $CO_2$ chemistry variables is never preformed and thus a flag value of 0 has unique interpretation."

C-06: In table 2, you list "b" "Data are not included in the GLODAPv2.2020 product files and their flags set to 9. " Does that mean that original flag 3 (Questionable but sometimes maybe real signal) are not included in the files ? However this is explained later, line 395
Yes, these are removed from the product file. We now explain this in the paragraph that introduces the table (see response to C-04).

C-07: In table 2, you list "c" for replicate: "Data are included, but flag set to 2 ". This suggests that all replicate are acceptable (or some were also identify as outliers and thus moved to flag 9 or deleted ?).
We now clearly state in the paragraph that introduces this table, that replicates are only kept if the value appears valid, please see response to C-04.

C-08: Page 6, Line 197: "comparison of deep-water averages". Specify the layers here ? How this is selected in the high latitude (e.g. bottom water formations, where anthropogenic CO2

is found to be relatively high in water column ?).
This is the introductory paragraph for Section 3, stating what is to be presented in the subsections to come, among them Sect. 3.2.2, where the full details of the comparisons and what depth layers are used are provided. To avoid repetition, we do not go into these details here.

C-09: Page 6, Line 200: Add reference to CANYON-B and CONTENT (first time listed here) ?
  • Changes made: Reference to Bittig et al., 2018, added.

C-10: Page 6, Line 226: "In areas where a strong trend in salinity was present". Any example for this version ?
This is a leftover from the earlier versions of this paper; no strong salinity trends were present in the crossovers evaluated.
  • Changes made: sentence deleted.

C-11: Page 7, Line 235: "convection occurs (such as the Nordic, Labrador, and Irminger seas)". How do you select the layer in region of bottom water formation (e.g. SR03 for this version) ? Might be interesting for new readers to show another QC example (as presented in Figure 3 for the North Pacific).
Whether to use 1500 or 2000 dbar is determined on a case by case basis, by looking at the crossover comparisons for the two options, with respect to the accuracy of the information provided on the comparability between the data and whether changes in some layers seems related to actual change. In regions of bottom water formation change is expected, and results are scrutinized in light of this. We have revised the text to make it clear that subjective choices are involved, and that we always evaluate the results for presence of actual change, in order to not adjust this away.
  • Changes made: The text on depth limits for crossover analysis has been extended and revised:
    "Either the 1500 or 2000 dbar depth surface was used as upper bound, depending on the number of available data, their variation at different depths, and the region in question. This was evaluated on a case-by-case basis by comparing crossovers with both depth limits and using the one that provided the most clear and robust information. In regions where deep mixing or convection occurs, such as the Nordic, Irminger and Labrador seas, the upper bound was always placed at 2000 dbar; while winter mixing in the first two regions is normally not deeper than this (Brakstad et al., 2019; Fröb et al., 2016), convection beyond this limit has occasionally been observed in the Labrador Sea (Yashayaev and Loder, 2016). However, using an upper depth limit deeper than 2000 dbar will quickly give too few data for robust analysis. In addition, even below the deepest winter mixed layers properties do change over the time periods considered (e.g., Falck and Olsen, 2010), so this limit does not guarantee steady conditions. In the Southern Ocean deep convection beyond 2000 dbar seldom occurs, an exception being the processes accompanying the formation of the Weddell Polynya in the 1970s (Gordon, 1978). Deep and bottom water formation usually occurs along the Antarctic coasts, where relatively thin nascent dense water plumes flow down the continental slope. We cautiously avoid such cases, which are easily recognizable. In order to avoid removing persistent temporal trends, all crossover results are also evaluated as a function of time (see below)."

C-12: Page 7, Line 238: Maybe recall that 49UP20160109 is new while 49UP20160703 was QCed in v2019.
  • Changes made: The underlined text has been added to this sentence: "As an

example of crossover analysis, the crossover for TCO$_2$ measured on the two cruises 49UP20160109, which is new to this version, and 49UP20160703, which was included in GLODAPv2.2019, is shown in Fig. 3."

C-13: The example in Figure 3 shows in 3a blue dots on the map, but I suspect these stations (far east) were not used to evaluate the QC.
This is correct indeed. Thank you for pointing this out, it is certainly worthwhile to mention that only stations shown in panel b are used for the crossover analysis.
- Changes made. The following clarification has been made in the caption of Figure 3: "Panel (a) show all station positions for the two cruises and (b) show the specific stations used for the crossover analysis."

C-14: Page 7, Lines 245-250: For 49UP20160109, maybe specify that no temporal changes was observed for salinity (i.e. you used TCO2 here, not normalized TCO2 as suggested in Line 227 for some cruises).
As mentioned under C-10, salinity normalization was not needed for any crossover, and therefore not mentioned in this manuscript anymore. Thus, we did not mention here that the data were not salinity normalized.

C-15: Page 7, Line 245: Figure 4 shows the TCO2 cross-over for 49UP20160109 versus GLODAPv2-v2019. The cruise 49UP20160703 is also plotted and thus was in GLODAPv2-v2019, although conducted after 49UP20160109 (just to clarify for a new user).
In response to comment C-12, we now mention that 49UP20160703 was included in GLODAPv2.2019.

C-16: Page 7, Line 256: "they are included in the product but with a secondary QC flag of 0 (Sect.6)". Sect 6 (?)
The statement on the lack of full QC on the Davis Strait cruises and the consequential assignment, and interpretation of, secondary QC flag 0 has been moved to Section 4.2 Adjustment Summary.
- Changes made: The following paragraph has been added at start of Section 4.2: "The secondary QC has 5 different outcomes, provided there are data. These are summarized in Table 5, along with the corresponding codes that appear in the online Adjustment Table and that are also occasionally used as shorthand for decisions in the coming text. The level of secondary QC varies among the cruises. Specifically, in some cases data were too shallow or geographically too isolated for full and conclusive consistency analyses. A secondary QC flag has been included in the merged product files to enable their identification, with "0" used for variables and cruises not subjected to full secondary QC (corresponding to code -888 in Table 5) and "1" for variables and cruises that were subjected to full secondary QC. The secondary QC flags are assigned per cruise and variable, not for individual data points and are independent of—and included in addition to—the primary (WOCE) QC flag. For example, interpolated (salinity, oxygen, nutrients) or calculated (TCO$_2$, TAlk, pH) values, which have a primary QC flag 0, may have a secondary QC flag of 1 if the measured data these values are based on have been subjected to full secondary QC. Conversely, individual data points may have a secondary QC flag of 0, even if their primary QC flag is 2 (good data). A 0 flag means that data were too shallow or geographically too isolated for consistency analyses or that these analyses were inconclusive, but that we have no reasons to believe that the data in question are of poor quality. Prominent examples of this for this version are the 10 new Davis Strait cruises: no data were available in this region in GLODAPv2.2019,

which, combined with complex hydrography and differences in sampling locations, rendered conclusive secondary QC impossible. As a consequence, most, but not all, of these data (some being excluded because of poor precision after consultation with the PI) are included with a secondary QC flag of 0."

C-17: Page 7, Line 259: "A few new cruises had no or very few valid crossovers with GLODAPv2 data." Which cruises ? Would it be relevant to add a column in Table- Annexe 1 with a remark specifying what kind of secondary QC has been performed for each cruise (e.g. Standard QC, MLR, no QC) ?
For the 106 new cruises, MLR and deep water averages were used in a complimentary fashion, i.e., none of secondary QC were only based on these types of analyses. We have revised Sect. 3.2.3 to convey this.
The type of secondary QC varies not only per cruise, but also per variable. Different types of QC (e.g. Standard QC, MLR, no QC) can be applied for different variables on certain cruises. The various QC types can also be applied in combination. It is not practically possible to include this information in Table – Annexe 1. The most important information regardless appears in the online adjustment table.
  • Changes made: The first sentences of Section 3.2.3 have been revised to: "MLR analyses and deep water averages, broadly following Jutterström et al. (2010), were also used for the secondary QC of salinity, oxygen, nutrients, $TCO_2$, and TAlk data. These approaches are particularly valuable when a cruise has either very few or no valid crossovers with GLODAPv2, but are used more generally to provide more insight on the consistency of the data. The latter was the case for the 106 new cruises; i.e., no adjustments were reached on the basis of MLR and deep water average analyses alone. "

C-18: Page 8, Section 3.2.3: I understand the description but what are the results and which cruise ? Would be interesting to show an example for a cruise that is QCed using MLR.
As no cruise was fully QC'd using MLR, we have not included such an example, but will consider this for the next version of GLODAP.

C-19: Page 8, Line 277: "Altogether 82 of the 106 new cruises included pH data." Here specify this is measured pH, not calculated (so there is no confusion with pH calculated for other cruises).
  • Changes made: Sentence revised to (new word underlined) "Altogether 82 of the 106 new cruises included measured pH data."

C-20: Page 8, Line 291: "The pH data of 840 of the 936 cruises in GLODAPv2.2020". Again, specify if pH data here were measured or calculated or both.
We agree that this is not clear, and not all of the 840 cruises included measured pH. This paragraph has been extensively expanded following comments from Dr. Williams, and the specific sentence has been altered to: "In contrast to past GLODAP pH QC,  evaluation of the internal consistency of $CO_2$ system variables was not used for the secondary quality control of the pH data of the 106 new cruises."

C-21: Page 8, Line 305: Maybe recall the mean uncertainty associated to CANYON-B and CONTENT (see table 1 in Bittig et al 2018, i.e. about twice the adjustment limits fixed for GLODAP listed in Table 3).
We are reluctant to mention specific uncertainties for CANYON-B and CONTENT. These vary with depth and with location, and specifically for nutrients, are stated in absolute terms (concentration) in Bittig et al. (2018), rather than relative as used for the adjustment limits,

so the comparability and transferability of directly stating these values is small. We do recognize the need for more clearly relaying that we did in fact explicitly consider these uncertainties in our assessment, however. Therefore we have revised and expanded the sentence in question.

- Changes made: The sentences:
  "Of course, we kept in mind that this relies on the accuracies of the T, S, and $O_2$ data and of CANYON-B and CONTENT in themselves. Used in the correct way and with caution this tool is a powerful supplement to the traditional crossover analyses. "

  has been replaced with the following:

  "Used in the correct way and with caution this tool is a powerful supplement to the traditional crossover analyses. Specifically, we gave no weight to comparisons were the crossover analyses had suggested that the S and/or $O_2$ data were biased as this would lead to error in the predicted values. We also considered the uncertainties of the CANYON-B and CONTENT estimates. These uncertainties are determined for each predicted value, and for each comparison the ratio of the difference (between measured and predicted values) to the local uncertainty was used to gauge the comparability."

C-22: Page 8, Line 305: As it is new results presented here (and probably also used in the next version), I think some more information is needed. For CANYON-B and CONTENT are you using results based on GLODAP-v2 data (Bittig et al 2018) or an updated version using GLODAPv2-2019. Is the comparison presented here (Figure 5) validate the QC for the new cruises or validate CANYON-B and CONTENT reconstructed fields? It is reassuring to get about the same results as CANYON-B and CONTENT were trained with GLODAP.
We already state that "These approaches were developed using the data included in the GLODAPv2 product" (line 299-300 in discussion paper). Moreover, from the text and context it is apparent that we validate the new cruises. Finally, we agree that the agreement is reassuring.

C-23: Page 8, Line 308: Figure 5: not easy to see the black dots (measured values).
This is true, and in large part a consequence of the overlap between the predicted and measured values. We prefer not editing the figure. One can see the black dots zooming in. We will add a sentence in the caption to explain that the black dots are in large part hidden by the red/blue dots.
- Changes made: The sentence in the caption explaining the color scheme, has been revised, new text underlined: "Black dots (which to a large extent hidden are by the predicted estimates) are the measured data, blue dots are CANYON-B estimates and red dots are the CONTENT estimates."

C-24: Figure 5: there is no units (to be added in captions ?).
- Changes made: Units have been stated in the caption.

C-25: Figure 5: Like for Figure 3 and 4, it would be nice to show another example, e.g. SR3 or Davis Strait ? Or an example where the comparison between QC from GLODAP and CANYON-B/CONTENT does not work (if any). This is a suggestion not absolutely needed.
Based on the current large numbers of figures in this manuscript, we chosen to not follow this suggestion.

C-26: Page 9, line 320: "Another advantage of CANYON-B and CONTENT is that by

considering the each data point in it self, primary QC issues has been revealed and corrected for some of the cruises." Which cruises ? Give some examples ?

We have revised the sentence and added an example.

- Changes made: The sentence in question has been revised to: "Another advantage of CANYON-B and CONTENT is that these procedures provide estimates at the level of individual data points, e.g., pH values are determined for every sampling location and depth where T, S, and $O_2$ data are available. Cases of strong differences between measured and estimated values are always examined. This has helped to identify primary QC issues for some variables and cruises, for example a case of an inverted pH profile at cruise 32PO20130829, which has been amended."

C-27: Page 9-10: Section 3.3.1. Lines 332-358: This is a list of revisions and would be better to move this section in an Annex but keep in Section 3.3.1 the fCO2 information (lines 359-375) as it is new data added in v2020.

While we agree that the list is tedious, we prefer to keep it the main text as this is very much what the intention of the manuscript is, documenting significant additions and *changes* to the dataset.

C-28: Page 10: Concerning fCO2, in the GLODAP files there are now both fCO2 measured and calculated in the same column. Authors indicate that all values were converted to 20_C. However, in the data-files, there are fCO2 values with fCO2temp fixed at -9999. I missed something here and not sure if all fCO2 values in the files are at the same temperature, pressure or at local temperature etc: : : Also, there are fCO2 values with flag 0 or 2. What was the criteria for fCO2 with flag 2 ? How users can easily separate the fCO2 measured and calculated in the files ? This is important to clarify if one uses both GLODAP (in surface) and SOCAT to merge both products.

We thank you for checking the product files carefully. Indeed, $fCO_2$ data without accompanying temperatures occurred. This is an error. The product files have been corrected now. $fCO_2$ data flagged 2 are measured, while $fCO_2$ values with flag 0 are calculated, as is the case for all seawater $CO_2$ data.

C-29: Page 10, line 364: "These calculated TAlk values were, however, not included in v2.2019." Does that mean that all TALK values with flag 0 in the files are only interpolated values (i.e. not calculated as an option suggested in table 2).

With the more extensive explanations of the flags added in Section 3.1 (see response to C-05) we hope that it has become clear that seawater $CO_2$ chemistry variables, such as TAlk, flagged 0, are not interpolated, only calculated.

Moreover, the sentence in question relates to the previous version of this product, v2.2019. We realize now, that this sentence might cause confusion and is unnecessary.

- Changes made: The sentence has been removed.

C-30; Page 11, Lines 397-398: For flags 6 and 7 now set to flag 2, recall that this only applied for valid data (i.e. obvious outliers deleted also for these replicates ?).

- Changes made: The underlined text has been added to the sentence: "All flags 6 (replicate measurement) and 7 (manual chromatographic peak measurement) were set to 2, provided the data appeared good."

C-31: Page 11, Line 399: "Missing sampling pressures or depths were calculated following UNESCO (1981)." This is obvious but maybe rewrite following: "Missing sampling pressures (resp. depths) were calculated from depths (reps. pressures) following UNESCO (1981)."

- Changes made: Revised according to suggestion.

C-32: Page 11-12, Lines 405 and 432: Flag 0 is used for both interpolated and calculated values. Why not using different flag ? (for next version)

As explained in response to comment C-05, interpretation of WOCE flag 0 is unique, and this is now clearly stated in Section 3.1. Nevertheless, we now also reiterate these principles in this section.

- Changes made: The underlines text has been added to the sentences in question:

    (Line 405) "Missing salinity, oxygen, nitrate, silicate, and phosphate values were vertically interpolated whenever practical, using a quasi-Hermetian piecewise polynomial. "Whenever practical" means that interpolation was limited to the vertical data separation distances given in Table 4 in Key et al. (2010). Interpolated salinity, oxygen, and nutrient values have been assigned a WOCE quality flag 0."

    (Line 432)"Calculated seawater $CO_2$ chemistry values have been assigned WOCE flag 0. Seawater $CO_2$ chemistry values have not been interpolated, so the interpretation of the 0 flag is unique."

C-33: Page 11, Line 416. Concerning the "Missing seawater CO2 chemistry variables". Are the calculated properties used only measured data (i.e. TALK and TCO2) or also interpolated values ? In other words, are the fCO2 and pH interpolated values based on calculated fCO2 and pH or recalculated with interpolated TALK/TCO2 ?

We hope that it is clear now, and also in the manuscript, that no seawater $CO_2$ chemistry variables were interpolated.

C-34: Page 13, Line 486: "For example, Arctic Ocean phosphate, Indian Ocean silicate and TCO2, and Pacific Ocean pH data all show considerable improvements." For Indian, in Table 6 improvement is for TALK, not TCO2 ?

Indeed, this is correct and has been amended.

- Changes made: TCO2 has been replaced with TAlk in the sentence in question.

C-35: Page 15, Line 544: Weatherall et al., (2015): not in references.

Thank you for pointing this out.

- Changes made: Weatherall et al., (2015) has been added to the reference list.

C-36: Now concerning the files, for curiosity I had a look at the Indian.cvs file and have few questions that could be also valid for other basin. The questions below are obvious for someone familiar with Glodap, but mainly addressed here to help new users.

C-36a: Why the QC flags for S or O2 are 0 for several cruises although flag WOCE are2 ? Is it because the secondary QC is not available for these cruises ?

This is correct. We have added text in Sect. 4.2 to explain this (see response to C-16).

C-36c: There are data withWOCE flag=0 for O2, Nitrate, Silicates, Phosphates, TCO2, TALK, pH, and associated to QC flag = 1. Is it because these are interpolated values for a cruise/station for which a secondary QC was performed ? If QC has been performed (QCF=1) one would expect a WOCE flag different from 0 ? I thought the QC is based on original data (not interpolated or calculated). Could that be clarified ?

This is correct. We have added text in Sect. 4.2 to explain this (see response to C-16).

C-36d: There are data with flag 9 associated to QC flag=1. Again, is it because QC flag (0,1)

are assigned for a cruise/station not for each data?

This is correct. We have added text in Sect. 4.2 to explain this (see response to C-16).

C-37: In the data files on-line (e.g. GLODAPv2.2020_Indian_Ocean.cvs) I would suggest to add units for each column.

Yes, and this has been discussed in the GLODAP group as well, and will likely be done for the next update.

C-38: And for next versions, I think for clarity a different flag should be assign for calculated (e.g. fCO2, pH) and interpolated values. This might help some users to select only measured+interpolated values. In references:

As stated earlier, the interpretation of WOCE flag 0 is unique for the different variables. As such there is no need to having a different flag for interpolated (salinity, oxygen, nutrients) vs calculated values ($TCO_2$, TAlk, pH, $fCO_2$). We hope this, now, is clear in the manuscript as well.

I think each reference should now have a DOI

Line 663: "Hood, E. M., Sabine, C. L., and Sloyan, B. M.: The GO-SHIP hydrography manual: A collection of expert reports and guidelines, 2010." Specify the publisher ? DOI ?

- Changes made: publication information has been completed to:
  Hood, E. M., Sabine, C. L., and Sloyan, B. M. (Eds).: The GO-SHIP hydrography manual: A collection of expert reports and guidelines, IOCCP Report Number 14, ICPO Publication Series Number 134, available at http://www.go-ship.org/HydroMan.html (last access: 16 October 2020), 2010.

---

## Author Comment (AC3) · 11 Nov 2020

Response to review by referee #3.

We thank the referee for the helpful comments and suggestions, each one is addressed below (comment in black, response in red).

This is a "living data" update document that discussed the addition of 106 cruises to the GLODAPv2.2019 data set. These data have been extremely valuable to the community and represent an important asset to maintain and update. The manuscript is well written and informative. I only have a few minor comments below.

Line 92-93: The authors don't distinguish between discrete and in situ sensor measurements here. I assume they are referring to CTD calibration problems with respect to the sensor measurements of salinity and oxygen, not the measurements of collected samples. Please clarify, particularly in light of the merging discussed in section 3.2.1.

Yes, indeed, we are referring to lacking calibration of the data from CTD mounted sensors.

- Changes made: Sentence revised to "For salinity and oxygen, lack of calibration of the data from the conductivity-temperature-depth (CTD) profiler mounted sensors is an additional and widespread problem, particularly for oxygen (Olsen et al., 2016)."

Lines 95-99: The manuscript uses some rather subjective terms without defining their meaning in this context. For example, "poor precision can render a set of data unusable" or "to minimize severe cases of bias". What is the definition of poor precision or severe bias?

We now provide more concrete information on what is meant with these terms, without going overboard with numbers and definitions as this is a general introduction, and as such we are reluctant to discuss details about each and every variable considered. Besides, the data are evaluated on a case-by-case basis, depending on region and availability of already existing data, for instance; we do not have a strictly enforced global set of limits.

- Changes made:
  The sentence "In rare cases poor precision can render a set of data unusable" has been revised to:
  "In rare cases poor precision - many multiples worse than that expected with current measurement techniques - can render a set of data of limited use."

  The sentence: Adjustments are applied on the data to minimize severe cases of bias"
  has been revised to:
  "Adjustments are applied to the data to minimize cases of bias that could be confidently established relative to the measurement precision for the variables and cruises considered. "

Lines 98, 108: There are a notable number of grammatical errors in the text that should be fixed. A couple of examples are, "Adjustments are applied on the data"(should be 'to the data') or "A particular important source" (should be 'A particularly important source'). Please review the entire document for these grammatical errors.

Thank you for pointing out these errors, which have been corrected. The text has been carefully read and corrected by all authors, many of whom are native English speakers. We hope the number of grammatical errors has been minimized.

Line 123-124: The authors decided to include cruises on the Merian, Meteor, and the Garcia del Cid that did not have any nutrient or carbon data. I thought nutrients and carbon were

the primary parameters for this data set. Why did the authors decide to include these data and not the thousands of other cruises that also do not have carbon data. This seem inconsistent with the goal of this project.

The emphasis for GLODAP is seawater inorganic carbon chemistry, as well as other carbon-relevant and related variables. This includes the transient tracers CFC-11, CFC-12, CFC-113 and $SF_6$, as these are frequently used to determine ocean inventories of anthropogenic carbon (e.g., Waugh et al., 2006). Rarely measured stable carbon isotopes are also relevant, as these are often used for the same purpose (e.g., Quay et al., 2017), and while we do not quality control such data, they are included to ensure their wider availability. There are not thousands of other cruises with such data. We have now included some text on these deliberations:

- Changes made: The following sentences have been included at the start of Section 2: "Not all cruises have data for all of the above-mentioned 12 core variables; for example, cruises with only seawater $CO_2$ chemistry or transient tracer data are still included even without accompanying nutrient data due to their value towards computation of, for example, carbon inventories. In some other cases, cruises without any of these properties measured were included – this was because they did contain data for other carbon related tracers such as carbon isotopes, with the main intention of ensuring their wider availability."

Line 150: define data center acronyms the first time they are used, or at least provide links to the data centers.

- Changes made, links to the data centers are now provided

Line 193-195: Were the original data generators consulted before adjustments were made to the data? I believe in the past there was a step that involved checking with the people that originally made the measurement to get their perspective on possible offsets.

Indeed, during preparation of the first version of GLODAP (Key et al., 2004), data originators were contacted for consultation on possible offsets. This practice was abandoned for GLODAPv2, with more than 700 cruises and over 1200 adjustments made, this became impractical. GLODAP is presently a volunteer effort and there is no capacity for routinely approaching principal investigators for every adjustment considered. However, members of the GLODAP Reference Group (i.e., the authors of this contribution) frequently possess first hand experience with the data, or are even the cruise PIs. In exceptional cases, for example where no primary QC seems to have been applied, we do reach out to the PIs.

Line 256: This is the first time that a -888 label is discussed in the text. What does this mean? The same comes in later with -777 and -666 labels.

Thank you for pointing this out. These labels hadn't really been properly explained in this manuscript, only in the GLODAPv2 article (Olsen et al., 2016). In addition, the text in the passage in question (i.e., line 256) better belong in Section 4.2, Adjustment summary as it mostly pertain results.

Changes made: The labels are now explained at the very start of Section 4.2, and presented in a new Table (Table 5). The text on the Davis Strait cruises, pointed out by the reviewer, has been moved from Sect 3.2.3, and used as an example of cruises not fully QCd. The first paragraph in Sect. 4.2 is now: "The secondary QC has 5 different outcomes, provided there are data. These are summarized in Table 5, along with the corresponding codes that appear in the online Adjustment Table and that are also occasionally used as shorthand for decisions in the coming text. The level of secondary QC varies among the cruises. Specifically, in some cases data were too shallow or geographically too isolated for full and conclusive consistency analyses. A secondary QC flag has been included in the merged

product files to enable their identification, with "0" used for variables and cruises not subjected to full secondary QC (corresponding to code -888 in Table 5) and "1" for variables and cruises that were subjected to full secondary QC. The secondary QC flags are assigned per cruise and variable, not for individual data points and are independent of—and included in addition to—the primary (WOCE) QC flag. For example, interpolated (salinity, oxygen, nutrients) or calculated ($TCO_2$, TAlk, pH) values, which have a primary QC flag 0, may have a secondary QC flag of 1 if the measured data these values are based on have been subjected to full secondary QC. Conversely, individual data points may have a secondary QC flag of 0, even if their primary QC flag is 2 (good data). A 0 flag means that data were too shallow or geographically too isolated for consistency analyses or that these analyses were inconclusive, but that we have no reasons to believe that the data in question are of poor quality. Prominent examples of this for this version are the 10 new Davis Strait cruises: no data were available in this region in GLODAPv2.2019, which, combined with complex hydrography and differences in sampling locations, rendered conclusive secondary QC impossible. As a consequence, most, but not all, of these data (some being excluded because of poor precision after consultation with the PI) are included with a secondary QC flag of 0. "

Lines 280-282: Why did the authors use the full GLODAPv2 data to estimate TAlk from Salinity. Wouldn't it make more sense to calculate an average ratio for the data from that cruise rather than use a global ratio that includes data from other oceans? Also, doesn't the ratio change with depth

TAlk is estimated here, with the purpose of converting pH measurement scale and/or reporting temperature/ pressure. The uncertainties introduced by a using global ratio instead of actually measured TAlk are very small. For the scale conversions the uncertainties are on the order of $10^{-7}$ pH units, which is fully negligible. For the temperature and pressure conversions the uncertainties are 0.001 pH units (evaluated using 2 standard deviations around the 67 ratio, i.e. TAlk/S = 67 ± 4.1 $\mu$mol/kg/permil). This is an order of magnitude smaller than the stated uncertainty for the pH in the merged product, 0.01-0.02 units.

Calculating the TAlk vs. Salinity ratio for the cruise in question is usually not possible since TAlk often has not been measured at all at these cruises (or very few measurements exist).

We do agree, though, that more sophisticated approaches exist for estimating alkalinity (Bittig et al., 2018; Broullon et al., 2019), and since Bittig et al. (2018) is already used to estimate missing PO4 and Si, it will be considered for missing TAlk data in future GLODAP updates.

- Changes made: We provide more quantitative information on the uncertainties introduced by the approximation, in section 3.2.4.

References:
Bittig, H. C., Steinhoff, T., Claustre, H., Fiedler, B., Williams, N. L., Sauzède, R., Körtzinger, A., and Gattuso, J.-P.: An alternative to static climatologies: Robust estimation of open ocean $CO_2$ variables and nutrient concentrations from T, S, and $O_2$ data using Bayesian Neural Networks, Frontiers in Marine Science, 5, 2018.
Broullon, D., Perez, F. F., Velo, A., Hoppema, M., Olsen, A., Takahashi, T., Key, R. M., Tanhua, T., Gonzalez-Davila, M., Jeansson, E., Kozyr, A., and van Heuven, S.: A global monthly climatology of total alkalinity: a neural network approach, Earth Syst Sci Data, 11, 1109-1127, 2019.
Olsen, A., Key, R. M., van Heuven, S., Lauvset, S. K., Velo, A., Lin, X. H., Schirnick, C., Kozyr, A.,

Tanhua, T., Hoppema, M., Jutterstrom, S., Steinfeldt, R., Jeansson, E., Ishii, M., Perez, F. F., and Suzuki, T.: The Global Ocean Data Analysis Project version 2 (GLODAPv2) - an internally consistent data product for the world ocean, Earth Syst Sci Data, 8, 297-323, 2016.

Quay, P., Sonnerup, R., Munro, D., and Sweeney, C.: Anthropogenic CO2 accumulation and uptake rates in the Pacific Ocean based on changes in the C-13/C-12 of dissolved inorganic carbon, Global Biogeochem Cy, 31, 59-80, 2017.

Waugh, D. W., Hall, T. M., McNeil, B. I., Key, R., and Matear, R. J.: Anthropogenic CO2 in the oceans estimated using transit time distributions, Tellus B, 58, 376-389, 2006.

---

## Author Comment (AC4) · 11 Nov 2020

Response to review by referee #4, Dr. Nancy Williams

We thank Dr. Williams for the helpful comments and suggestions, each one is adressed below (comment in black, response in red).

General Comments:

This is an update to the GLODAPv2.2019 by adding 106 new cruises from 2004-2019, expanding the coverage of GLODAP to 946 cruises over 47 years, 1972–2019. Most of the new cruises are from the western North Pacific and the Davis Strait, with a few from the Atlantic, South Indian, and U.S. West coast. The methods for primary and secondary quality control (QC) are essentially the same as in the earlier version. However, there has been no full consistency analysis of the entire data product as was done with the original GLODAPv2 product. A full consistency analysis will be performed in the future for the next GLODAP update (will be termed "GLODAPv3") which is set to occur after the completion of the third GO-SHIP survey around year 2023. The researchers have also fixed some minor errors in the GLODAPv2.2019 dataset.

Throughout the manuscripts the researchers discuss alternate ways of adjusting the dataset, and tend to take a conservative approach, saving any major changes for the next full GLODAP update, i.e., GLODAPv3. As such, this update could be considered by some to be incremental, but it should be noted that incremental and timely updates to GLODAP are critical to advancing ocean observing. GLODAP, and other such data products that have come before it, forms the backbone for studying largescale changes in water column properties and has also become increasingly important as autonomous platforms and sensors rapidly begin to fill the world's oceans. Many autonomous biogeochemical sensors are prone to drift and rely on GLODAP data and methods such as linearly interpolated regressions (LIRs; Carter et al. (2016, 2018) or machine-learning methods such as CANYON/CONTENT (Bittig et al., 2018, Sauzède et al. 2017) for ongoing quality control after deployment. GLODAP also serves as a benchmark for background concentrations in ocean and earth system models.

Where available, the researchers have also added isotopic data for _13C, _18O, and **D**14C which are not quality controlled/adjusted in the same way as the core GLODAP variables but can provide context for the other data.

They have also added discrete fCO2 values which will be useful in addressing inconsistencies in the carbonate system variables. Importantly, fCO2 has not been subjected to any secondary QC. There has also been more extensive use of CANYON-B and CONTENT predictions to evaluate offsets in nutrients and CO2 data.

One important change that has been made to this version is that there is no internal consistency evaluation of seawater CO2 chemistry variables to evaluate pH. This leads to an inconsistency between the pH data for cruises added in this version, and pH data in previous versions of GLODAP. My understanding is that this will likely manifest as a bias, and not a random uncertainty. This potential bias is indeed encompassed by the stated consistency of "0.01 to 0.02 pH units," but will be critically important for those using this dataset and should be explained more clearly earlier in the manuscript, and perhaps even in the abstract. I also do not think that the consistency for pH should be stated as a range. Yes, it varies by region but unless each region/cruise/data point has its own uncertainty estimate, the overall consistency should be stated as ± 0.02 pH units. If it is the case that there is only

one region where the consistency is ± 0.02 pH units, and the rest of the ocean is closer to ± 0.01, then that region should be explicitly defined.

Indeed, no internal consistency evaluation was conducted for pH for the data added in this version. No pH data were adjusted either. If adjustments had been made, they would adjust the data from the new cruises, to the pH values of cruises already part of GLODAP (which are used as reference) and evaluated in the earlier efforts. As such, this would not have led to inconsistencies between the pH data for cruises added in this version, and pH data in previous versions of GLODAP.

Regarding stating the consistency for pH as a range. We agree that this was somewhat murky in the submitted manuscript, and we now provide clearer reasoning and identify regions of high vs low uncertainty.

Changes made: The final paragraph of section 3.2.4, where these issues were discussed, have been substantially expanded, to: "In contrast to past GLODAP pH QC, evaluation of the internal consistency of $CO_2$ system variables was not used for the secondary quality control of the pH data of the 106 new cruises; only crossover analysis was used as supplemented by CONTENT and CANYON-B (Sect. 3.2.5). Recent literature has demonstrated that internal consistency evaluation procedures are subject to errors owing to incomplete understanding of the thermodynamic constants, major ion concentrations, measurement biases, and potential contribution of organic compounds or other unknown protolytes to alkalinity (Takeshita et al., 2020), which lead to pH dependent offsets in calculated pH (Álvarez et al., 2020; Carter et al., 2018): these may be interpreted as biases and generate false corrections. The offsets are particularly strong at pH levels below 7.7, when calculated and measured pH are different by on average between 0.01 and 0.02 units. For the North Pacific this is a problem as pH values below 7.7 can occur at the depths interrogated during the QC (>1500 dbar for this region, Olsen et al., 2016). Since any corrections, which may thus be an artifact, are applied to the full profiles, we assign an uncertainty of 0.02 to the North Pacific pH data in the merged product files. Elsewhere, the uncertainties that have arisen are smaller, since deep pH is typically larger than 7.7 (Lauvset et al., 2020), and at such levels the difference between calculated and measured pH is less than 0.01 on average (Álvarez et al., 2020; Carter et al., 2018). Outside the North Pacific, we believe, therefore that the pH data are consistent to 0.01. Avoiding interconsistency considerations for these intermediate products helps to reduce the problem, but since the reference data set (also as used for the generation of the CONTENT and CANYON-B algorithms) has these issues, a full re-evaluation, envisioned for GLODAPv3, is needed to address the problem satisfactorily. "

The original and adjusted data, a detailed adjustment table, and a "known issues" document are available online at the links provided in several formats, and as both global and regional subsets. The "known issues document" is updated regularly and users are encouraged to consult that document when using the data products and identify new issues when they find them.

I was also expecting to hear if/when the next GLODAP gridded product will be produced. Will it always only come with "major" GLODAP updates or are there any plans to do incremental updates?

There are no plans for making incremental updates to the GLODAP gridded product. The changes would likely be rather small anyhow, as the main source of uncertainty in the gridded product is lack of observations in certain regions. The data added in GLODAPv2.2019 and GLODAPv2.2020 are mostly repeat observations, extending the coverage in time and not in space. We cannot commit, now, to making new climatologies for the next full update. While we hope it will be possible, it will depend on the funding situation. Therefore, we

simply add a statement that the intermediate products are not accompanied by a gridded product update.

- Changes made: The sentence "Additionally, the GLODAP mapped climatologies (Lauvset et al., 2016) are not updated for these intermediate products." has been included in the second final paragraph of the introduction.

Specific comments:

Line 249: An adjustment of -3 μmol/kg is made for a cruise which has a mean offset of 3.68 μmol/kg. Are adjustments always whole numbers? If so, do you always round down?

Adjustments are typically round numbers relative to the precision of the variable considered. There are no particular rules about rounding down or up; we look for example, on whether there is a difference in the offset in recent vs older crossovers. We also consider additional evidence from the other methods. Here, we settled for -3 μmol/kg, as the CANYON-B and CONTENT analyses suggested a bias of 3.4 and 2.7 μmol kg$^{-1}$, respectively. This also helps to make the adjustment as small as meaningfully possible, in case there actually is an increasing trend in TCO$_2$ from uptake of anthropogenic carbon.

Changes made: The sentence in question has been revised to : "In this case -3 μmol kg$^{-1}$ was applied: this is somewhat less than indicated by the crossover analysis, but a smaller adjustment is supported by the CANYON-B and CONTENT results (Sect. 3.2.3). Adjustments are typically round numbers relative to the precision of the variable being considered (e.g., -3 not -3.4 for TCO$_2$ and 0.005 not 0.0047 for pH) to avoid the communicating that the ideal adjustments are known to high precision."

Line 251: Because they are an exception, provide more detail about how these eight Japanese Sea cruises were adjusted.

Changes made: The following paragraph has been added in section 4.2: "For the Sea of Japan cruises, (where two existed in GLODAPv2.2019 and six were added in this version - Sect. 3.2.2), the crossover results showed biased TCO$_2$ data for one of the older cruises (49HS20081021, which is now adjusted up by 6 μmol kg$^{-1}$), and biased TAlk data for two of the presently added cruises (49UF20111004 and 49UF20121024, adjusted up by 5 and 6 μmol kg$^{-1}$, respectively)."

Line 319-320: Needs editing for clarity.

This has now been edited for clarity, and we have included an example as well, following a suggestion by reviewer 2.

- Changes made: The text has been revised to: "Another advantage of CANYON-B and CONTENT is that these procedures provide estimates at the level of individual data points, e.g., individual pH values are determined for every sampling location and depth were T, S, and O$_2$ data are available. Cases of strong differences between measured and estimated values are always examined. This has helped to identify primary QC issues (outliers) for some variables and cruises, for example a case of an inverted pH profile at cruise 32PO20130829, which has been amended."

Lines 280-282: While it is stated that TAlk estimated from 67 times salinity is sufficient for such pH conversions, it would be useful to explicitly state the amount of uncertainty introduced to pH by such a TAlk approximation.

Yes, we agree.

- Changes made:  The following text has been added in Sect. 3.2.4: "The uncertainties introduced with this approximation are negligible (order 10$^{-7}$ pH units) for the scale conversions and order 10$^{-3}$ pH units for the temperature and pressure conversion (evaluated by repeating conversions with 2 times the standard deviation of the ratio,

i.e., 67 ± 4.1). This is sufficiently accurate relative to other sources of uncertainty, which are discussed below."

Lines 427-429: Why was this decision made to replace measured values with calculated values?

This decision was made when GLODAPv2 was prepared. Often, for such cruises where the number of measured data points for a $CO_2$ chemistry variable is much less than the number that can be calculated, the accuracy of the measured data cannot be confidently established – there are too few data for good crossover analyses – and it makes most sense replacing these with values calculated from the two other better QC'd variables. Evaluating the approprioate action on a per cruise basis is time consuming, so we made the decision to draw the line at less than 1/3 (of the combined number of calculated and measured values)

- Changes made: We have simplified the sentences a bit, and added the reason for replacing measured values "For calculations involving $TCO_2$, TAlk, and pH, if less than a third of the total number of values, measured and calculated combined, for a specific cruise were measured, then all these were replaced by calculated values. The reason for this, is that secondary QC of the few measured values was often not possible in such cases, for example due to a limited number of deep data avaliable"

Lines 537-541 and 558-559: It is acknowledged twice in the summary that the surface data are both seasonally biased and not examined for consistency in GLODAP. This is an important caveat and should be stated in the introduction.

- Changes made: We have added the following sentence to the introduction (in former line # 98: "The secondary quality controlled focused on deep data, where natural variability is mimimal"

Figures 3, 5, 8, 10: Include a legend for the colors

Figure 3 and 5 are produced by the crossover and CANYON-B/CONTENT software. It is not possible to add legends at this stage. The meaning of the colors are now explained in the caption.

- Changes made: Legends have been added to Fig. 8 and Fig 10.

---

## Author Comment (AC5) · 11 Nov 2020

Response to short comment #1, by Dr. Jens Müller

We thank Dr. Müller for the helpful comments and suggestions, each one is adressed below (comment in black, response in red).

**Short summary**
The authors present an update of the GLODAPv2.2019 data product, by adding new data from 106 cruises. Before addition, observations of 12 core variables have undergone a primary (f flag) and secondary (qc flag) quality control. The secondary quality control is based on the comparison of new data with those contained within GLODAPv2.2019. Adjustments were - if necessary - applied to the new data, in order to correct for biases between measurements from different cruises, but preserve temporal trends in the variables. The merged data product includes observations from 946 cruises and extends until 2019.

**General comments**
The overall quality of this data product and its description in the companion manuscript appear very high. I have no general comments which would require a revision of fundamental aspects of the data set as a whole. The updated product GLODAPv2.2020 is an invaluable contribution for the scientific community and an essential prerequisite to reach the stated goal of documenting "the state and the evolving changes in physical and chemical ocean properties, e.g., the inventory of the excess CO2 in the ocean".
This review is written from the perspective of a new user of the product.

**Specific comments**
Following specific issues were identified and might (if taken into account) require a revision of some aspects of the data product:

-l.412: "Neutral density was calculated using Sérazin (2011)." It should be noted that the reference given here refers to a master thesis and that the proposed polynomial approximation of neutral density in this thesis has not undergone peer review. Furthermore, polynomials were fitted to a preliminary neutral density data set with known issues (pers. comm. P. Barker and G. Sérazin). To take those limitations into account, the computed density variable gamma could either be revised, removed or labelled as preliminary in the main text.
Thank you for alerting us on this issue.
- Changes made: We have replaced the neutral density values in the merged product files with values calculated according to Jackett and McDougall (1997). This is described in Sects. 3.3.1 and 3.3.2.

-It might be helpful for some users if the f flag value would distinguish between interpolated and calculated values.
A WOCE flag value of 0 does indeed indicate values that either have been interpolated or calculated. Interpolation is only carried out for salinity, oxygen and nutrients while calculations are only carried out for seawater $CO_2$ chemistry variables. As such, interpretation of the 0 flag is unique. This is now clearly stated in the manuscript, in section 3.1, 3.3.2 and 6. Whether to change this and introduce a new flag, is a topic that will be considered for future updates.

-l.190: It is stated that "not all offsets larger than the initial minimum limits have been

adjusted for…. Conversely, in some cases where data and offsets were very precise and the cruise had been conducted in a region where variability is expected to be small, adjustments lower than the minimum limits were applied." I was wondering whether at all an initial minimum adjustment limit needs to be defined and what the added value of this definition is. Would it be possible to define an offset-to-precision ratio that could rigorously be applied to all decisions?

This is true, a limit based on the criteria mentioned (offset-to-precision ratio) seems more meaningful, and we will explore ways to implement this for future versions of this data product.

-l.249: An adjustment of -3 $\mu$mol kg$^{-1}$ was applied, although an offset of 3.68 ± 0.83 $\mu$mol kg$^{-1}$ was found. Is this difference intentional? What is the general rule on how the adjustment values are set?

Adjustments are typically round numbers relative to the precision of the variable considered. There are no particular rules about rounding down or up; we look for example, on whether there is a difference in the offset in recent vs older crossovers. We also consider additional evidence from the other methods. Here, we settled for -3 $\mu$mol/kg, as the CANYON-B and CONTENT analyses suggested a bias of 3.4 and 2.7 $\mu$mol kg$^{-1}$, respectively. This also helps to make the adjustment as small as meaningfully possible, in case there actually is an increasing trend in TCO$_2$ from uptake of anthropogenic carbon.

- Changes made: The sentence in question has been revised to : "In this case -3 $\mu$mol kg$^{-1}$ was applied. This is somewhat less than indicated by the crossover analysis, but such a small adjustment is supported by the CANYON-B and CONTENT results (Sect. 3.2.3)."

**Technical corrections**
Following comments address the presentation of the data product, and cover also aspects that are not purely technically:

-The presentation of the flagging scheme could be improved, aiming at clarity from a user perspective. Taking table 2 as an example, it confused me that labels 0-9 are presented, whereas the data product only uses f flag values 0, 2, and 9. Readers currently need to refer to footnotes in column "Merged product files" to find out that WOCE flags 6 and 7 were set to 2, whereas 3, 4, 5, and 8 were set to 9. Furthermore, the term "Not used" might add to the confusion, as it can easily be misinterpreted as "observations were not used" rather than the intended "the flag value was not used". Starting table 2 with the first column indicating f flag values that are actually used in the data product would greatly improve clarity and avoid potential misinterpretation of the flagging scheme.

We agree that this should be better described and have made changes in the text and in table 2, which hopefully convey differences in flagging schemes between the original exchange formatted data files and the merged product files.

- Changes made: The underlined text has been added to the paragraph where the WOCE flags are first mentioned in Sect. 3.1: "Each data column (except temperature and pressure, which are assumed "good" if they exist) has an associated column of data flags. For the original data exchange files, these flags conform to the WOCE definitions for water samples and are listed in Table 2. For the merged and adjusted product files these flags are simplified: questionable (WOCE flag 3) and bad (WOCE flag 4) data are removed and their flags are set to 9. The same procedure is applied to data flagged 8 (very few such data exist). WOCE flags 1 (Data not received) and 5 (Data not reported) are also set to 9, while 6 (Mean of replicate measurement) and 7 (Manual chromatographic peak measurement) are set to 2, if the data appear

good. Also, in the merged product file a flag of 0 is used to indicate a value that could be measured but is somehow approximated: for salinity, oxygen, phosphate, nitrate, and silicate, the approximation is conducted using vertical interpolation; for seawater $CO_2$ chemistry variables ($TCO_2$, TAlk, pH, and $fCO_2$), the approximation is conducted using calculation from two measured $CO_2$ chemistry variables (Sect 3.2.2). Importantly, interpolation of $CO_2$ chemistry variables is never preformed, and thus a flag value of 0 has unique interpretation."

- Changes have also been made in Table 2, specifically, we have replaced 'Not used', in the third colum, with 'Flag not used', to make it more clear that it is the flags that are not used. We prefer to leave the column order unchanged, as having the scheme for original files first and product files last, aligns with the extent to which files are modified with our procedures.

Likewise, in table 5 rownames (first column) are not intuitive. I was wondering what -888 does stand for. Does this label occur in the data set?

Reviewer 3 also pointed out lacking explanation of the -888, and similar, codes, which are used in the online Adjustment Table, and as shorthand for various actions in the manuscript. We agree these needs explanation.

- Changes made: the meaning of -888 and the other codes are now explained in a new paragraph added to the start of Section 4.2: "The secondary QC has 5 different outcomes, provided there are data. These are summarized in Table 5, along with the corresponding codes that appear in the online Adjustment Table and that are also occasionally used as shorthand for decisions in the coming text. The level of secondary QC varies among the cruises. Specifically, in some cases data were too shallow or geographically too isolated for full and conclusive consistency analyses. A secondary QC flag has been included in the merged product files to enable their identification, with "0" used for variables and cruises not subjected to full secondary QC (corresponding to code -888 in Table 5) and "1" for variables and cruises that were subjected to full secondary QC. The secondary QC flags are assigned per cruise and variable, not for individual data points and are independent of—and included in addition to—the primary (WOCE) QC flag. For example, interpolated (salinity, oxygen, nutrients) or calculated ($TCO_2$, TAlk, pH) values, which have a primary QC flag 0, may have a secondary QC flag of 1 if the measured data these values are based on have been subjected to full secondary QC. Conversely, individual data points may have a secondary QC flag of 0, even if their primary QC flag is 2 (good data). A 0 flag means that data were too shallow or geographically too isolated for consistency analyses or that these analyses were inconclusive, but that we have no reasons to believe that the data in question are of poor quality. Prominent examples of this for this version are the 10 new Davis Strait cruises: no data were available in this region in GLODAPv2.2019, which, combined with complex hydrography and differences in sampling locations, rendered conclusive secondary QC impossible. As a consequence, most, but not all, of these data (some being excluded because of poor precision after consultation with the PI) are included with a secondary QC flag of 0. "

- A new table 5 has been added:

**Table 5: Possible outcomes of the secondary QC and their codes in the online Adjustment Table**

| Secondary QC result | Code |
|---|---|
| The data are of good quality, consistent with the rest of the dataset and should not be adjusted. | 0/1[a] |
| The data are of good quality but are biased: adjust by adding (for salinity, $TCO_2$, TAlk, pH) or by multiplying (for oxygen, nutrients, CFCs) the adjustment value | Adjustment value |
| The data have not been QC'd, are of uncertain quality, and suspended until full secondary QC has been carried out | -666 |
| The data are of poor quality and excluded from the data product. | -777 |
| The data appear of good quality but their nature, being from shallow depths, coastal regions, without crossovers or similar, prohibits full secondary QC | -888 |
| No data exist for this variable for the cruise in question | -999 |

[a]The value of 0 is used for variables with additive adjustments (salinity, $TCO_2$, TAlk, pH) and 1 for variables with multiplicative adjustments (for oxygen, nutrients, CFCs). This is mathematically equivalent to 'no adjustment' in each case

Finally, several important information about flags are given in section 3.3.2 (Merging), but might be better placed in 3.1 (Data assembly and primary quality control) and 3.2 (Secondary quality control).

- Changes made. The information about WOCE flags has been added to Sect 3.1, as explained in response to Technical Correction #1, and information about the secondary QC flags has been added to paragraph 4.2, in response to Technical Correction #2

-l.45: The entire data product contains "measurements from more than 1.2 million water samples". However, this number decreases significantly when the number of available core variables is considered. As an example, I found in the merged master file <0.5 million dissolved inorganic carbon (tco2) observations and <10.000 observations with all core variables being available (in both cases ignoring f and qc flags). To this end, readers might benefit from a more detailed description of the data set. Giving expected row numbers for a few exemplary combinations of subsetting conditions would be valuable

- Changes made We have included some illustrative examples in a new paragraph in Section 6: "The total number of data records are 1 275 558. Records with measurements for all 12 core variables, salinity, oxygen, nitrate, silicate, phosphate, $TCO_2$, TAlk, pH, CFC-11, CFC-12, CFC-113, and $CCl_4$ are very rare; only 2026 records have measured data for all 12 in the merged product file (interpolated and calculated data excluded). Requiring only two measured seawater $CO_2$ chemistry variables in addition to all the other core variables brings the number of available records up to 9 230, so this is also very rare. A major limiting factor is simultaneous availability of data for all four freon species, only 26 277 records have measurements of CFC-11, CFC-12, CFC-113, and $CCl_4$ while 400 587 have data for at least one of these (not considering availability of other core variables). A total of 398 757 records have measured data for two out of the three $CO_2$ chemistry core variables. The number of measured $fCO_2$ data are 33 924; note that these data were not subjected to quality control. The number of records with measured data for salinity, oxygen, and nutrients are 798 703, while the number of records with salinity and oxygen data are 1 077 859. All of these numbers are for measured data, not interpolated or calculated values. "

References

Jackett, D. R. and McDougall, T. J.: A neutral density variable for the world's oceans, J Phys Oceanogr, 27, 237-263, 1997.